# Data Fusion for Partial Identification of Causal Effects

**Quinn Lanners**
Duke University
qml@duke.edu

**Cynthia Rudin**
Duke University
cynthia@cs.duke.edu

**Alexander Volfovsky**
Duke University
av136@duke.edu

**Harsh Parikh**
Yale University
harsh.parikh@yale.edu

## Abstract

Data fusion techniques integrate information from heterogeneous data sources to improve learning, generalization, and decision-making across data sciences. In causal inference, these methods leverage rich observational data to improve causal effect estimation, while maintaining the trustworthiness of randomized controlled trials. Existing approaches often relax the strong "no unobserved confounding" assumption by instead assuming exchangeability of counterfactual outcomes across data sources. However, when both assumptions simultaneously fail—a common scenario in practice—current methods cannot identify or estimate causal effects. We address this limitation by proposing a novel partial identification framework that enables researchers to answer key questions such as: *Is the causal effect positive/negative?* and *How severe must assumption violations be to overturn this conclusion?* Our approach introduces interpretable sensitivity parameters that quantify assumption violations and derives corresponding causal effect bounds. We develop doubly robust estimators for these bounds and operationalize breakdown frontier analysis to understand how causal conclusions change as assumption violations increase. We apply our framework to the Project STAR study, which investigates the effect of classroom size on students' third-grade standardized test performance. Our analysis reveals that the Project STAR results are robust to simultaneous violations of key assumptions, both on average and across various subgroups of interest. This strengthens confidence in the study's conclusions despite potential unmeasured biases in the data.

## 1 Introduction

Modern evidence-based decision-making increasingly relies on combining information from various sources – a practice known as *data fusion*. From integrating satellite imagery across multiple spatial resolutions (Tang et al., 2016; Yan et al., 2021) to merging genetic markers with electronic health records (Hall et al., 2016; Conroy et al., 2023; Zawistowski et al., 2023), data fusion enables more robust, generalizable, and efficient analysis (Meng et al., 2020). In causal inference, data fusion has emerged as a popular paradigm, recently recognized as one of the top ten research directions for advancing the field (Mitra et al., 2022; Bareinboim and Pearl, 2016). Data fusion approaches in causal inference have focused on generalizing or transporting evidence from experimental studies (Degtiar and Rose, 2023; Pearl, 2015; Dahabreh et al., 2019; Lu et al., 2019), precisely estimating heterogeneous causal effects (Brantner et al., 2023; Yang et al., 2023), improving efficiency (Rosenman et al., 2023; Lin et al., 2025), and mitigating estimation bias (Kallus et al., 2018; Colnet et al., 2024b), among other things.

39th Conference on Neural Information Processing Systems (NeurIPS 2025).

Consider the example of the Project STAR study, which investigated the impact of class size on students' academic performance. The dataset comprises both a randomized controlled trial (RCT) – where students were randomly assigned to different classroom sizes – and an observational cohort where students self-selected into classrooms (Mosteller, 1995; Achilles et al., 2008). While the experimental data likely ensures internal validity, it may suffer from limited external validity or generalizability (von Hippel and Wagner, 2018; Justman, 2018). Conversely, the observational data better reflects real-world settings but may suffer from unobserved confounding (Athey et al., 2020; Parikh et al., 2023). Merging the two sources can yield more precise and externally valid treatment effect estimates under milder, partially testable assumptions (Parikh et al., 2023; Wu and Yang, 2022). Specifically, the average treatment effect (ATE) becomes identifiable if *either* the RCT generalizes well to the target population or the observational data satisfies no unmeasured confounding (NUC) (Lin et al., 2025; Yang et al., 2023).

However, if both assumptions fail – i.e., the RCT lacks external validity and the observational data is confounded – then the treatment effect is no longer point-identifiable, even in the limit of infinite data. In such settings, classical estimators break down. Nevertheless, researchers may still answer important questions like: *Is the treatment effect positive?* or *How severe must assumption violations be to overturn this conclusion?* These are questions of *partial identification*, where the goal is to estimate a plausible range or *bounds* on the treatment effect rather than a single point estimate (Cornfield et al., 1959; Manski, 2003).

While a rich literature on partial identification exists, many approaches rely on strong distributional or parametric assumptions and often ignore opportunities to tighten bounds by leveraging multiple datasets (Rosenbaum and Rubin, 1983; Blackwell, 2014; Ding and VanderWeele, 2016; Bonvini and Kennedy, 2022; Nguyen et al., 2017, 2018; Nie et al., 2021; Colnet et al., 2022; Dahabreh et al., 2023; Huang, 2024). This creates a critical gap in sensitivity analysis frameworks that are both flexible and informative when combining experimental and observational data.

**Contributions.** We propose a general framework for partially identifying treatment effects by integrating complementary strengths of experimental and observational studies. Our key contributions are:

1. We introduce *interpretable sensitivity parameters*, $\gamma$ and $\rho$, that quantify the extent of external validity violations and unmeasured confounding, respectively.

2. We develop a *double machine learning estimator* based on the efficient influence function (EIF) for estimating treatment effect bounds as a function of $(\gamma, \rho)$, without relying on strong distributional or parametric assumptions.

3. We operationalize an efficient *breakdown frontier analysis*, which characterizes regions in the $(\gamma, \rho)$ space where the treatment effect remains conclusively positive (or negative) – allowing for assessment of the robustness of causal conclusions under simultaneous assumption violations.

Our framework enables comprehensive sensitivity analyses in data fusion settings, helping researchers transparently explore the consequences of assumption violations. By leveraging the internal validity of RCTs and the representativeness of observational studies, our approach yields tighter, more interpretable bounds on treatment effects and offers a principled way to assess robustness.

## 2  Preliminaries

We consider the setting where we have a sample $\mathcal{D}_n = \{1, \ldots, n\}$ of $n$ units across an experimental cohort ($\mathcal{D}_e$) and an observational study ($\mathcal{D}_o$) drawn identically and independently from $\mathcal{P}$. For each unit $i \in \mathcal{D}_n = \mathcal{D}_e \cup \mathcal{D}_o$, $S_i = \mathbb{1}[i \in \mathcal{D}_e]$ is a binary experimental cohort indicator, $T_i \in \{0, 1\}$ is the binary treatment indicator, $Y_i$ is the observed outcome, and $\mathbf{X}_i$ is the vector of pretreatment covariates. We assume the outcome space is bounded and *positive*, but note that, without loss of generality, our approach can be applied to all bounded outcome scenarios by simply shifting the outcome domain. $Y_i(0)$ and $Y_i(1)$ denote the two potential outcomes for unit $i$.

Typically, one is interested in using these datasets to estimate the following two standard estimands, namely *the average treatment effect (ATE):* $\tau = \mathbb{E}_{\mathcal{P}}[Y(1) - Y(0)]$, and the *conditional average treatment effect (CATE):* $\tau(\mathbf{x}) = \mathbb{E}_{\mathcal{P}}[Y(1) - Y(0) \mid \mathbf{X} = \mathbf{x}]$.

We assume the following standard conditions hold:

**Assumption A1.** *[SUTVA and Consistency]. There is no interference between units, there is a single version of each treatment, and for all units,* $Y_i = T_i Y_i(1) + (1 - T_i) Y_i(0)$.

**Assumption A2.** *[Treatment Positivity]. For* $s \in \{0, 1\}$ *and all* $\mathbf{x}$, $\exists c > 0$ *such that*
$c < P(T = 1 \mid \mathbf{X} = \mathbf{x}, S = s) < 1 - c$.

**Assumption A3.** *[Study Positivity]. For all* $\mathbf{x}$, $\exists c > 0$ *such that* $c < P(S = 1 \mid \mathbf{X} = \mathbf{x}) \leq 1$

**Assumption A4.** *[Internal Validity of the Experiment].* $(Y(0), Y(1)) \perp\!\!\!\perp T \mid \mathbf{X} = \mathbf{x}, S = 1$

However, we acknowledge the possibility of unobserved confounders that concurrently influence $S$, $T$, and $Y$. Due to such unobserved confounding, the following exchangeability assumptions, which are standard in the literature, may fail to hold:

**Assumption A5.** *[No Unobserved Confounding (NUC) in the Observational Data].*
$(Y(0), Y(1)) \perp\!\!\!\perp T \mid \mathbf{X} = \mathbf{x}, S = 0$

**Assumption A6.** *[Study Exchangeability].* $(Y(0), Y(1)) \perp\!\!\!\perp S \mid \mathbf{X} = \mathbf{x}$

In this paper, we explicitly consider scenarios in which A5 and A6 assumptions are simultaneously violated, thereby challenging the point identifiability of $\tau$ and $\tau(\mathbf{x})$.

**Discussion of Assumptions**. A1 imposes the standard SUTVA and consistency conditions, ensuring that potential outcomes are well-defined for each treatment level and that the observed outcome corresponds to the treatment received. A2 is the standard treatment positivity assumption, ensuring overlap between treated and control groups. A4 and A5 are structurally equivalent, differing only in the sample subset (experimental vs. observational units). Internal validity in RCTs is generally accepted due to randomization, whereas NUC is stronger, as treatment may depend on unobserved confounders.

Combining experimental and observational samples requires the additional A3 and A6 assumptions. A3 states that each unit must have a nonzero probability of being an experimental unit and is necessary to ensure overlap between the two study cohorts. A6 is the study exchangeability assumption, which states that, conditional on covariates, potential outcomes are exchangeable across studies. Like NUC, it can be a strong assumption, as study participation may depend on unobservables.

## 2.1 Quantifying Assumption Violations

We introduce two additional terms, $\rho$ and $\gamma$, that separately quantify violations of A5 and A6, respectively. The value of $\rho \geq 0$ quantifies the level of unobserved confounding in the observational data, corresponding to a violation of A5 when $\rho > 0$. The value of $\gamma \geq 0$ quantifies the difference in potential outcomes between the RCT data and the observational data, corresponding to a violation of A6 when $\gamma > 0$. Both terms report the level of violation as relative measures of the observed outcomes. For example, $\rho = 0.2$ corresponds to the setting that unobserved confounding in the observational dataset affects outcomes by 20%. In Section 4, we formally define $\rho$ and $\gamma$, expand on their interpretation, and employ them as sensitivity parameters for partial identification of $\tau$ and $\tau(\mathbf{x})$.

## 2.2 Breakdown Frontiers

Masten and Poirier (2020) introduce an approach for visualizing how conclusions about a parameter of interest vary as a set of assumptions are relaxed. We leverage this framework to plot how treatment effect conclusions change as we relax A5 and A6 via our sensitivity parameters $\rho$ and $\gamma$.

Figure 1 illustrates a breakdown frontier plot constructed by bounding the treatment effect for various $(\rho, \gamma)$ pairs. The x-axis represents violations of A5, expressed as percentages corresponding to $(100 \times \rho)\%$. Likewise, the y-axis represents violations of A6, expressed as $(100 \times \gamma)\%$. As a result, the bottom-left of the plot corresponds to stronger assumptions (small $\rho, \gamma$), and the top-right to weaker assumptions (large $\rho, \gamma$). For each $(\rho, \gamma)$ pair, we estimate upper and lower bounds on the treatment effect. These estimates divide the plot into four regions: (i) **Conclusive**: The point estimates of the upper and lower bounds are both positive (or both negative), and both confidence intervals exclude zero at the chosen confidence level (equivalent to the robust region of Masten and Poirier, 2020). (ii) **Tentative**: The point estimates of the upper and lower bounds are both positive (or both negative), but at least one

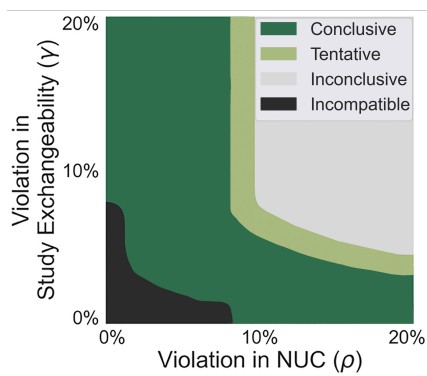

Figure 1: Example breakdown frontier plot for $\rho$ and $\gamma$.

of the corresponding confidence intervals includes zero. (iii) **Inconclusive**: The point estimates of the upper and lower bounds are not the same sign. (iv) **Incompatible**: The sensitivity parameter values lie outside the admissible range for this dataset; they imply assumptions that contradict observed discrepancies between the study groups (see Section 4). We note that Figure 1 is an example for illustration, and that the size and shape of the regions vary by dataset.

# 3 Relevant Literature

**Data Fusion for Causal Inference.** Data fusion methods leverage randomized trials to mitigate unmeasured confounding in observational data and have become central to causal inference. Broadly, approaches differ based on whether treatments and outcomes are observed only in the trial (Degtiar and Rose, 2023) or in both datasets (Brantner et al., 2023; Lin et al., 2024). Our setting aligns with the latter. A more detailed review is provided in Appendix A.

**Partial Identification.** Partial identification (ID) and sensitivity analysis frameworks are widely used to assess robustness to assumption violations in observational studies (Cornfield et al., 1959; Rosenbaum and Rubin, 1983; Liu et al., 2013; Ding and VanderWeele, 2016; Bonvini and Kennedy, 2022). In data fusion, most sensitivity approaches focus on violations of study exchangeability when observational treatments or outcomes are missing (Nguyen et al., 2017, 2018; Nie et al., 2021; Colnet et al., 2022; Dahabreh et al., 2023; Huang, 2024).

In settings where treatments and outcomes are available from the observational cohort (like ours), both non-unmeasured confounding (NUC) and study exchangeability must be addressed. Most existing work assumes exchangeability and focuses on NUC violations (Lin et al., 2024, 2025; Triantafillou et al., 2023; Chen et al., 2021; Oberst et al., 2022; Kallus et al., 2018; Yang et al., 2024; Rosenman et al., 2023; Yang et al., 2023). Yang et al. (2023) and Parikh et al. (2023) propose tests for assumption violations: the former attributing test failures to NUC, the latter recognizing that failures may stem from either assumption, though requiring knowledge of which one fails—a difficult task in practice.

**Partial Id *w/* Data Fusion.** While partial ID and sensitivity methods are well-developed for fusion without observational treatments/outcomes, they remain sparse when these are available. Related work includes partial ID approaches in contextual bandits (Joshi et al., 2024a) and structural causal modeling with qualitative knowledge (Zhang et al., 2022). Most closely related, Yu et al. (2024a) develop a two-parameter sensitivity analysis for fusion settings, addressing NUC and study positivity, but assuming study exchangeability.

# 4 Partial Identification

In this section, we present a general framework for partial identification of treatment effects under simultaneous violations of the no unmeasured confounding assumption (A5) and study exchangeability (A6). When either assumption fails, point identification of the average treatment effect (ATE) and conditional average treatment effect (CATE) becomes impossible.

To quantify the degree of these violations, we introduce two interpretable parameters: $\rho$, capturing the extent of unmeasured confounding in the observational cohort, and $\gamma$, capturing the extent of study exchangeability violation between the experimental and observational populations. We define:

$$\rho := \sup_{\mathbf{x},t} \left| 1 - \frac{\boxed{\mathbb{E}_{\mathcal{P}}[Y(t) \mid \mathbf{X} = \mathbf{x}, S = 0, T = 1 - t]}}{\boxed{\mathbb{E}_{\mathcal{P}}[Y(t) \mid \mathbf{X} = \mathbf{x}, S = 0, T = t]}} \right|, \gamma := \sup_{\mathbf{x},t} \left| 1 - \frac{\boxed{\mathbb{E}_{\mathcal{P}}[Y(t) \mid \mathbf{X} = \mathbf{x}, S = 0]}}{\boxed{\mathbb{E}_{\mathcal{P}}[Y(t) \mid \mathbf{X} = \mathbf{x}, S = 1]}} \right|.$$

The blue denominator terms represent quantities point-identifiable without assumptions A5 or A6, whereas the orange numerator terms involve counterfactual quantities that are not directly observed. If A5 or A6 holds, then $\rho = 0$ or $\gamma = 0$, respectively, and treatment effects are point-identifiable. Conversely, dissimilarity between $\mathbb{E}_{\mathcal{P}}[Y(t) \mid \mathbf{X} = \mathbf{x}, S = 0, T = 1 - t]$ and $\mathbb{E}_{\mathcal{P}}[Y(t) \mid \mathbf{X} = \mathbf{x}, S = 0, T = t]$ reflects unobserved confounding (i.e., a violation of A5) and leads to nonzero values of $\rho$. Similarly, dissimilarity between $\mathbb{E}_{\mathcal{P}}[Y(t) \mid \mathbf{X} = \mathbf{x}, S = 0]$ and $\mathbb{E}_{\mathcal{P}}[Y(t) \mid \mathbf{X} = \mathbf{x}, S = 1]$ reflects study selection bias (i.e. a violation of A6) and corresponds to nonzero values of $\gamma$. Both $\rho$ and $\gamma$ are relative measures of dissimilarity; for example, $\rho = 1$ implies a 100% difference between the observed outcome expectation and the counterfactual counterpart. Note that while $\rho$ and $\gamma$ could be extended to functions of $\mathbf{x}$ and $t$, we conservatively treat $\rho$ and $\gamma$ as scalars, taking the supremum over all covariate-treatment profiles $(\mathbf{x}, t)$. This enables tractable, worst-case sensitivity analyses.

We chose a ratio-based sensitivity model in part for its scale-invariance, which allows it to generalize across different outcome scales and covariate profiles. While alternative approaches, such as a difference-in-means model, could also be used, we see this as a modeling choice rather than a structural limitation, and our framework could accommodate such a formulation with mild adjustments.

We now focus on using $\rho$ and $\gamma$ to bound $\tau$ and $\tau(\mathbf{x})$ when A5 and A6 are simultaneously violated. For parsimony, we define the following estimable quantities: (i) study selection score: $g_s(\mathbf{x}) = \mathbb{P}_{\mathcal{P}}(S = s \mid \mathbf{X} = \mathbf{x})$, (ii) treatment propensity score: $e_t(\mathbf{x}, s) = \mathbb{P}_{\mathcal{P}}(T = t \mid \mathbf{X} = \mathbf{x}, S = s)$, and (iii) expected outcome: $\mu(\mathbf{x}, s, t) = \mathbb{E}_{\mathcal{P}}[Y \mid \mathbf{X} = \mathbf{x}, S = s, T = t]$. Using the law of iterated expectation over study selection, we express:

$$\begin{aligned}
\mathbb{E}_{\mathcal{P}}[Y(t) \mid \mathbf{X} = \mathbf{x}] &= g_1(\mathbf{x})\mathbb{E}_{\mathcal{P}}[Y(t) \mid \mathbf{X} = \mathbf{x}, S = 1] + g_0(\mathbf{x})\mathbb{E}_{\mathcal{P}}[Y(t) \mid \mathbf{X} = \mathbf{x}, S = 0] \\
&= g_1(\mathbf{x})\mu(\mathbf{x}, 1, t) + g_0(\mathbf{x})\mathbb{E}_{\mathcal{P}}[Y(t) \mid \mathbf{X} = \mathbf{x}, S = 0],
\end{aligned} \quad (1)$$

where we invoke A4 to replace the potential outcome expectation $\mathbb{E}_{\mathcal{P}}[Y(t) \mid \mathbf{X} = \mathbf{x}, S = 1]$ with the observed outcome expectation $\mu(\mathbf{x}, 1, t)$, since treatment is randomized in the experimental study $(S = 1)$. In contrast, without A5 or A6, the term $\mathbb{E}_{\mathcal{P}}[Y(t) \mid \mathbf{X} = \mathbf{x}, S = 0]$ remains unidentifiable. However, we can leverage $\rho$ and $\gamma$ to construct sharp, identifiable bounds on $\mathbb{E}_{\mathcal{P}}[Y(t) \mid \mathbf{X} = \mathbf{x}, S = 0]$, and thereby obtain bounds on the overall potential outcome $\mathbb{E}_{\mathcal{P}}[Y(t) \mid \mathbf{X} = \mathbf{x}]$. Towards this, we define two estimable functions that upper bound $\mathbb{E}_{\mathcal{P}}[Y(t) \mid \mathbf{X} = \mathbf{x}, S = 0]$:

$$v(\mathbf{x}, t, \gamma) := (1 + \gamma)\mu(\mathbf{x}, 1, t), \quad w(\mathbf{x}, t, \rho) := e_t(\mathbf{x}, 0)\mu(\mathbf{x}, 0, t) + e_{1-t}(\mathbf{x}, 0)(1 + \rho)\mu(\mathbf{x}, 0, t).$$

The function $v(\mathbf{x}, t, \gamma)$ is derived from experimental study data. It is based on the relative deviation of $\mathbb{E}_{\mathcal{P}}[Y(t) \mid \mathbf{X} = \mathbf{x}, S = 0]$ from the identifiable quantity $\mathbb{E}_{\mathcal{P}}[Y(t) \mid \mathbf{X} = \mathbf{x}, S = 1] = \mu(\mathbf{x}, 1, t)$, as governed by the parameter $\gamma$. Conversely, the function $w(\mathbf{x}, t, \rho)$ is derived from observational study data. It combines the identifiable component $\mathbb{E}_{\mathcal{P}}[Y(t) \mid \mathbf{X} = \mathbf{x}, S = 0, T = t] = \mu(\mathbf{x}, 0, t)$, weighted by the treatment propensity, $e_t(\mathbf{x}, 0)$, with a term that inflates the counterfactual component $\mathbb{E}_{\mathcal{P}}[Y(t) \mid \mathbf{X} = \mathbf{x}, S = 0, T = 1 - t]$ according to the parameter $\rho$.

In Lemma 1, we combine the upper bounds $v(\mathbf{x}, t, \gamma)$ and $w(\mathbf{x}, t, \rho)$, along with their lower bound counterparts $v(\mathbf{x}, t, -\gamma)$ and $w(\mathbf{x}, t, -\rho)$, to construct identifiable bounds on $\mathbb{E}_{\mathcal{P}}[Y(t) \mid \mathbf{X} = \mathbf{x}]$. Specifically, we replace the unidentifiable term $\mathbb{E}_{\mathcal{P}}[Y(t) \mid \mathbf{X} = \mathbf{x}, S = 0]$ in Equation 1 with the $\min$ of the two upper bounds and the $\max$ of the two lower bounds.

**Lemma 1** (Conditional Potential Outcome Bounds). *Suppose A1-A4 hold. Then for any $t \in \{0, 1\}$ and given $\mathbf{x}$, if $v(\mathbf{x}, t, -\gamma) \leq w(\mathbf{x}, t, \rho)$ and $w(\mathbf{x}, t, -\rho) \leq v(\mathbf{x}, t, \gamma)$, the conditional potential outcome satisfies*

$$\mathbb{E}_{\mathcal{P}}[Y(t) \mid \mathbf{X} = \mathbf{x}] \in [l(\mathbf{x}, t, \rho, \gamma), u(\mathbf{x}, t, \rho, \gamma)], \quad where$$
$$l(\mathbf{x}, t, \rho, \gamma) = g_1(\mathbf{x})\mu(\mathbf{x}, 1, t) + g_0(\mathbf{x}) \max\{w(\mathbf{x}, t, -\rho), v(\mathbf{x}, t, -\gamma)\},$$
$$u(\mathbf{x}, t, \rho, \gamma) = g_1(\mathbf{x})\mu(\mathbf{x}, 1, t) + g_0(\mathbf{x}) \min\{w(\mathbf{x}, t, \rho), v(\mathbf{x}, t, \gamma)\}.$$

A full derivation of the results leading to Lemma 1 is provided in Appendix B.1. Building on this results, we next derive bounds on the conditional and average treatment effects.

**Theorem 1** (Treatment Effect Bounds). *Suppose A1-A4 hold and that for each $\mathbf{x}$ and $t \in \{0, 1\}$, $v(\mathbf{x}, t, -\gamma) \leq w(\mathbf{x}, t, \rho)$ and $w(\mathbf{x}, t, -\rho) \leq v(\mathbf{x}, t, \gamma)$. Then, the conditional average treatment effect satisfies: $l(\mathbf{x}, 1, \rho, \gamma) - u(\mathbf{x}, 0, \rho, \gamma) \leq \tau(\mathbf{x}) \leq u(\mathbf{x}, 1, \rho, \gamma) - l(\mathbf{x}, 0, \rho, \gamma)$.*

*Further, if this holds for all $\mathbf{x}$ such that $\mathbb{P}_{\mathcal{P}}(\mathbf{X} = \mathbf{x}) > 0$, then the average treatment effect satisfies:*

$$\mathbb{E}_{\mathcal{P}}[l(\mathbf{X}, 1, \rho, \gamma) - u(\mathbf{X}, 0, \rho, \gamma)] \leq \tau \leq \mathbb{E}_{\mathcal{P}}[u(\mathbf{X}, 1, \rho, \gamma) - l(\mathbf{X}, 0, \rho, \gamma)].$$

We derive doubly robust estimators for the treatment effect bounds established in Theorem 1 in the next section. Before proceeding, we discuss infeasibility conditions in Remark 1.

**Remark 1** (*(In)compatible $\rho$ and $\gamma$.*). *Every $(\rho, \gamma)$ pair corresponds to a data-generating process that could, in principle, have produced the observed data. However, some values of $(\rho, \gamma)$ imply assumptions that conflict with what we observe. To illustrate this, consider a scenario where $\exists(\mathbf{x}, t)$ such that the conditional expectations of the outcome differ across study groups, i.e. $|\mathbb{E}_{\mathcal{P}}[Y \mid \mathbf{X} = \mathbf{x}, S = 1, T = t] - \mathbb{E}_{\mathcal{P}}[Y \mid \mathbf{X} = \mathbf{x}, S = 0, T = t]| = \Delta(t) > 0$. Parikh et al. (2023) shows that $\Delta(t) > 0$ implies that A5 and/or A6 is violated. Therefore, setting $(\rho, \gamma) = (0, 0)$—which implies both assumptions hold—contradicts the observed difference $\Delta(t) > 0$.*

*More broadly, the bounds in Lemma 1 and Theorem 1 are valid only when both $v(\mathbf{x}, t, -\gamma) \leq w(\mathbf{x}, t, \rho)$ and $w(\mathbf{x}, t, -\rho) \leq v(\mathbf{x}, t, \gamma)$; a violation makes the parameters incompatible. Because checking the inequalities at every $(\mathbf{x}, t)$ is infeasible in most settings, we test them in expectation over $\mathbf{X}$ for each treatment arm. The null distribution is estimated with a resampling test that keeps the fitted propensity and outcome models fixed, as generating resamples that satisfy the null and re-estimate these models is non-trivial. This may label some pairs incompatible that a full bootstrap would not. Intuition behind and estimation of (in)compatibility are discussed further in Appendix D.*

## 5 Semiparametrically Efficient Estimation

We now turn to the problem of estimating the bounds identified in Section 4. Our goal is to construct doubly robust estimators that offer both statistical efficiency and robustness to model misspecification (Chernozhukov et al., 2018). However, a key challenge arises: the presence of non-differentiable $\max$ and $\min$ operators in our estimands makes it intractable to directly derive the efficient influence functions (EIFs) needed for such estimators.

Recent work has shown that statistical functionals of this form arise in a range of modern causal inference problems. Some papers establish necessary and sufficient conditions under which such non-smooth functionals are pathwise differentiable and thus admit direct efficient estimation without smoothing (Luedtke and Van Der Laan, 2016; Bonvini and Kennedy, 2022; Kennedy et al., 2019). Other work studies smooth approximation-based strategies that replace non-differentiable operators with smooth counterparts that converge to the target functional (Levis et al., 2025; Ben-Michael, 2025). We follow the latter approach and introduce a smooth approximation based on the Boltzmann operator (Section 5.1), which provides a unified and computationally tractable way to approximate the bounds while enabling efficient estimation. This smoothing step allows us to derive the EIFs and construct bias-corrected estimators in Section 5.2. While we focus on bounds for $\tau$, the same strategy applies to bounds on $\tau(\mathbf{x})$.

### 5.1 Smooth Bounds

For any $x_1, x_2 \in \mathbb{R}$, the Boltzmann operator is of the form $\lambda_1 x_1 + \lambda_2 x_2$ where

$$\lambda_1 := \frac{\exp(\alpha x_1)}{\exp(\alpha x_1) + \exp(\alpha x_2)}, \lambda_2 := \frac{\exp(\alpha x_2)}{\exp(\alpha x_1) + \exp(\alpha x_2)}.$$

This operator is similar to the popular softmax function in that $\lambda_1 + \lambda_2 = 1$ and their relative magnitudes are linked to $x_1$ and $x_2$. However, it differs in its incorporation of the hyperparameter $\alpha$ which causes $\lambda_1 x_1 + \lambda_2 x_2$ to approach $\max(x_1, x_2)$ as $\alpha \to \infty$ and $\min(x_1, x_2)$ as $\alpha \to -\infty$.

In Lemma 2, we show that the Boltzmann operator can be used to construct smooth approximations of our partial identification bounds. Specifically, we replace the functions $l(\mathbf{X}, t, \rho, \gamma)$ and $u(\mathbf{X}, t, \rho, \gamma)$

from Lemma 1 with $b(\mathbf{X}, t, \rho, \gamma, \alpha)$, which uses the Boltzmann operator in place of the $\max$ and $\min$ functions. We then establish treatment effect bounds using these smooth approximations—analogous to Theorem 1—and show that, as $\alpha$ increases, these estimates converge to the $\max$ and $\min$ bounds.

**Lemma 2** (Smooth Bounds). *Consider a setting where A1-A4 hold but A5 and A6 may not. Define*

$$b(\mathbf{X}, t, \rho, \gamma, \alpha) := g_1(\mathbf{X})\mu(\mathbf{X}, 1, t) + g_0(\mathbf{X})\left\{\lambda_1(\mathbf{X}, t, \rho, \gamma, \alpha)v + \lambda_2(\mathbf{X}, t, \rho, \gamma, \alpha)w\right\}, \text{ where}$$

$$\lambda_1(\mathbf{X}, t, \rho, \gamma, \alpha) = \frac{\exp(\alpha v)}{\exp(\alpha v) + \exp(\alpha w)}, \quad \lambda_2(\mathbf{X}, t, \rho, \gamma, \alpha) = \frac{\exp(\alpha w)}{\exp(\alpha v) + \exp(\alpha w)},$$

*and $v = v(\mathbf{X}, t, \gamma)$, $w = w(\mathbf{X}, t, \rho)$. Then for any $\alpha > 0$, $\rho$, and $\gamma$ such that $\forall t \in \{0, 1\}$ and $\forall \mathbf{x}$ for which $\mathbb{P}_\mathcal{P}(\mathbf{X} = \mathbf{x}) > 0$, $b(\mathbf{x}, t, -\rho, -\gamma, \alpha) \le b(\mathbf{x}, t, \rho, \gamma, -\alpha)$, it follows that*

$$\mathbb{E}_\mathcal{P}[b(\mathbf{X}, 1, -\rho, -\gamma, \alpha) - b(\mathbf{X}, 0, \rho, \gamma, -\alpha)] \le \tau \le \mathbb{E}_\mathcal{P}[b(\mathbf{X}, 1, \rho, \gamma, -\alpha) - b(\mathbf{X}, 0, -\rho, -\gamma, \alpha)],$$

*and*

$$\lim_{\alpha \to \infty} \mathbb{E}_\mathcal{P}[b(\mathbf{X}, 1, -\rho, -\gamma, \alpha) - b(\mathbf{X}, 0, \rho, \gamma, -\alpha)] = \mathbb{E}_\mathcal{P}[l(\mathbf{X}, 1, \rho, \gamma) - u(\mathbf{X}, 0, \rho, \gamma)],$$

$$\lim_{\alpha \to \infty} \mathbb{E}_\mathcal{P}[b(\mathbf{X}, 1, \rho, \gamma, -\alpha) - b(\mathbf{X}, 0, -\rho, -\gamma, \alpha)] = \mathbb{E}_\mathcal{P}[u(\mathbf{X}, 1, \rho, \gamma) - l(\mathbf{X}, 0, \rho, \gamma)].$$

## 5.2 Efficient Estimators

Having established smooth, differentiable approximations of our bounds, we now derive their corresponding EIFs, which form the basis for the bias-corrected estimators. We begin by defining $\theta(t, \rho, \gamma, \alpha) := \mathbb{E}_\mathcal{P}[b(\mathbf{X}, t, \rho, \gamma, \alpha)]$. Using this, we can express the bounds on $\tau$ as

$$\theta(1, -\rho, -\gamma, \alpha) - \theta(0, \rho, \gamma, -\alpha) \le \tau \le \theta(1, \rho, \gamma, -\alpha) - \theta(0, -\rho, -\gamma, \alpha),$$

where each bound is written as a difference between two instances of $\theta$ with different parameter settings. Therefore, once we establish an EIF for the general form $\theta(t, \rho, \gamma, \alpha)$, we can obtain EIFs for both the upper and lower bounds by leveraging the linearity of EIFs.

Following the approach of Schuler and van der Laan (2024), we derive the EIF for $\theta(t, \rho, \gamma, \alpha)$, denoted by $\phi(Z; t, \rho, \gamma, \alpha)$ with $Z = (\mathbf{X}, S, T, Y)$. The full, centered EIF is given below with each term tagged by a superscript $(\cdot)$ for reference. For brevity, we omit the explicit arguments of $v$, $w$, $\lambda_1$, and $\lambda_2$, which match those of $\phi$.

$$\phi(Z; t, \rho, \gamma, \alpha) = \left[S\mu(\mathbf{X}, 1, t) + (1 - S)\{\lambda_1 v + \lambda_2 w\}\right]^{(i)} + \left[\frac{S\mathbb{I}(T = t)}{e_t(\mathbf{X}, 1)}\{Y - \mu(\mathbf{X}, 1, t)\}\right]^{(ii)}$$

$$+ \left[S(1 + \gamma)\{\lambda_1 + \alpha\lambda_1\lambda_2(v - w)\}\frac{\mathbb{I}_t(T)g_0}{e_t(\mathbf{X}, 1)g_1}\{Y - \mu(\mathbf{X}, 1, t)\}\right]^{(iii)}$$

$$+ \left[(1 - S)\{\lambda_2 + \alpha\lambda_1\lambda_2(w - v)\}\left\{\frac{\mathbb{I}_t(T)}{e_t(\mathbf{X}, 0)}\{Y - \mu(\mathbf{X}, 0, t)\}(1 + \rho e_{1-t}(\mathbf{X}, 0))\right.\right.$$

$$\left.\left. + \rho\,\mu(\mathbf{X}, 0, t)(\mathbb{I}_{1-t}(T) - e_{1-t}(\mathbf{X}, 0))\right\}\right]^{(iv)} - \theta(t, \rho, \gamma, \alpha)^{(v)}.$$

The five superscripted terms correspond to: (i) A plug-in term from the experimental and observational samples. (ii) A correction term for $g_1(\mathbf{X})\mu(\mathbf{X}, 1, t)$, using experimental samples. (iii) A correction term for $g_0(\mathbf{X})\lambda_1 v$, using experimental samples. (iv) A correction term for $g_0(\mathbf{X})\lambda_2 w$, using observational samples. (v) A centering term, $-\theta(t, \rho, \gamma, \alpha)$, to ensure $\mathbb{E}_\mathcal{P}[\phi(Z; t, \rho, \gamma, \alpha)] = 0$.

We will use the EIF for the generic $\theta(t, \rho, \gamma, \alpha)$ to obtain EIFs for the lower and upper bounds. Denoting the lower bound estimand as $\theta_{LB}(\rho, \gamma, \alpha) = \theta(1, -\rho, -\gamma, \alpha) - \theta(0, \rho, \gamma, -\alpha)$, and the upper bound as $\theta_{UB}(\rho, \gamma, \alpha) = \theta(1, \rho, \gamma, -\alpha) - \theta(0, -\rho, -\gamma, \alpha)$, their EIFs are given by

$$\phi_{LB}(Z; \rho, \gamma, \alpha) = \phi(Z; 1, -\rho, -\gamma, \alpha) - \phi(Z; 0, \rho, \gamma, -\alpha),$$
$$\phi_{UB}(Z; \rho, \gamma, \alpha) = \phi(Z; 1, \rho, \gamma, -\alpha) - \phi(Z; 0, -\rho, -\gamma, \alpha).$$

We use these EIFs to construct bias-corrected estimators for the lower and upper bounds. Let $\hat{g}_s$, $\hat{e}_t$, and $\hat{\mu}$ denote the estimated study selection, treatment propensity, and outcome regression functions used to compute $\theta_{LB}$ and $\theta_{UB}$. These are commonly referred to as nuisance functions, as they are not themselves of interest but are necessary for estimation. We collectively denote them by $\hat{\eta} = (\hat{g}_s, \hat{e}_t, \hat{\mu})$, where the hat symbol $\hat{}$ indicates an estimated quantity. The bias-corrected estimator allows these components to be estimated with flexible machine learning models, which helps protect against model misspecification, while still enabling valid inference (Chernozhukov et al., 2018). The form of the lower bound estimator is

$$\hat{\theta}_{LB}^{bc}(\rho, \gamma, \alpha; \hat{\eta}) = \hat{\theta}_{LB}^{plugin}(\rho, \gamma, \alpha; \hat{\eta}) + \frac{1}{n}\sum_{i=1}^{n}\hat{\phi}_{LB}(Z_i; \rho, \gamma, \alpha; \hat{\eta}),$$

where $\hat{\theta}_{LB}^{plugin}(\rho, \gamma, \alpha; \hat{\eta})$ is the plug-in estimate and $\hat{\phi}_{LB}(Z_i; \rho, \gamma, \alpha; \hat{\eta})$ is the corresponding centered EIF evaluated at each sample $Z_i = (\mathbf{X}_i, S_i, T_i, Y_i)$. The estimator for the upper bound is defined analogously using the corresponding plug-in and EIF components.

To ensure valid inference, we use $K$-fold cross-fitting and assume standard convergence rates on nuisance functions, leading to asymptotic normality of the bias-corrected estimators (Kennedy, 2024; Chernozhukov et al., 2018; Rudolph et al., 2025; Schuler and van der Laan, 2024). We also enforce that, in the observational arm, the estimated treatment propensities sum to one ($\hat{e}_0(\mathbf{X}, 0) + \hat{e}_1(\mathbf{X}, 0) = 1$). We present full implementation details in Algorithm 1 (Appendix E) and formally state the asymptotic properties of our estimators in Theorem 2.

**Theorem 2** (Asymptotic Properties). *Assume A1-A4 and bounded outcomes. Suppose we use cross-fitting, for all $s, t \in \{0, 1\}$ the nuisance errors satisfy*

$$\|\hat{\mu}(\mathbf{X}, s, t) - \mu(\mathbf{X}, s, t)\|_2, \quad \|\hat{e}_t(\mathbf{X}, s) - e_t(\mathbf{X}, s)\|_2, \quad \|\hat{g}_s(\mathbf{X}) - g_s(\mathbf{X})\|_2 = o_p(n^{-1/4}),$$

*where $\| \cdot \|$ is the $\mathcal{L}_2$ norm, and $\hat{e}_0(\mathbf{X}, 0) + \hat{e}_1(\mathbf{X}, 0) = 1$. Let $\alpha > 0$ be fixed, and take any $(\rho, \gamma)$ such that $\forall t \in \{0, 1\}$ and $\forall\mathbf{x}$ for which $\mathbb{P}_{\mathcal{P}}(\mathbf{X} = \mathbf{x}) > 0$, $b(\mathbf{x}, t, -\rho, -\gamma, \alpha) \leq b(\mathbf{x}, t, \rho, \gamma, -\alpha)$. Then, $\sqrt{n}\left(\hat{\theta}_{LB}^{bc}(\rho, \gamma, \alpha; \hat{\eta}) - \theta_{LB}(\rho, \gamma, \alpha)\right) \xrightarrow{d} \mathcal{N}(0, \sigma_{LB}^2)$ and $\sqrt{n}\left(\hat{\theta}_{UB}^{bc}(\rho, \gamma, \alpha; \hat{\eta}) - \theta_{UB}(\rho, \gamma, \alpha)\right) \xrightarrow{d} \mathcal{N}(0, \sigma_{UB}^2)$, where $\sigma_{LB}^2 = Var[\phi_{LB}(Z; \rho, \gamma, \alpha)]$ and $\sigma_{UB}^2 = Var[\phi_{UB}(Z; \rho, \gamma, \alpha)]$.*

Theorem 2 leverages results on estimators derived from EIFs (Kennedy, 2024; Chernozhukov et al., 2018) and establishes that our estimators for the partial identification bounds are asymptotically unbiased under standard regularity conditions. Variance can be estimated using the sample variance of the estimated influence functions or via resampling methods such as the bootstrap, enabling valid confidence interval construction. We note that while larger values of $\alpha$ yield closer approximations to the non-smooth bounds, they may also lead to estimator instability in small samples due to the increasingly steep gradients of the smoothed function near the $\max/\min$ crossover point.

## 6 Experimental Results

In this section, we bring together the partial identification bounds developed in Section 4, the estimators derived in Section 5, and the breakdown frontier plots from Masten and Poirier (2020) introduced in Section 2.2 to demonstrate how our sensitivity parameters, $\rho$ and $\gamma$, enable comprehensive sensitivity analysis. We begin with synthetic data to illustrate key properties of our framework under varying data generating processes. We then return to the Project STAR study from the Introduction, examining the robustness of treatment effect estimates in the presence of unobserved confounding. Relevant source code to implement our algorithm and replicate these results can be found in the accompanying GitHub repository.[1]

### 6.1 Simulation Study

We consider a data generating process with an unobserved confounder, $U$, which simultaneously affects study selection ($S$), treatment assignment among observational units ($T|S = 0$), and outcomes

---

[1] `https://github.com/harsh-parikh/Partial-Identification-Data-Fusion`

$(Y)$. We generate a baseline dataset with a positive treatment effect, as well as four variants where we (a) increase the treatment effect, (b) decrease the treatment effect, (c) increase the amount of unobserved confounding, and (d) decrease the amount of unobserved confounding. We plot the breakdown frontiers for each of these datasets in Figure 2.

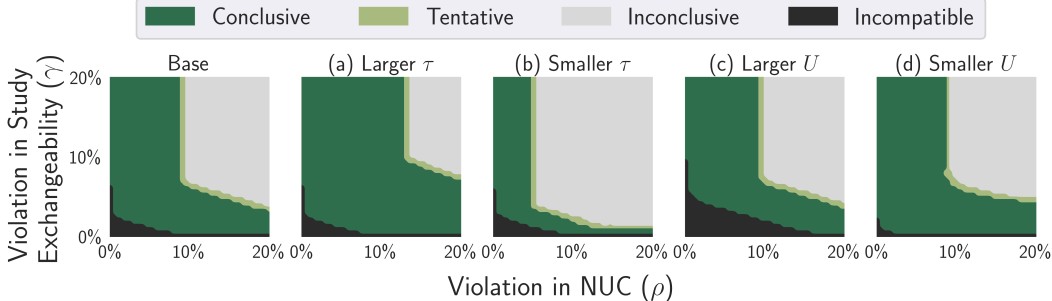

Figure 2: Breakdown frontier plots for various synthetic datasets. Figure titles indicate the relation between the data used to generate that plot to the data used to generate the *Base* plot. Conclusive and tentative regions are distinguished using 95% confidence intervals, computed from the sample variance of the efficient influence function.

We observe that the conclusive region (dark green) expands as the magnitude of the treatment effect increases. Conversely, weaker effects lead to a smaller conclusive region. We also observe that as the amount of unobserved confounding shrinks, so does the incompatible region (black), while greater confounding enlarges it. This behavior demonstrates how the breakdown frontier plot effectively summarizes the strength of evidence for a conclusive treatment effect by incorporating both the effect size and the observed discrepancies between observational and experimental data.

Specific details of the data generating process are provided in Appendix F, and the algorithm used to construct the breakdown frontier plot is presented in Appendix E.2. The procedure involves a handful of hyperparameters, including the minimum and maximum values of $\rho$ and $\gamma$, the confidence level, and the $\alpha$ scale used in the Boltzmann operator. Guidance for selecting these hyperparameters is provided in Appendix E. Additional simulation results, examining different sample sizes, choices of the $\alpha$ parameter, and cases where model assumptions are moderately violated, are reported in Appendix G.

## 6.2 Project STAR

Project STAR was a large-scale study conducted in Tennessee to investigate the effect of class size on student learning outcomes (Mosteller, 1995; Achilles et al., 2008). The experimental cohort included 11,601 students randomly assigned to one of two groups: small classes (13–17 students) $(T = 0)$ and regular classes (22–25 students) $(T = 1)$. An observational cohort of 1,780 students—assigned to the same class size types but without randomization—was also available. Demographic data, including gender, race, birth year, birth month, and free lunch eligibility, were collected for both groups. Learning outcomes were measured using standardized test administered from kindergarten through third grade. Our analysis focuses on test scores from third grade.

Figure 3(a) presents a breakdown frontier analysis of the Project STAR ATE, varying the sensitivity parameters $\rho$ and $\gamma$. The incompatible region at small values of both parameters aligns with prior findings on unmeasured confounding in the dataset (von Hippel and Wagner, 2018; Justman, 2018; Athey et al., 2020; Parikh et al., 2023). In contrast to existing estimation approaches, which require assuming either A6 or A5 holds, our framework enables investigation of causal effects under simultaneous violations of both. The analysis show that as long as study exchangeability violations remain below 5%, there is conclusive evidence of a positive ATE—even under substantial NUC violations. Given that scores range from 486–745 (mean 618), this suggests that study selection bias would need to shift outcomes by over 30 points on average to render the results inconclusive.

Beyond the population ATE, our framework supports subgroup comparisons. Figure 3(b) shows breakdown frontier plots for students who enrolled in kindergarten before age six (left) and at six or older (right). Consistent with simulation insights, the positive treatment effect is more robust to assumption violations for the older subgroup, suggesting a larger benefit for older entrants. Developmental Psychology describes significant changes in cognitive development around the typical

kindergarten entry age (Piaget, 1964), and education research has shown that students who begin kindergarten at an older age tend to experience early learning advantages (Datar, 2006). While neither directly addresses class size, these findings provide context for why older students may be better positioned to benefit from the learning environment of smaller classes—a hypothesis further supported by our analysis. The older subgroup also exhibits a larger incompatible region, potentially reflecting additional unmeasured confounding related to delayed school entry (ages six to eight) and its influence on study participation, class assignment, and outcomes.

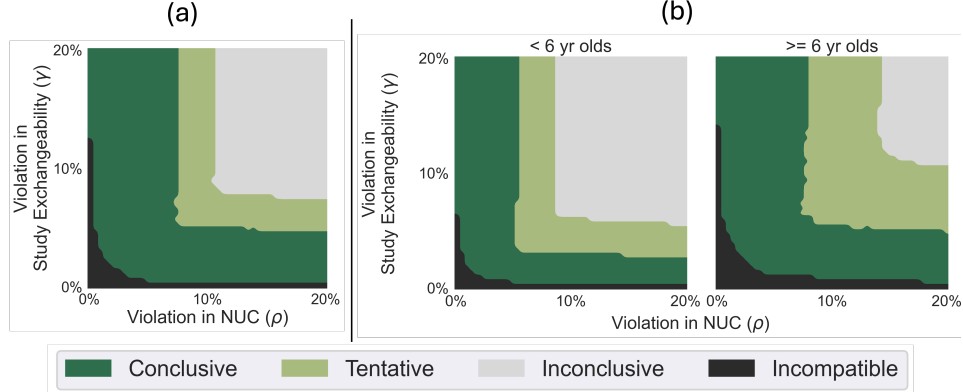

Figure 3: Breakdown frontier plots for Project STAR (a) population ATE and (b) subgroup-specific CATEs for students enrolled before age six (left) and at age six or older (right). Conclusive and tentative regions are based on 95% confidence intervals computed via bootstrap resampling.

## 7 Conclusion

Causal inference methods for data fusion typically assume either study exchangeability or NUC. Our work addresses settings where both assumptions may be violated, filling a gap in partial identification and sensitivity analysis. We introduce interpretable sensitivity parameters— $\gamma$ for external validity violations and $\rho$ for unmeasured confounding—that enable transparent robustness assessments. We derive treatment effect bounds under these parameters and develop double machine learning estimators. We use breakdown frontier plots to visualize regions where treatment effects remain conclusively positive or negative. Applications to synthetic data and Project STAR highlight our method's utility. In the Project STAR analysis, we find that the positive effect of small class sizes is robust under substantial violation of both A6 and A5. Subgroup analysis further reveals heterogeneity in this robustness, with stronger conclusions for students who enrolled at older ages.

**Alternative Estimands.** We focused on the CATE over the combined RCT *and* observational population. This perspective aligns with the generalizability framework outlined by Parikh et al. (2023), where the objective is to extend findings from an RCT to a larger cohort, rather than transport them to an external population. That said, in many settings the observational population ($S = 0$) may represent the target of interest. Our methodology is flexible with respect to the estimand, and can be adapted to this case. This involves adjusting the marginal covariate distribution in the estimand from $P(\mathbf{X})$ to $P(\mathbf{X} \mid S = 0)$, with corresponding modifications to the breakdown frontier calculation.

**Limitations & Future Work.** Our framework currently does not accommodate multiple experimental or observational datasets, continuous treatments, or dynamic treatment regimes. Extending our methodology to these settings is an important direction for future research. In addition, sharper bounds may be attainable by incorporating known outcome support and we view this as a promising direction for future research. While the Boltzmann smoothing approach provides valid and practical approximations to the sharp bounds, with Lemma 2 guaranteeing convergence as $\alpha \to \infty$, a formal characterization of the approximation error for finite $\alpha > 0$, similar to analyses in Levis et al. (2025) and Ben-Michael (2025), would further strengthen theoretical results.

Our procedure for detecting incompatible $(\rho, \gamma)$ pairs relies on a resampling test that does not account for uncertainty in nuisance function estimation (Appendix D). Improving this test is a direction for future work. Finally, our framework supports sensitivity analysis across $(\rho, \gamma)$ values but does not prescribe how to select them. Although our use of relative measures helps, domain expertise is needed to interpret plausible violation levels. Appendix E.2 discusses guidance and considerations here.

## Acknowledgments

The authors would like to thank the funders, particularly the NSF and NIH, for supporting the effort on this work. Harsh Parikh was supported by NIH NIDA R01DA056407. Quinn Lanners, Alexander Volfovsky and Cynthia Rudin were supported by NSF and Amazon under grant NSF IIS-2147061.

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

# Data Fusion for Partial Identification of Causal Effects: Appendix

## A Extended Literature Review on Data Fusion for Causal Inference

In data fusion for causal inference, it is helpful to organize prior work by the information each source contributes, the assumptions that enable transport across sources, and the inferential target (point identification, partial identification, or decision-making under uncertainty). For broad syntheses on combining randomized and observational data, external validity, efficiency/heterogeneity gains, and practical guidance, see Degtiar and Rose (2023); Colnet et al. (2024b); Brantner et al. (2023); Lin et al. (2024).

In certain scenarios randomized trial (RCT) data provide treatment $T$ and outcome $Y$, while the real-world data (RWD) only has pre-treatment covariates $X$. Typically, the task, here, is to extend internally valid trial findings to a target population whose $X$-distribution may differ. Two closely related targets are *generalization* (trial sample drawn from the target) and *transportation* (trial-eligible differs from the target) (Degtiar and Rose, 2023). Estimators typically fall into three families: reweighting/matching—most often via inverse probability of sampling weighting (IPSW) built from inclusion propensities (Cole and Stuart, 2010; Stuart et al., 2011; Buchanan et al., 2018; Colnet et al., 2024a); outcome-regression plug-in estimators fit in the trial and marginalized over the target $X$-distribution (Kern et al., 2016; Lesko et al., 2017); and doubly-robust hybrids (augmented IPSW) that achieve consistency if either the inclusion model or the outcome model is correct and are efficient when both are well-specified (Glynn and Quinn, 2010; Dahabreh et al., 2020; Colnet et al., 2024b). Identification can also be characterized graphically via selection diagrams and transport formulas (Pearl and Bareinboim, 2014; Bareinboim and Pearl, 2016). Practical diagnostics focus on limited support/overlap—when the target $X$-region is not covered by the trial (Huang et al., 2024)—and on sensitivity of transported estimates to unmeasured effect modifiers that differ across settings (Huang, 2024; Nie et al., 2021).

Additionally, data fusion literature considers the case where both the RCT and the RWD contain $T$ and $Y$. This unlocks efficiency gains and learning of heterogeneity, but raises the risk of unmeasured confounding in the observational cohort. One line of work *combines separate estimators*—e.g., shrinkage or entropy approaches that blend high-variance unbiased trial estimators with low-variance but potentially biased observational estimators (Rosenman et al., 2023; Oberst et al., 2022; Chen et al., 2021). A complementary line *bias-corrects observational learners* using experimental information—examples include variants that learn CATEs in RWD and estimate/adjust confounding bias using RCT data (Kallus et al., 2018; Yang et al., 2024; Hatt et al., 2022). A third line *trains integrative estimators jointly* on both sources, using experimental data for identification and observational data for efficiency, e.g., the Adaptive-TMLE (van der Laan et al., 2024), integrative $R$-learner (Wu and Yang, 2022) and power-likelihood methods that temper the observational contribution via a data-adaptive learning rate (Lin et al., 2025). Several of these approaches also contribute tests that decide when and how to borrow from RWD (Yang et al., 2023; Parikh et al., 2023).

Beyond this setup, another line integrates *mismatched outcomes or time horizons.* When long-term outcomes are unavailable in the RCT but short-term surrogates exist (or vice versa), one can combine experimental and observational information to identify or estimate effects on the long-term endpoint under additional structural assumptions (Athey et al., 2020; Ghassami et al., 2022). Relatedly, the surrogate-index literature develops conditions and estimators that use multiple short-term proxies to stand in for a long-term causal target (Athey et al., 2019; Kallus and Mao, 2020). A recent cautionary note shows that integrating studies with *disparate outcome measures* can help only under strong cross-study measurement assumptions and may otherwise incur bias; articulating and stress-testing these assumptions is crucial (Parikh et al., 2025). Finally, when the aim is to *predict* heterogeneous effects for a new target setting—rather than to point-identify them—multi-trial integration with uncertainty-conscious prediction intervals helps translate trial-learned heterogeneity to new populations (Brantner et al., 2025).

A complementary thread tackles *proximal/negative-control identification*, in which measured proxies enable recovery of causal effects despite unmeasured confounding. Although conceptually distinct

from transportability, proximal methods can be paired with data fusion when proxy variables are available in at least one source (Miao et al., 2018; Tchetgen Tchetgen et al., 2020). These ideas intersect with "validation-study" designs that combine a large main dataset lacking some confounders with a smaller validation set (Yang et al., 2024).

Spanning all of these settings is a growing literature on *sensitivity analysis and partial identification*. Historically, sensitivity analysis has focused on evaluating the robustness of causal conclusions in purely observational studies, for example to unmeasured confounding, while more recent work extends these ideas to settings with multiple datasets or other complex observational designs (Cornfield et al., 1959; Rosenbaum and Rubin, 1983; Liu et al., 2013; Ding and VanderWeele, 2016; Bonvini and Kennedy, 2022; Schweisthal et al., 2024). In trial-to-target settings without $T, Y$ in RWD, most sensitivity frameworks quantify departures from conditional exchangeability between study and target (Nguyen et al., 2017, 2018; Dahabreh et al., 2023; Colnet et al., 2022; Nie et al., 2021; Huang, 2024). For fusion with $T, Y$ observed in the RWD, sensitivity analysis must simultaneously confront *no-unmeasured-confounding (NUC)* violations in the observational sample and *non-exchangeability* across studies. Much of the efficiency-focused work implicitly assumes exchangeability and models violations of NUC (Lin et al., 2024, 2025; Triantafillou et al., 2023; Chen et al., 2021; Oberst et al., 2022; Kallus et al., 2018; Yang et al., 2024; Rosenman et al., 2023; Yang et al., 2023), with recent contributions proposing tests for whether borrowing from RWD is safe (Yang et al., 2023; Parikh et al., 2023). Two-parameter sensitivity frameworks that jointly benchmark internal and external validity are emerging (Yu et al., 2024b). In parallel, structural approaches provide *partial counterfactual identification* from arbitrary mixtures of observational and experimental data under qualitative model knowledge (Zhang et al., 2022). Finally, decision-making under partial identifiability—e.g., safe policy learning that maximizes guaranteed performance across feasible models—offers a principled way to act with bounds rather than point estimates (Joshi et al., 2024b).

Our contribution sits here: when both NUC and study exchangeability may fail simultaneously, we introduce interpretable sensitivity parameters, sharp bounds, doubly-robust estimators for those bounds, and breakdown-frontier analyses that characterize how conclusions evolve as the two forms of violation increase.

# B Theoretical Results

## B.1 Section 4 Proofs: Lemma 1 & Theorem 1

**Lemma 1** (Conditional Potential Outcome Bounds). *Suppose A1-A4 hold. Then for any $t \in \{0, 1\}$ and given* $\mathbf{x}$*, if $v(\mathbf{x}, t, -\gamma) \leq w(\mathbf{x}, t, \rho)$ and $w(\mathbf{x}, t, -\rho) \leq v(\mathbf{x}, t, \gamma)$, the conditional potential outcome satisfies*

$$\mathbb{E}_{\mathcal{P}}[Y(t) \mid \mathbf{X} = \mathbf{x}] \in [l(\mathbf{x}, t, \rho, \gamma), u(\mathbf{x}, t, \rho, \gamma)], \quad where$$
$$l(\mathbf{x}, t, \rho, \gamma) = g_1(\mathbf{x})\mu(\mathbf{x}, 1, t) + g_0(\mathbf{x}) \max \{w(\mathbf{x}, t, -\rho), v(\mathbf{x}, t, -\gamma)\},$$
$$u(\mathbf{x}, t, \rho, \gamma) = g_1(\mathbf{x})\mu(\mathbf{x}, 1, t) + g_0(\mathbf{x}) \min \{w(\mathbf{x}, t, \rho), v(\mathbf{x}, t, \gamma)\}.$$

*Proof.* Start by rewriting

$$\mathbb{E}_{\mathcal{P}}[Y(t) \mid \mathbf{X} = \mathbf{x}] = \mathbb{E}_{\mathcal{P}}[Y(t) \mid \mathbf{X} = \mathbf{x}, S = 1]\mathbb{P}_{\mathcal{P}}(S = 1 | \mathbf{X} = \mathbf{x})$$
$$+ \mathbb{E}_{\mathcal{P}}[Y(t) \mid \mathbf{X} = \mathbf{x}, S = 0](1 - \mathbb{P}_{\mathcal{P}}(S = 1 | \mathbf{X} = \mathbf{x}))$$

using the law of iterated expectation over study selection. By the internal calidity of the RCT treatment randomization (Assumption A4), we have that

$$\mathbb{E}_{\mathcal{P}}[Y(t) \mid \mathbf{X} = \mathbf{x}, S = 1] = \mathbb{E}_{\mathcal{P}}[Y(t) \mid \mathbf{X} = \mathbf{x}, S = 1, T = t] = \mathbb{E}_{\mathcal{P}}[Y \mid \mathbf{X} = \mathbf{x}, S = 1, T = t].$$

Plugging this in, we get

$$\mathbb{E}_{\mathcal{P}}[Y(t) \mid \mathbf{X} = \mathbf{x}] = \mathbb{E}[Y \mid \mathbf{X} = \mathbf{x}, S = 1, T = t]\mathbb{P}_{\mathcal{P}}(S = 1 | \mathbf{X} = \mathbf{x})$$
$$+ \mathbb{E}_{\mathcal{P}}[Y(t) \mid \mathbf{X} = \mathbf{x}, S = 0](1 - \mathbb{P}_{\mathcal{P}}(S = 1 | \mathbf{X} = \mathbf{x}))$$
$$= \mu(\mathbf{x}, 1, t)g_1(\mathbf{x}) + \mathbb{E}_{\mathcal{P}}[Y(t) \mid \mathbf{X} = \mathbf{x}, S = 0](1 - g_1(\mathbf{x})),$$

where $\mu(\mathbf{x}, 1, t)$ and $g_1(\mathbf{x})$ are the shorthand notation for these identifiable quantities. Our attention now turns to bounding $\mathbb{E}_{\mathcal{P}}[Y(t) \mid \mathbf{X} = \mathbf{x}, S = 0]$. Recall that $\gamma$ is defined as

$$\gamma = \sup_{\mathbf{x},t} \left| \frac{\mathbb{E}_{\mathcal{P}}[Y(t)|\mathbf{X} = \mathbf{x}, S = 1] - \mathbb{E}_{\mathcal{P}}[Y(t)|\mathbf{X} = \mathbf{x}, S = 0]}{\mathbb{E}_{\mathcal{P}}[Y(t)|\mathbf{X} = \mathbf{x}, S = 1]} \right|.$$

We again use Assumption A4 to replace $\mathbb{E}_{\mathcal{P}}[Y(t)|\mathbf{X} = \mathbf{x}, S = 1]$ with its identifiable shorthand notation $\mu(\mathbf{x}, 1, t)$. Then, for any $\mathbf{x}$ and $t \in \{0, 1\}$ we have that

$$\gamma \geq \left| \frac{\mu(\mathbf{x}, 1, t) - \mathbb{E}_{\mathcal{P}}[Y(t) \mid \mathbf{X} = \mathbf{x}, S = 0]}{\mu(\mathbf{x}, 1, t)} \right|$$
$$\gamma \cdot \mu(\mathbf{x}, 1, t) \geq |\mu(\mathbf{x}, 1, t) - \mathbb{E}_{\mathcal{P}}[Y(t) \mid \mathbf{X} = \mathbf{x}, S = 0]|$$

where we use the fact that the outcome space is positive to multiply each side by $\mu(\mathbf{x}, 1, t)$. We can bound the quantity inside the absolute value using the positive and negative versions of the left-hand side, and then rearrange terms to establish identifiable lower and upper bounds on $\mathbb{E}_{\mathcal{P}}[Y(t) \mid \mathbf{X} = \mathbf{x}, S = 0]$, as shown below:

$$-\gamma \cdot \mu(\mathbf{x}, 1, t) \leq \mu(\mathbf{x}, 1, t) - \mathbb{E}_{\mathcal{P}}[Y(t) \mid \mathbf{X} = \mathbf{x}, S = 0] \leq \gamma \cdot \mu(\mathbf{x}, 1, t)$$
$$(1 - \gamma)\mu(\mathbf{x}, 1, t) \leq \mathbb{E}_{\mathcal{P}}[Y(t) \mid \mathbf{X} = \mathbf{x}, S = 0] \leq (1 + \gamma)\mu(\mathbf{x}, 1, t)$$

We then define $v(\mathbf{x}, t, \gamma) = (1 + \gamma)\mu(\mathbf{x}, 1, t)$ and have that

$$\mathbb{E}_{\mathcal{P}}[Y(t) \mid \mathbf{X} = \mathbf{x}, S = 0] \in [v(\mathbf{x}, t, -\gamma), v(\mathbf{x}, t, \gamma)].$$

Now, we turn our objective to using $\rho$ to bound $\mathbb{E}_{\mathcal{P}}[Y(t)|\mathbf{X} = \mathbf{x}, S = 0]$. Recall that $\rho$ is defined as

$$\rho = \sup_{\mathbf{x},t} \left| \frac{\mathbb{E}_{\mathcal{P}}[Y(t)|\mathbf{X} = \mathbf{x}, S = 0, T = t] - \mathbb{E}_{\mathcal{P}}[Y(t)|\mathbf{X} = \mathbf{x}, S = 0, T = 1 - t]}{\mathbb{E}_{\mathcal{P}}[Y(t)|\mathbf{X} = \mathbf{x}, S = 0, T = t]} \right|$$
$$= \sup_{\mathbf{x},t} \left| \frac{\mu(\mathbf{x}, 0, t) - \mathbb{E}_{\mathcal{P}}[Y(t)|\mathbf{X} = \mathbf{x}, S = 0, T = 1 - t]}{\mu(\mathbf{x}, 0, t)} \right|,$$

where $\mu(\mathbf{x}, 0, t)$ is the shorthand notation for the identifiable quantity $\mathbb{E}_{\mathcal{P}}[Y(t)|\mathbf{X} = \mathbf{x}, S = 0, T = t]$. Then, for any $\mathbf{x}$ and $t \in \{0, 1\}$, using the fact that the outcome space is positive, we have that

$$\rho \geq \left| \frac{\mu(\mathbf{x}, 0, t) - \mathbb{E}_{\mathcal{P}}[Y(t)|\mathbf{X} = \mathbf{x}, S = 0, T = 1 - t]}{\mu(\mathbf{x}, 0, t)} \right|$$
$$\rho \cdot \mu(\mathbf{x}, 0, t) \geq |\mu(\mathbf{x}, 0, t) - \mathbb{E}_{\mathcal{P}}[Y(t)|\mathbf{X} = \mathbf{x}, S = 0, T = 1 - t]|.$$

Just as before, we bound the quantity inside the absolute value using the positive and negative versions of the left-hand side and rearrange terms to establish identifiable lower and upper bounds on $\mathbb{E}_{\mathcal{P}}[Y(t)|\mathbf{X} = \mathbf{x}, S = 0, T = 1 - t]$.

$$-\rho \cdot \mu(\mathbf{x}, 0, t) \leq \mu(\mathbf{x}, 0, t) - \mathbb{E}_{\mathcal{P}}[Y(t)|\mathbf{X} = \mathbf{x}, S = 0, T = 1 - t] \leq \rho \cdot \mu(\mathbf{x}, 0, t)$$
$$(1 - \rho)\mu(\mathbf{x}, 0, t) \leq \mathbb{E}_{\mathcal{P}}[Y(t)|\mathbf{X} = \mathbf{x}, S = 0, T = 1 - t] \leq (1 + \rho)\mu(\mathbf{x}, 0, t).$$

From here, we note that the bound established from $\rho$ is for the incorrect quantity. Namely, we have thus-far used $\rho$ to bound $\mathbb{E}_{\mathcal{P}}[Y(t)|\mathbf{X} = \mathbf{x}, S = 0, T = 1 - t]$, not $\mathbb{E}_{\mathcal{P}}[Y(t)|\mathbf{X} = \mathbf{x}, S = 0]$. To reconcile this, first observe that we can use the iterated expectation over treatment selection in the RWD to decompose $\mathbb{E}_{\mathcal{P}}[Y(t)|\mathbf{X} = \mathbf{x}, S = 0]$ as

$$\mathbb{E}_{\mathcal{P}}[Y(t)|\mathbf{X} = \mathbf{x}, S = 0] = \mathbb{E}_{\mathcal{P}}[Y(t) \mid \mathbf{X} = \mathbf{x}, S = 0, T = t]P(T = t \mid \mathbf{X} = \mathbf{x}, S = 0)$$
$$+ \mathbb{E}_{\mathcal{P}}[Y(t) \mid \mathbf{X} = \mathbf{x}, S = 0, T = 1 - t]P(T = 1 - t \mid \mathbf{X} = \mathbf{x}, S = 0)$$
$$= \mathbb{E}_{\mathcal{P}}[Y \mid \mathbf{X} = \mathbf{x}, S = 0, T = t]P(T = t \mid \mathbf{X} = \mathbf{x}, S = 0)$$
$$+ \mathbb{E}_{\mathcal{P}}[Y(t) \mid \mathbf{X} = \mathbf{x}, S = 0, T = 1 - t]P(T = 1 - t \mid \mathbf{X} = \mathbf{x}, S = 0)$$
$$= \mu(\mathbf{x}, 0, t)e_t(\mathbf{x}, 0) + \mathbb{E}_{\mathcal{P}}[Y(t) \mid \mathbf{X} = \mathbf{x}, S = 0, T = 1 - t](1 - e_t(\mathbf{x}, 0)),$$

where $e_t(\mathbf{x}, 0)$ are the shorthand notation for the propensity score in the RWD. Then, we can simply plug in our above bound for $\mathbb{E}_{\mathcal{P}}[Y(t) \mid \mathbf{X} = \mathbf{x}, S = 0, T = 1 - t]$ to bound $\mathbb{E}_{\mathcal{P}}[Y(t)|\mathbf{X} = \mathbf{x}, S = 0]$:

$$
\begin{aligned}
\mu(\mathbf{x}, 0, t)e_t(\mathbf{x}, 0) &+ (1 - \rho)\mu(\mathbf{x}, 0, t)(1 - e_t(\mathbf{x}, 0)) \\
&\leq \mathbb{E}_{\mathcal{P}}[Y(t) \mid \mathbf{X} = \mathbf{x}, S = 0] \\
&\leq \mu(\mathbf{x}, 0, t)e_t(\mathbf{x}, 0) + (1 + \rho)\mu(\mathbf{x}, 0, t)(1 - e_t(\mathbf{x}, 0)).
\end{aligned}
$$

We then define $w(\mathbf{x}, t, \rho) = (1 + \rho)\mu(\mathbf{x}, 0, t)(1 - e_t(\mathbf{x}, 0))$ and have that

$$
\mathbb{E}_{\mathcal{P}}[Y(t) \mid \mathbf{X} = \mathbf{x}, S = 0] \in [w(\mathbf{x}, t, -\rho), w(\mathbf{x}, t, \rho)].
$$

Now, we have used our sensitivity parameters $\gamma$ and $\rho$ to construct two intervals around $\mathbb{E}_{\mathcal{P}}[Y(t) \mid \mathbf{X} = \mathbf{x}, S = 0]$. Namely,

$$
\begin{aligned}
\mathbb{E}_{\mathcal{P}}[Y(t) \mid \mathbf{X} = \mathbf{x}, S = 0] &\in [v(\mathbf{x}, t, -\gamma), v(\mathbf{x}, t, \gamma)], \text{ and} \\
\mathbb{E}_{\mathcal{P}}[Y(t) \mid \mathbf{X} = \mathbf{x}, S = 0] &\in [w(\mathbf{x}, t, -\rho), w(\mathbf{x}, t, \rho)].
\end{aligned}
$$

We can use these two separate interval to construct a tightest interval by simply taking the maximum of the two lower bounds and the minimum of the two upper bounds. In particular, we have that

$$
\mathbb{E}_{\mathcal{P}}[Y(t) \mid \mathbf{X} = \mathbf{x}, S = 0] \in [\max\{v(\mathbf{x}, t, -\gamma), w(\mathbf{x}, t, -\rho)\}, \min\{v(\mathbf{x}, t, \gamma), w(\mathbf{x}, t, \rho)\}].
$$

From here, we note that this interval is valid if and only if $\max\{v(\mathbf{x}, t, -\gamma), w(\mathbf{x}, t, -\rho)\} \leq \min\{v(\mathbf{x}, t, \gamma), w(\mathbf{x}, t, \rho)\}$. Since $v(\mathbf{x}, t, -\gamma) \leq v(\mathbf{x}, t, \gamma)$ and $w(\mathbf{x}, t, -\rho) \leq w(\mathbf{x}, t, \rho)$ (because the outcome space is strictly positive), $\max\{v(\mathbf{x}, t, -\gamma), w(\mathbf{x}, t, -\rho)\} \leq \min\{v(\mathbf{x}, t, \gamma), w(\mathbf{x}, t, \rho)\}$ is equivalent to the condition in the lemma that $v(\mathbf{x}, t, -\gamma) \leq w(\mathbf{x}, t, \rho)$ and $w(\mathbf{x}, t, -\rho) \leq v(\mathbf{x}, t, \gamma)$.

We can then use these bounds on $\mathbb{E}_{\mathcal{P}}[Y(t) \mid \mathbf{X} = \mathbf{x}, S = 0]$ to establish that $\mathbb{E}_{\mathcal{P}}[Y(t) \mid \mathbf{X} = \mathbf{x}] \in [l(\mathbf{x}, t, \rho, \gamma), u(\mathbf{x}, t, \rho, \gamma)]$ where

$$
\begin{aligned}
l(\mathbf{x}, t, \rho, \gamma) &= g_1(\mathbf{x})\mu(\mathbf{x}, 1, t) + g_0(\mathbf{x}) \max\{w(\mathbf{x}, t, -\rho), v(\mathbf{x}, t, -\gamma)\}, \\
u(\mathbf{x}, t, \rho, \gamma) &= g_1(\mathbf{x})\mu(\mathbf{x}, 1, t) + g_0(\mathbf{x}) \min\{w(\mathbf{x}, t, \rho), v(\mathbf{x}, t, \gamma)\}.
\end{aligned}
$$

$\square$

**Theorem 1** (Treatment Effect Bounds). *Suppose A1-A4 hold and that for each $\mathbf{x}$ and $t \in \{0, 1\}$, $v(\mathbf{x}, t, -\gamma) \leq w(\mathbf{x}, t, \rho)$ and $w(\mathbf{x}, t, -\rho) \leq v(\mathbf{x}, t, \gamma)$. Then, the conditional average treatment effect satisfies:* $l(\mathbf{x}, 1, \rho, \gamma) - u(\mathbf{x}, 0, \rho, \gamma) \leq \tau(\mathbf{x}) \leq u(\mathbf{x}, 1, \rho, \gamma) - l(\mathbf{x}, 0, \rho, \gamma)$.

*Further, if this holds for all $\mathbf{x}$ such that $\mathbb{P}_{\mathcal{P}}(\mathbf{X} = \mathbf{x}) > 0$, then the average treatment effect satisfies:*

$$
\mathbb{E}_{\mathcal{P}}[l(\mathbf{X}, 1, \rho, \gamma) - u(\mathbf{X}, 0, \rho, \gamma)] \leq \tau \leq \mathbb{E}_{\mathcal{P}}[u(\mathbf{X}, 1, \rho, \gamma) - l(\mathbf{X}, 0, \rho, \gamma)].
$$

*Proof.* We first provide a proof for the bound on the CATE. Start by noting that

$$
\tau(\mathbf{x}) = \mathbb{E}_{\mathcal{P}}[Y(1) - Y(0) \mid \mathbf{X} = \mathbf{x}] = \mathbb{E}_{\mathcal{P}}[Y(1) \mid \mathbf{X} = \mathbf{x}] - \mathbb{E}_{\mathcal{P}}[Y(0) \mid \mathbf{X} = \mathbf{x}]
$$

by the linearity of expectation. Given that for both $t = 0$ and $t = 1$, $v(\mathbf{x}, t, -\gamma) \leq w(\mathbf{x}, t, \rho)$ and $w(\mathbf{x}, t, -\rho) \leq v(\mathbf{x}, t, \gamma)$, we have from Lemma 1 that

$$
\begin{aligned}
\mathbb{E}_{\mathcal{P}}[Y(1) \mid \mathbf{X} = \mathbf{x}] &\in [l(\mathbf{x}, 1, \rho, \gamma), u(\mathbf{x}, 1, \rho, \gamma)], \text{ and} \\
\mathbb{E}_{\mathcal{P}}[Y(0) \mid \mathbf{X} = \mathbf{x}] &\in [l(\mathbf{x}, 0, \rho, \gamma), u(\mathbf{x}, 0, \rho, \gamma)].
\end{aligned}
$$

From here, we note that the maximum possible value of $\tau(\mathbf{x})$ is the difference between the maximum value of $\mathbb{E}_{\mathcal{P}}[Y(1) \mid \mathbf{X} = \mathbf{x}]$ (i.e. $u(\mathbf{x}, 1, \rho, \gamma)$) and the minimum value of $\mathbb{E}_{\mathcal{P}}[Y(0) \mid \mathbf{X} = \mathbf{x}]$ (i.e. $l(\mathbf{x}, 0, \rho, \gamma)$). And coversely, the minimum possible value of $\tau(\mathbf{x})$ is the difference between the minimum value of $\mathbb{E}_{\mathcal{P}}[Y(1) \mid \mathbf{X} = \mathbf{x}]$ (i.e. $l(\mathbf{x}, 1, \rho, \gamma)$) and the maximum value of $\mathbb{E}_{\mathcal{P}}[Y(0) \mid \mathbf{X} = \mathbf{x}]$ (i.e. $u(\mathbf{x}, 0, \rho, \gamma)$). Summarizing, we can bound the conditional average treatment effect at $\mathbf{x}$ as

$$
l(\mathbf{x}, 1, \rho, \gamma) - u(\mathbf{x}, 0, \rho, \gamma) \leq \tau(\mathbf{x}) \leq u(\mathbf{x}, 1, \rho, \gamma) - l(\mathbf{x}, 0, \rho, \gamma).
$$

We now turn to proving the bound on the ATE. We start by using the law of iterated expectation to write

$$\tau = \mathbb{E}_{\mathcal{P}}[Y(1) - Y(0)]$$

$$= \int \mathbb{E}_{\mathcal{P}}[Y(1) - Y(0)|\mathbf{X} = \mathbf{x}]d\mathbb{P}_{\mathbf{X}}(\mathbf{x})$$

$$= \int \tau(\mathbf{x})d\mathbb{P}_{\mathbf{X}}(\mathbf{x})$$

where $\mathbb{P}_{\mathbf{X}}$ is the probability measure induced by $\mathbf{X}$ over $\mathcal{P}$.

Then, for any $\mathbf{x}$ where $\mathbb{P}_{\mathbf{X}}(\mathbf{x}) > 0$, because for both $t = 0$ and $t = 1$, $v(\mathbf{x}, t, -\gamma) \leq w(\mathbf{x}, t, \rho)$ and $w(\mathbf{x}, t, -\rho) \leq v(\mathbf{x}, t, \gamma)$, we have from the above result that

$$l(\mathbf{x}, 1, \rho, \gamma) - u(\mathbf{x}, 0, \rho, \gamma) \leq \tau(\mathbf{x}) \leq u(\mathbf{x}, 1, \rho, \gamma) - l(\mathbf{x}, 0, \rho, \gamma).$$

Thus,

$$\int \left(l(\mathbf{x}, 1, \rho, \gamma) - u(\mathbf{x}, 0, \rho, \gamma)\right) d\mathbb{P}_{\mathbf{X}}(\mathbf{x}) \leq \tau \leq \int \left(u(\mathbf{x}, 1, \rho, \gamma) - l(\mathbf{x}, 0, \rho, \gamma)\right) d\mathbb{P}_{\mathbf{X}}(\mathbf{x}).$$

Since, for $t \in \{0, 1\}$, $l(\mathbf{x}, t, \rho, \gamma)$ and $u(\mathbf{x}, t, \rho, \gamma)$ depend on $\mathbf{x}$, we can express the bounds in expectation notation as

$$\mathbb{E}_{\mathcal{P}}[l(\mathbf{X}, 1, \rho, \gamma) - u(\mathbf{X}, 0, \rho, \gamma)] \leq \tau \leq \mathbb{E}_{\mathcal{P}}[u(\mathbf{X}, 1, \rho, \gamma) - l(\mathbf{X}, 0, \rho, \gamma)].$$

$\square$

## B.2 Section 5 Proofs: Lemma 2 & Theorem 2

**Lemma 2** (Smooth Bounds). *Consider a setting where A1-A4 hold but A5 and A6 may not. Define*

$$b(\mathbf{X}, t, \rho, \gamma, \alpha) := g_1(\mathbf{X})\mu(\mathbf{X}, 1, t) + g_0(\mathbf{X})\left\{\lambda_1(\mathbf{X}, t, \rho, \gamma, \alpha)v + \lambda_2(\mathbf{X}, t, \rho, \gamma, \alpha)w\right\}, \text{ where}$$

$$\lambda_1(\mathbf{X}, t, \rho, \gamma, \alpha) = \frac{\exp(\alpha v)}{\exp(\alpha v) + \exp(\alpha w)}, \quad \lambda_2(\mathbf{X}, t, \rho, \gamma, \alpha) = \frac{\exp(\alpha w)}{\exp(\alpha v) + \exp(\alpha w)},$$

*and $v = v(\mathbf{X}, t, \gamma)$, $w = w(\mathbf{X}, t, \rho)$. Then for any $\alpha > 0$, $\rho$, and $\gamma$ such that $\forall t \in \{0, 1\}$ and $\forall \mathbf{x}$ for which $\mathbb{P}_{\mathcal{P}}(\mathbf{X} = \mathbf{x}) > 0$, $b(\mathbf{x}, t, -\rho, -\gamma, \alpha) \leq b(\mathbf{x}, t, \rho, \gamma, -\alpha)$, it follows that*

$$\mathbb{E}_{\mathcal{P}}[b(\mathbf{X}, 1, -\rho, -\gamma, \alpha) - b(\mathbf{X}, 0, \rho, \gamma, -\alpha)] \leq \tau \leq \mathbb{E}_{\mathcal{P}}[b(\mathbf{X}, 1, \rho, \gamma, -\alpha) - b(\mathbf{X}, 0, -\rho, -\gamma, \alpha)],$$

*and*

$$\lim_{\alpha \to \infty} \mathbb{E}_{\mathcal{P}}[b(\mathbf{X}, 1, -\rho, -\gamma, \alpha) - b(\mathbf{X}, 0, \rho, \gamma, -\alpha)] = \mathbb{E}_{\mathcal{P}}[l(\mathbf{X}, 1, \rho, \gamma) - u(\mathbf{X}, 0, \rho, \gamma)],$$

$$\lim_{\alpha \to \infty} \mathbb{E}_{\mathcal{P}}[b(\mathbf{X}, 1, \rho, \gamma, -\alpha) - b(\mathbf{X}, 0, -\rho, -\gamma, \alpha)] = \mathbb{E}_{\mathcal{P}}[u(\mathbf{X}, 1, \rho, \gamma) - l(\mathbf{X}, 0, \rho, \gamma)].$$

*Proof.* As we did using the hard $\max$ and $\min$ functions in Lemma 1, we can use the weighted Boltzmann operator quantities to bound

$$\mathbb{E}_{\mathcal{P}}[Y(t) \mid \mathbf{X} = \mathbf{x}, S = 0] \in \big[\lambda_1(\mathbf{x}, t, -\rho, -\gamma, \alpha)\, v(\mathbf{x}, t, \rho) + \lambda_2(\mathbf{x}, t, -\rho, -\gamma, \alpha)\, w(\mathbf{x}, t, \gamma),$$
$$\lambda_1(\mathbf{x}, t, \rho, \gamma, -\alpha)\, v(\mathbf{x}, t, \rho) + \lambda_2(\mathbf{x}, t, \rho, \gamma, -\alpha)\, w(\mathbf{x}, t, \gamma)\big].$$

so long as

$$\lambda_1(\mathbf{x}, t, -\rho, -\gamma, \alpha)\, v(\mathbf{x}, t, \rho) + \lambda_2(\mathbf{x}, t, -\rho, -\gamma, \alpha)\, w(\mathbf{x}, t, \gamma) \leq \lambda_1(\mathbf{x}, t, \rho, \gamma, -\alpha)\, v(\mathbf{x}, t, \rho)$$
$$+ \lambda_2(\mathbf{x}, t, \rho, \gamma, -\alpha)\, w(\mathbf{x}, t, \gamma).$$

Because $\mathbf{x}$ and $t$ are constant in this inequality, this is equivalent to the condition that

$$b(\mathbf{x}, t, -\rho, -\gamma, \alpha) \leq b(\mathbf{x}, t, \rho, \gamma, -\alpha).$$

Then, under these conditions we have that

$$\mathbb{E}_{\mathcal{P}}[Y(t) \mid \mathbf{X} = \mathbf{x}] \in [b(\mathbf{x}, t, -\rho, -\gamma, \alpha), b(\mathbf{x}, t, \rho, \gamma, -\alpha)].$$

Then, we can proceed as we did in the proof for Theorem 1 to bound $\tau(\mathbf{x})$, and subsequently, $\tau$. Namely, we note that we can bound $\tau(\mathbf{x})$ between (i) the difference of the lower bound on $\mathbb{E}_{\mathcal{P}}[Y(1) \mid \mathbf{X} = \mathbf{x}]$ and the upper bound on $\mathbb{E}_{\mathcal{P}}[Y(0) \mid \mathbf{X} = \mathbf{x}]$ and (ii) the difference between the upper bound on $\mathbb{E}_{\mathcal{P}}[Y(1) \mid \mathbf{X} = \mathbf{x}]$ and the lower bound on $\mathbb{E}_{\mathcal{P}}[Y(0) \mid \mathbf{X} = \mathbf{x}]$. Concretely, we have that

$$b(\mathbf{x}, 1, -\rho, -\gamma, \alpha) - b(\mathbf{x}, 0, \rho, \gamma, -\alpha) \leq \tau(\mathbf{x}) \leq b(\mathbf{x}, 1, \rho, \gamma, -\alpha) - b(\mathbf{x}, 0, -\rho, -\gamma, \alpha).$$

Then, since for any $\mathbf{x}$ where $\mathbb{P}_{\mathbf{X}}(\mathbf{x}) > 0$ and $t \in \{0, 1\}$, $b(\mathbf{x}, t, -\rho, -\gamma, \alpha) \leq b(\mathbf{x}, t, \rho, \gamma, -\alpha)$, we can take the expectation over the upper and lower $\tau(\mathbf{X})$ bounds as we did in the proof for Theorem 1 to conclude that

$$\mathbb{E}_{\mathcal{P}}[b(\mathbf{X}, 1, -\rho, -\gamma, \alpha) - b(\mathbf{X}, 0, \rho, \gamma, -\alpha)] \leq \tau \leq \mathbb{E}_{\mathcal{P}}[b(\mathbf{X}, 1, \rho, \gamma, -\alpha) - b(\mathbf{X}, 0, -\rho, -\gamma, \alpha)].$$

We now prove the convergence of these smooth bounds to their sharp counterparts as $\alpha \to \infty$. By the properties of the Boltzmann operator, we have that for any $\rho, \gamma, \mathbf{x}$, and $t \in \{0, 1\}$

$$\lambda_1(\mathbf{x}, t, \rho, \gamma, \alpha)v(\mathbf{x}, t, \rho) + \lambda_2(\mathbf{x}, t, \rho, \gamma, \alpha)w(\mathbf{x}, t, \gamma) \to \max\{v(\mathbf{x}, t, \rho), w(\mathbf{x}, t, \gamma)\} \text{ as } \alpha \to \infty,$$
$$\lambda_1(\mathbf{x}, t, \rho, \gamma, \alpha)v(\mathbf{x}, t, \rho) + \lambda_2(\mathbf{x}, t, \rho, \gamma, \alpha)w(\mathbf{x}, t, \gamma) \to \min\{v(\mathbf{x}, t, \rho), w(\mathbf{x}, t, \gamma)\} \text{ as } \alpha \to -\infty.$$

Then, note that

$$\begin{aligned}
g_0(\mathbf{X}) \min\{v(\mathbf{x}, t, \rho), w(\mathbf{x}, t, \gamma)\} &\leq g_0(\mathbf{X})\big[\lambda_1(\mathbf{X}, t, \rho, \gamma, \alpha)\, v(\mathbf{X}, t, \rho) \\
&\quad + \lambda_2(\mathbf{X}, t, \rho, \gamma, \alpha)\, w(\mathbf{X}, t, \gamma)\big] \\
&\leq g_0(\mathbf{X}) \max\{v(\mathbf{x}, t, \rho), w(\mathbf{x}, t, \gamma)\}
\end{aligned}$$

and

$$-\infty < E[g_0(\mathbf{X}) \min\{v(\mathbf{x}, t, \rho), w(\mathbf{x}, t, \gamma)\}] \leq E[g_0(\mathbf{X}) \max\{v(\mathbf{x}, t, \rho), w(\mathbf{x}, t, \gamma)\}] < \infty$$

because the outcome space is bounded. We then have by linearity of expectation and the dominated convergence theorem that

$$\begin{aligned}
&\lim_{\alpha \to \infty} \mathbb{E}_{\mathcal{P}}\big[b(\mathbf{X}, 1, -\rho, -\gamma, \alpha) - b(\mathbf{X}, 0, \rho, \gamma, -\alpha)\big] \\
={}& \lim_{\alpha \to \infty} \mathbb{E}_{\mathcal{P}}\big[g_1(\mathbf{X})\mu(\mathbf{X}, 1, 1) + g_0(\mathbf{X})\{\lambda_1(\mathbf{X}, 1, -\rho, -\gamma, \alpha)\, v(\mathbf{X}, 1, -\rho) \\
&\qquad\qquad + \lambda_2(\mathbf{X}, 1, -\rho, -\gamma, \alpha)\, w(\mathbf{X}, 1, -\gamma)\} \\
&\qquad\qquad - g_1(\mathbf{X})\mu(\mathbf{X}, 1, 0) - g_0(\mathbf{X})\{\lambda_1(\mathbf{X}, 0, \rho, \gamma, -\alpha)\, v(\mathbf{X}, 0, \rho) \\
&\qquad\qquad + \lambda_2(\mathbf{X}, 0, \rho, \gamma, -\alpha)\, w(\mathbf{X}, 0, \gamma)\}\big] \\
={}& \mathbb{E}_{\mathcal{P}}[g_1(\mathbf{X})\mu(\mathbf{X}, 1, 1)] + \mathbb{E}_{\mathcal{P}}\big[\lim_{\alpha \to \infty} g_0(\mathbf{X})\{\lambda_1(\mathbf{X}, 1, -\rho, -\gamma, \alpha)\, v(\mathbf{X}, 1, -\rho) \\
&\qquad\qquad + \lambda_2(\mathbf{X}, 1, -\rho, -\gamma, \alpha)\, w(\mathbf{X}, 1, -\gamma)\}\big] \\
&\quad - \mathbb{E}_{\mathcal{P}}[g_1(\mathbf{X})\mu(\mathbf{X}, 1, 0)] - \mathbb{E}_{\mathcal{P}}\big[\lim_{\alpha \to \infty} g_0(\mathbf{X})\{\lambda_1(\mathbf{X}, 0, \rho, \gamma, -\alpha)\, v(\mathbf{X}, 0, \rho) \\
&\qquad\qquad + \lambda_2(\mathbf{X}, 0, \rho, \gamma, -\alpha)\, w(\mathbf{X}, 0, \gamma)\}\big] \\
={}& \mathbb{E}_{\mathcal{P}}[g_1(\mathbf{X})\mu(\mathbf{X}, 1, 1)] + \mathbb{E}_{\mathcal{P}}\big[g_0(\mathbf{X}) \max\{v(\mathbf{x}, 1, -\rho), w(\mathbf{x}, 1, -\gamma)\}\big] \\
&\quad - \mathbb{E}_{\mathcal{P}}[g_1(\mathbf{X})\mu(\mathbf{X}, 1, 0)] - \mathbb{E}_{\mathcal{P}}\big[g_0(\mathbf{X}) \min\{v(\mathbf{x}, 0, \rho), w(\mathbf{x}, 0, \gamma)\}\big] \\
={}& \mathbb{E}_{\mathcal{P}}\big[l(\mathbf{X}, 1, \rho, \gamma)\big] - \mathbb{E}_{\mathcal{P}}\big[u(\mathbf{X}, 0, \rho, \gamma)\big] \\
={}& \mathbb{E}_{\mathcal{P}}\big[l(\mathbf{X}, 1, \rho, \gamma) - u(\mathbf{X}, 0, \rho, \gamma)\big].
\end{aligned}$$

The same steps can be done to show that

$$\lim_{\alpha \to \infty} \mathbb{E}_{\mathcal{P}}[b(\mathbf{X}, 1, \rho, \gamma, -\alpha) - b(\mathbf{X}, 0, -\rho, -\gamma, \alpha)] = \mathbb{E}_{\mathcal{P}}[u(\mathbf{X}, 1, \rho, \gamma) - l(\mathbf{X}, 0, \rho, \gamma)].$$

$\square$

**Theorem 2** (Asymptotic Properties). *Assume A1-A4 and bounded outcomes. Suppose we use cross-fitting, for all $s, t \in \{0, 1\}$ the nuisance errors satisfy*

$$\|\hat{\mu}(\mathbf{X}, s, t) - \mu(\mathbf{X}, s, t)\|_2, \quad \|\hat{e}_t(\mathbf{X}, s) - e_t(\mathbf{X}, s)\|_2, \quad \|\hat{g}_s(\mathbf{X}) - g_s(\mathbf{X})\|_2 = o_p(n^{-1/4}),$$

where $\| \cdot \|$ is the $\mathcal{L}_2$ norm, and $\hat{e}_0(\mathbf{X}, 0) + \hat{e}_1(\mathbf{X}, 0) = 1$. Let $\alpha > 0$ be fixed, and take any $(\rho, \gamma)$ such that $\forall t \in \{0, 1\}$ and $\forall \mathbf{x}$ for which $\mathbb{P}_\mathcal{P}(\mathbf{X} = \mathbf{x}) > 0$, $b(\mathbf{x}, t, -\rho, -\gamma, \alpha) \leq b(\mathbf{x}, t, \rho, \gamma, -\alpha)$. Then, $\sqrt{n}\left(\hat{\theta}^{bc}_{LB}(\rho, \gamma, \alpha; \hat{\eta}) - \theta_{LB}(\rho, \gamma, \alpha)\right) \xrightarrow{d} \mathcal{N}(0, \sigma^2_{LB})$ and $\sqrt{n}\left(\hat{\theta}^{bc}_{UB}(\rho, \gamma, \alpha; \hat{\eta}) - \theta_{UB}(\rho, \gamma, \alpha)\right) \xrightarrow{d} \mathcal{N}(0, \sigma^2_{UB})$, where $\sigma^2_{LB} = Var[\phi_{LB}(Z; \rho, \gamma, \alpha)]$ and $\sigma^2_{UB} = Var[\phi_{UB}(Z; \rho, \gamma, \alpha)]$.

*Proof.* We prove the result for $\hat{\theta}^{bc}_{LB}$ and note that the case of $\hat{\theta}^{bc}_{UB}$ is analogous. We start by writing the standard decomposition $\hat{\theta}^{bc}_{LB} - \theta_{LB} = \mathcal{P}_n \phi_{LB} + (\mathcal{P}_n - \mathcal{P})(\hat{\phi}_{LB} - \phi_{LB}) + R_{LB}$ where $\mathcal{P}$ is the true distribution and $\mathcal{P}_n$ is the empirical distribution (Schuler and van der Laan, 2024). By the central limit theorem, $\sqrt{n}(\mathcal{P}_n \phi_{LB}) \xrightarrow{d} \mathcal{N}(0, \sigma^2_{LB})$. Therefore, what is left to show is that the remaining two terms are $o_p(n^{-1/2})$.

The second term, $(\mathcal{P}_n - \mathcal{P})(\hat{\phi}_{LB} - \phi_{LB})$, is referred to as the empirical process term. Previous results have shown that this term is always $o_p(n^{-1/2})$ when $K$-fold cross-fitting is used (Chernozhukov et al., 2018; Schuler and van der Laan, 2024).

Lastly, $R_{LB}$ is referred to as the second-order remainder. This term can be decomposed as $R_{LB}(\hat{\eta}) = \hat{\theta}^{plugin}_{LB} - \theta_{LB} + \mathcal{P}\left[\hat{\phi}_{LB}\right]$ for nuisance estimates $\hat{\eta} = (\hat{g}_s, \hat{e}_t, \hat{\mu})$. In Appendix B.2.1, we decompose $R_{LB}(\hat{\eta})$ and show that under bounded outcomes, study positivity (Assumption A3), treatment positivity (Assumption A2), and the condition $\hat{e}_0(\mathbf{X}, 0) = 1 - \hat{e}_1(\mathbf{X}, 0)$, this term is a sum of products/squares of nuisance errors. Then using the assumption that for all $s, t \in \{0, 1\}$ the nuisance errors satisfy

$$\|\hat{\mu}(\mathbf{X}, s, t) - \mu(\mathbf{X}, s, t)\|_2, \quad \|\hat{e}_t(\mathbf{X}, s) - e_t(\mathbf{X}, s)\|_2, \quad \|\hat{g}_s(\mathbf{X}) - g_s(\mathbf{X})\|_2 = o_p(n^{-1/4}),$$

we obtain $R_{LB}(\hat{\eta}) = o_p(n^{-1/2})$ $\qquad \square$

### B.2.1 Decomposition of the second-order remainder term

Similar to the proof for Theorem 2, we focus here on the decomposition of the second-order remainder term for $\hat{\theta}^{plugin}_{LB}$. The steps are analogous for $\hat{\theta}^{plugin}_{UB}$.

Recall that the second-order remainder term is of the form

$$\hat{\theta}^{plugin}_{LB}(\rho, \gamma, \alpha; \hat{\eta}) - \theta_{LB}(\rho, \gamma, \alpha) + \mathbb{E}_\mathcal{P}\left[\hat{\phi}_{LB}(Z; \rho, \gamma, \alpha; \hat{\eta})\right].$$

We can expand this as

$$\left\{\hat{\theta}^{plugin}(1, -\rho, -\gamma, \alpha) - \hat{\theta}^{plugin}(0, \rho, \gamma, -\alpha)\right\}$$
$$- \left\{\theta(1, -\rho, -\gamma, \alpha) - \theta(0, \rho, \gamma, -\alpha)\right\}$$
$$+ \mathbb{E}_\mathcal{P}\left\{\hat{\phi}(Z; 1, -\rho, -\gamma, \alpha; \hat{\eta}) - \hat{\phi}(Z; 0, \rho, \gamma, -\alpha; \hat{\eta})\right\}$$

and then group the treatment arm specify terms, with the $t = 1$ terms being

$$\hat{\theta}^{plugin}(1, -\rho, -\gamma, \alpha) - \theta(1, -\rho, -\gamma, \alpha) + \mathbb{E}_\mathcal{P}\left[\hat{\phi}(Z; 1, -\rho, -\gamma, \alpha; \hat{\eta})\right]$$

and $t = 0$ terms being

$$-\hat{\theta}^{plugin}(0, \rho, \gamma, -\alpha) + \theta(0, \rho, \gamma, -\alpha) - \mathbb{E}_\mathcal{P}\left[\hat{\phi}(Z; 0, \rho, \gamma, -\alpha; \hat{\eta})\right].$$

Note that if we can show each group of terms are $o_p(n^{-1/2})$, then the sum of them is also $o_p(n^{-1/2})$. The derivation of the second-order remainder term for both treatment arms is similar, so we show the form for $t = 1$ and note that similar steps can be taken for $t = 0$. Towards this, we plug in the full

form of $\hat{\phi}(Z; 1, -\rho, -\gamma, \alpha; \hat{\eta})$,

$$\hat{\theta}^{plugin}(1, -\rho, -\gamma, \alpha) - \theta(1, -\rho, -\gamma, \alpha)$$

$$+\mathbb{E}\left\{\left[S\hat{\mu}(\mathbf{X}, 1, 1) + (1 - S)\{\hat{\lambda}_1(\mathbf{X})\hat{v}(\mathbf{X}) + \hat{\lambda}_2(\mathbf{X})\hat{w}(\mathbf{X})\}\right] + \left[\frac{S\mathbb{I}_1(T)}{\hat{e}_1(\mathbf{X}, 1)}\{Y - \hat{\mu}(\mathbf{X}, 1, 1)\}\right]\right.$$

$$+\left[S(1 - \gamma)\{\hat{\lambda}_1(\mathbf{X}) + \alpha\hat{\lambda}_1(\mathbf{X})\hat{\lambda}_2(\mathbf{X})(\hat{v}(\mathbf{X}) - \hat{w}(\mathbf{X}))\}\frac{\mathbb{I}_1(T)\hat{g}_0(\mathbf{X})}{\hat{e}_1(\mathbf{X}, 1)\hat{g}_1(\mathbf{X})}\{Y - \hat{\mu}(\mathbf{X}, 1, 1)\}\right]$$

$$+\left[(1 - S)\{\hat{\lambda}_2(\mathbf{X}) + \alpha\hat{\lambda}_1(\mathbf{X})\hat{\lambda}_2(\mathbf{X})(\hat{w}(\mathbf{X}) - \hat{v}(\mathbf{X}))\}\left\{\frac{\mathbb{I}_1(T)}{\hat{e}_1(\mathbf{X}, 0)}\{Y - \hat{\mu}(\mathbf{X}, 0, 1)\}(1 - \rho\hat{e}_0(\mathbf{X}, 0))\right.\right.$$

$$\left.\left.-\rho\,\hat{\mu}(\mathbf{X}, 0, 1)(\mathbb{I}_0(T) - \hat{e}_0(\mathbf{X}, 0))\right\}\right] - \hat{\theta}^{plugin}(1, -\rho, -\gamma, \alpha)\right\}.$$

We dropped the $\mathcal{P}$ subscript from the expectation notation and omit the $t, \rho, \gamma,$ and $\alpha$ parameter values from $v, w, \lambda_1, \lambda_2$ (given they are constant throughout) for simplicity going forward. We cancel out $\hat{\theta}^{plugin}(1, -\rho, -\gamma, \alpha)$ and write the full form of $\theta(1, -\rho, -\gamma, \alpha)$:

$$\mathbb{E}\left\{\left[S\hat{\mu}(\mathbf{X}, 1, 1) + (1 - S)\{\hat{\lambda}_1(\mathbf{X})\hat{v}(\mathbf{X}) + \hat{\lambda}_2(\mathbf{X})\hat{w}(\mathbf{X})\}\right] + \left[\frac{S\mathbb{I}_1(T)}{\hat{e}_1(\mathbf{X}, 1)}\{Y - \hat{\mu}(\mathbf{X}, 1, 1)\}\right]\right.$$

$$+\left[S(1 - \gamma)\{\hat{\lambda}_1(\mathbf{X}) + \alpha\hat{\lambda}_1(\mathbf{X})\hat{\lambda}_2(\mathbf{X})(\hat{v}(\mathbf{X}) - \hat{w}(\mathbf{X}))\}\frac{\mathbb{I}_1(T)\hat{g}_0(\mathbf{X})}{\hat{e}_1(\mathbf{X}, 1)\hat{g}_1(\mathbf{X})}\{Y - \hat{\mu}(\mathbf{X}, 1, 1)\}\right]$$

$$+\left[(1 - S)\{\hat{\lambda}_2(\mathbf{X}) + \alpha\hat{\lambda}_1(\mathbf{X})\hat{\lambda}_2(\mathbf{X})(\hat{w}(\mathbf{X}) - \hat{v}(\mathbf{X}))\}\left\{\frac{\mathbb{I}_1(T)}{\hat{e}_1(\mathbf{X}, 0)}\{Y - \hat{\mu}(\mathbf{X}, 0, 1)\}(1 - \rho\hat{e}_0(\mathbf{X}, 0))\right.\right.$$

$$\left.\left.-\rho\,\hat{\mu}(\mathbf{X}, 0, 1)(\mathbb{I}_0(T) - \hat{e}_0(\mathbf{X}, 0))\right\}\right]\right\} -$$

$$\mathbb{E}\left\{g_1(\mathbf{X})\mu(\mathbf{X}, 1, 1) + g_0(\mathbf{X})\left[\lambda_1(\mathbf{X})v(\mathbf{X}) + \lambda_2(\mathbf{X})w(\mathbf{X})\right]\right\}$$

Let's group these terms into four groups for easier analysis.

$$\mathbb{E}\left\{\left[S\hat{\mu}(\mathbf{X}, 1, 1) + \frac{S\mathbb{I}_1(T)}{\hat{e}_1(\mathbf{X}, 1)}\{Y - \hat{\mu}(\mathbf{X}, 1, 1)\} - g_1(\mathbf{X})\mu(\mathbf{X}, 1, 1)\right]^{(i)}\right.$$

$$+\left[(1 - S)\{\hat{\lambda}_1(\mathbf{X})\hat{v}(\mathbf{X}) + \hat{\lambda}_2(\mathbf{X})\hat{w}(\mathbf{X})\} - g_0(\mathbf{X})\left[\lambda_1(\mathbf{X})v(\mathbf{X}) + \lambda_2(\mathbf{X})w(\mathbf{X})\right]\right]^{(ii)}$$

$$+\left[S(1 - \gamma)\{\hat{\lambda}_1(\mathbf{X}) + \alpha\hat{\lambda}_1(\mathbf{X})\hat{\lambda}_2(\mathbf{X})(\hat{v}(\mathbf{X}) - \hat{w}(\mathbf{X}))\}\frac{\mathbb{I}_1(T)\hat{g}_0(\mathbf{X})}{\hat{e}_1(\mathbf{X}, 1)\hat{g}_1(\mathbf{X})}\{Y - \hat{\mu}(\mathbf{X}, 1, 1)\}\right]^{(iii)}$$

$$+\left[(1 - S)\{\hat{\lambda}_2(\mathbf{X}) + \alpha\hat{\lambda}_1(\mathbf{X})\hat{\lambda}_2(\mathbf{X})(\hat{w}(\mathbf{X}) - \hat{v}(\mathbf{X}))\}\left\{\frac{\mathbb{I}_1(T)}{\hat{e}_1(\mathbf{X}, 0)}\{Y - \hat{\mu}(\mathbf{X}, 0, 1)\}(1 - \rho\hat{e}_0(\mathbf{X}, 0))\right.\right.$$

$$\left.\left.\left.-\rho\,\hat{\mu}(\mathbf{X}, 0, 1)(\mathbb{I}_0(T) - \hat{e}_0(\mathbf{X}, 0))\right\}\right]^{(iv)}\right\}$$

For readability, going forward we suppress explicit arguments from the nuisance functions when analyzing each block. In particular, we let $g_s = g_s(\mathbf{X})$, $e_t^{(s)} = e_t(\mathbf{X}, s)$, $\mu_s = \mu(\mathbf{X}, s, 1)$, $v = v(\mathbf{X})$, $w = w(\mathbf{X})$, and $\lambda_a = \lambda_a(\mathbf{X})$ (and similarly for all $\hat{\ }$ equivalents). Starting with (i), note that by the law of iterated expectation we have that

$$\mathbb{E}\{S\hat{\mu}_1\} = \mathbb{E}\{\mathbb{E}[S|\mathbf{X}]\hat{\mu}_1\} = \mathbb{E}\{g_1\hat{\mu}_1\}$$

and

$$\mathbb{E}\left\{\frac{S\mathbb{I}_1(T)}{\hat{e}_1^{(1)}}\{Y - \hat{\mu}_1\}\right\} =$$

$$\mathbb{E}\left\{\frac{1}{\hat{e}_1^{(1)}}\mathbb{E}[S|\mathbf{X}]\,\mathbb{E}[\mathbb{I}_1(T) \mid \mathbf{X}, S = 1]\,(\mathbb{E}[Y \mid \mathbf{X}, S = 1, T = 1] - \hat{\mu}_1)\right\} =$$

$$\mathbb{E}\left\{\frac{g_1 e_1^{(1)}}{\hat{e}_1^{(1)}}\,(\mu_1 - \hat{\mu}_1)\right\}$$

Therefore, we have that the second-order remainder of this portion can be expressed as

$$\mathbb{E}\{(i)\} = \mathbb{E}\left\{g_1\hat{\mu}_1 + \frac{g_1 e_1^{(1)}}{\hat{e}_1^{(1)}}\{\mu_1 - \hat{\mu}_1\} - g_1\mu_1\right\} = \mathbb{E}\left\{\frac{g_1}{\hat{e}_1^{(1)}}\left(e_1^{(1)} - \hat{e}_1^{(1)}\right)(\mu_1 - \hat{\mu}_1)\right\}$$

Turning our attention term (ii), we continue to use the law of iterated expectation, but show less steps for brevity's sake. We can rewrite (ii) as:

$$\mathbb{E}\left\{g_0(\hat{h} - h)\right\}$$

where $h = \lambda_1 v + \lambda_2 w$ and $\hat{h} = \hat{\lambda}_1\hat{v} + \hat{\lambda}_2\hat{w}$. From here, define $\delta = w - v$, which allows us to write $h = v + m(\delta)$ where $m(\delta) = \lambda_2\delta = \frac{\delta}{1+\exp(-\alpha\delta)}$. Further, define $\Delta v = \hat{v} - v$, $\Delta w = \hat{w} - w$, and $\Delta\delta = \hat{\delta} - \delta = \Delta w - \Delta v$. Then, we can rewrite $\hat{h} - h = (\hat{v} - v) + (m(\hat{\delta}) - m(\delta)) = \Delta v + m(\delta + \Delta\delta) - m(\delta)$.

Using Taylor's theorem with remainder, for some $\zeta \in (\delta, \delta + \Delta\delta)$,

$$m(\delta + \Delta\delta) - m(\delta) = m'(\delta)\Delta\delta + \frac{1}{2}m''(\zeta)(\Delta\delta)^2$$

where $m'(\delta) = \lambda_2 + \alpha\lambda_1\lambda_2\delta$ and $m''(\zeta) = 2\alpha\lambda_1\lambda_2 + \alpha^2\lambda_1\lambda_2(1 - 2\lambda_2)\zeta$. Therefore, we can write

$$\hat{h} - h = \Delta v + m'(\delta)\Delta\delta + \frac{1}{2}m''(\zeta)(\Delta\delta)^2$$

$$= [\lambda_1 + \alpha\lambda_1\lambda_2(v - w)]\,\Delta v + [\lambda_2 + \alpha\lambda_1\lambda_2(w - v)]\,\Delta w + \frac{1}{2}m''(\zeta)(\Delta\delta)^2$$

$$= B_1\Delta v + B_2\Delta w + \frac{1}{2}m''(\zeta)(\Delta\delta)^2$$

where $B_1 = \lambda_1 + \alpha\lambda_1\lambda_2(v - w)$ and $B_2 = \lambda_2 + \alpha\lambda_1\lambda_2(w - v)$. The expectation of (ii) is then

$$\mathbb{E}\{(ii)\} = \mathbb{E}\{g_0\,(B_1\Delta v + B_2\Delta w)\} + \frac{1}{2}\mathbb{E}\left\{g_0 m''(\zeta)\,(\Delta w - \Delta v)^2\right\}$$

The first expectation above contains two linear components, $\mathbb{E}\{g_0 B_1\Delta v\}$ and $\mathbb{E}\{g_0 B_2\Delta w\}$, that we will subsequently show cancel out from components of terms (iii) and (iv), leaving us with

$$\mathbb{E}\{(ii)\} = \frac{1}{2}\mathbb{E}\left\{g_0 m''(\zeta)\,(\Delta w - \Delta v)^2\right\}$$

Focusing now on (iii), we apply the law of iterated expectation to rewrite this portion as

$$\mathbb{E}\left\{g_1(1 - \gamma)\{\hat{\lambda}_1 + \alpha\hat{\lambda}_1\hat{\lambda}_2(\hat{v} - \hat{w})\}\frac{\hat{g}_0}{\hat{g}_1}\frac{e_1^{(1)}}{\hat{e}_1^{(1)}}\{\mu_1 - \hat{\mu}_1\}\right\} =$$

$$\mathbb{E}\left\{g_1(1 - \gamma)\hat{B}_1\frac{\hat{g}_0}{\hat{g}_1}\frac{e_1^{(1)}}{\hat{e}_1^{(1)}}\{\mu_1 - \hat{\mu}_1\}\right\} = \mathbb{E}\left\{-\hat{B}_1\Delta v g_1\frac{\hat{g}_0}{\hat{g}_1}\frac{e_1^{(1)}}{\hat{e}_1^{(1)}}\right\}.$$

where the last equality comes from the fact that $(1 - \gamma)[\mu_1 - \hat{\mu}_1] = -\Delta v$. We can rewrite the last three components of this term as

$$g_1 \frac{\hat{g}_0}{\hat{g}_1} \frac{e_1^{(1)}}{\hat{e}_1^{(1)}} = \left( g_1 \frac{\hat{g}_0}{\hat{g}_1} - g_0 \right) \frac{e_1^{(1)}}{\hat{e}_1^{(1)}} + g_0 \frac{e_1^{(1)}}{\hat{e}_1^{(1)}}$$

$$= \left( \frac{g_1}{\hat{g}_1} - 1 \right) \frac{e_1^{(1)}}{\hat{e}_1^{(1)}} + g_0 \frac{e_1^{(1)}}{\hat{e}_1^{(1)}}$$

$$= g_0 + \left( \frac{g_1}{\hat{g}_1} - 1 \right) \frac{e_1^{(1)}}{\hat{e}_1^{(1)}} + g_0 \left( \frac{e_1^{(1)}}{\hat{e}_1^{(1)}} - 1 \right).$$

Plugging this in, we get

$$\mathbb{E}\left\{ -g_0 \hat{B}_1 \Delta v - \left( \frac{g_1}{\hat{g}_1} - 1 \right) \frac{e_1^{(1)}}{\hat{e}_1^{(1)}} \hat{B}_1 \Delta v - g_0 \left( \frac{e_1^{(1)}}{\hat{e}_1^{(1)}} - 1 \right) \hat{B}_1 \Delta v \right\} =$$

$$\mathbb{E}\left\{ -g_0 B_1 \Delta v - g_0 \left( \hat{B}_1 - B_1 \right) \Delta v - \left( \frac{g_1}{\hat{g}_1} - 1 \right) \frac{e_1^{(1)}}{\hat{e}_1^{(1)}} \hat{B}_1 \Delta v - g_0 \left( \frac{e_1^{(1)}}{\hat{e}_1^{(1)}} - 1 \right) \hat{B}_1 \Delta v \right\}$$

The first term then cancels with $\mathbb{E}\left\{ g_0 \hat{B}_1 \Delta v \right\}$ from term (ii), leaving us with

$$\mathbb{E}\{(iii)\} = -\mathbb{E}\left\{ g_0 \left( \hat{B}_1 - B_1 \right) \Delta v + \left( \frac{g_1}{\hat{g}_1} - 1 \right) \frac{e_1^{(1)}}{\hat{e}_1^{(1)}} \hat{B}_1 \Delta v + g_0 \left( \frac{e_1^{(1)}}{\hat{e}_1^{(1)}} - 1 \right) \hat{B}_1 \Delta v \right\}$$

Now, applying the law of iterated expectation to term (iv), we get

$$\mathbb{E}\left\{ g_0 \left\{ \hat{\lambda}_2 + \alpha \hat{\lambda}_1 \hat{\lambda}_2 (\hat{w} - \hat{v}) \right\} \left[ \frac{e_1^{(0)}}{\hat{e}_1^{(0)}} [\mu_0 - \hat{\mu}_0] \left[ 1 - \rho \hat{e}_0^{(0)} \right] - \rho \hat{\mu}_0 \left[ e_0^{(0)} - \hat{e}_0^{(0)} \right] \right] \right\} =$$

$$\mathbb{E}\left\{ g_0 \hat{B}_2 \left[ A [\mu_0 - \hat{\mu}_0] + \rho \hat{\mu}_0 \Delta e_0^{(0)} \right] \right\}$$

where $A = \left[ 1 - \rho \hat{e}_0^{(0)} \right] \frac{e_1^{(0)}}{\hat{e}_1^{(0)}}$ and $\Delta e_t^{(s)} = \hat{e}_t^{(s)} - e_t^{(s)}$.

With the objective of canceling out $\mathbb{E}\{g_0 B_2 \Delta w\}$ from term (ii), define $\kappa = e_1^{(0)} + (1 - \rho)e_0^{(0)}$ and $\hat{\kappa} = \hat{e}_1^{(0)} + (1 - \rho)\hat{e}_0^{(0)}$ and note that $w = \kappa\mu_0$ and $\hat{w} = \hat{\kappa}\hat{\mu}_0$. Then, $-\Delta w = \kappa\mu_0 - \hat{\kappa}\hat{\mu}_0 = \kappa(\mu_0 - \hat{\mu}_0) - \hat{\mu}_0(\hat{\kappa} - \kappa) = \kappa(\mu_0 - \hat{\mu}_0) - \hat{\mu}_0(\Delta e_1^{(0)} + (1 - \rho)\Delta e_0^{(0)})$. From here, add and subtract $\kappa(\mu_0 - \hat{\mu}_0)$ to get

$$\mathbb{E}\{(iv)\} = \mathbb{E}\left\{ g_0 \hat{B}_2 \left[ \kappa(\mu_0 - \hat{\mu}_0) + (A - \kappa)(\mu_0 - \hat{\mu}_0) + \rho \hat{\mu}_0 \Delta e_0^{(0)} \right] \right\}$$

and then replace $\kappa(\mu_0 - \hat{\mu}_0)$ with $-\Delta w + \hat{\mu}_0(\Delta e_1^{(0)} + (1 - \rho)\Delta e_0^{(0)})$ to get

$$\mathbb{E}\{(iv)\} = \mathbb{E}\left\{ g_0 \hat{B}_2 \left[ -\Delta w + \hat{\mu}_0(\Delta e_1^{(0)} + (1 - \rho)\Delta e_0^{(0)}) + (A - \kappa)(\mu_0 - \hat{\mu}_0) + \rho \hat{\mu}_0 \Delta e_0^{(0)} \right] \right\}$$

$$= \mathbb{E}\left\{ -g_0 \hat{B}_2 \Delta w + g_0 \hat{B}_2 \left[ \hat{\mu}_0(\Delta e_1^{(0)} + \Delta e_0^{(0)}) + (A - \kappa)(\mu_0 - \hat{\mu}_0) \right] \right\}$$

$$= \mathbb{E}\left\{ -g_0 B_2 \Delta w - g_0(\hat{B}_2 - B_2)\Delta w + g_0 \hat{B}_2 \left[ \hat{\mu}_0(\Delta e_1^{(0)} + \Delta e_0^{(0)}) + (A - \kappa)(\mu_0 - \hat{\mu}_0) \right] \right\}$$

As before, the first term cancels with $\mathbb{E}\{g_0 B_2 \Delta w\}$ from term (ii), leaving us with

$$\mathbb{E}\{(iv)\} = \mathbb{E}\left\{ -g_0(\hat{B}_2 - B_2)\Delta w + g_0 \hat{B}_2 \left[ \hat{\mu}_0(\Delta e_1^{(0)} + \Delta e_0^{(0)}) + (A - \kappa)(\mu_0 - \hat{\mu}_0) \right] \right\}.$$

Further observe that because $e_0^{(0)} = (1 - e_1^{(0)})$ and $\hat{e}_0^{(0)} = (1 - \hat{e}_1^{(0)})$, we have that $\Delta e_1^{(0)} + \Delta e_0^{(0)} = 0$. Therefore,

$$\mathbb{E}\{(iv)\} = \mathbb{E}\left\{ -g_0(\hat{B}_2 - B_2)\Delta w + g_0 \hat{B}_2(A - \kappa)(\mu_0 - \hat{\mu}_0) \right\}.$$

Gathering all of these terms together, we have that the second-order remainder term can be expressed as

$$
\mathbb{E}\left\{\left[\frac{g_1}{\hat{e}_1^{(1)}}\left(e_1^{(1)}-\hat{e}_1^{(1)}\right)(\mu_1-\hat{\mu}_1)\right]^{(i)}+\left[\frac{1}{2}g_0 m''(\zeta)(\Delta w-\Delta v)^2\right]^{(ii)}\right.
$$

$$
-\left[g_0\left(\hat{B}_1-B_1\right)\Delta v+\left(\frac{g_1}{\hat{g}_1}-1\right)\frac{e_1^{(1)}}{\hat{e}_1^{(1)}}\hat{B}_1\Delta v+g_0\left(\frac{e_1^{(1)}}{\hat{e}_1^{(1)}}-1\right)\hat{B}_1\Delta v\right]^{(iii)}
$$

$$
\left.+\left[-g_0(\hat{B}_2-B_2)\Delta w+g_0\hat{B}_2(A-\kappa)(\mu_0-\hat{\mu}_0)\right]^{(iv)}\right\}
$$

We will show that each of these components is $o_p(n^{-1/2})$ under our assumption that for $s,t\in\{0,1\}$ each nuisance function satisfies

$$
\|\hat{\mu}(\mathbf{X},s,t)-\mu(\mathbf{X},s,t)\|_2,\quad\|\hat{e}_t(\mathbf{X},s)-e_t(\mathbf{X},s)\|_2,\quad\|\hat{g}_s(\mathbf{X})-g_s(\mathbf{X})\|_2=o_p(n^{-1/4}).
$$

We will repeatedly use the following consequences of boundedness/positivity:

$$
\left\|\frac{g_s}{\hat{g}_s}-1\right\|_2\leq C\|\hat{g}_s-g_s\|_2=o_p(n^{-1/4}),\quad\left\|\frac{e_t^{(s)}}{\hat{e}_t^{(s)}}-1\right\|_2\leq C\|\hat{e}_t^{(s)}-e_t^{(s)}\|_2=o_p(n^{-1/4}),\quad(2)
$$

$$
\|\Delta v\|_2\leq C\|\hat{\mu}_1-\mu_1\|_2=o_p(n^{-1/4}),\tag{3}
$$

$$
\|\Delta w\|_2\leq C\big(\|\hat{\mu}_0-\mu_0\|_2+\|\hat{e}_1^{(0)}-e_1^{(0)}\|_2+\|\hat{e}_0^{(0)}-e_0^{(0)}\|_2\big)=o_p(n^{-1/4}).\tag{4}
$$

$$
\text{for }j\in\{1,2\},\quad\|\hat{B}_j-B_j\|_2\leq C\big(\|\Delta v\|_2+\|\Delta w\|_2\big)=o_p(n^{-1/4}),\tag{5}
$$

$$
|m''(\zeta)|\leq C,\tag{6}
$$

$$
\|A-\kappa\|_2\leq C\big(\|\hat{e}_1^{(0)}-e_1^{(0)}\|_2+\|\hat{e}_0^{(0)}-e_0^{(0)}\|_2\big)=o_p(n^{-1/4}).\tag{7}
$$

For (i), by Cauchy–Schwarz and treatment positivity (Assumption A2),

$$
\left|\mathbb{E}\big[\frac{g_1}{\hat{e}_1^{(1)}}(e_1^{(1)}-\hat{e}_1^{(1)})(\mu_1-\hat{\mu}_1)\big]\right|\leq C\|\hat{e}_1^{(1)}-e_1^{(1)}\|_2\,\|\hat{\mu}_1-\mu_1\|_2=o_p(n^{-1/2}).
$$

For (ii), using (3), (4), and (6) along with the fact that $(a-b)^2\leq2(a^2+b^2)$, we get

$$
\left|\frac{1}{2}\mathbb{E}\big[g_0 m''(\zeta)(\Delta w-\Delta v)^2\big]\right|\leq C\big(\|\Delta w\|_2^2+\|\Delta v\|_2^2\big)=o_p(n^{-1/2}).
$$

For (iii), there are three pieces. By (3) and (5)

$$
\left|\mathbb{E}\big[g_0(\hat{B}_1-B_1)\Delta v\big]\right|\leq\|\hat{B}_1-B_1\|_2\,\|\Delta v\|_2\leq C\big(\|\Delta v\|_2+\|\Delta w\|_2\big)\|\Delta v\|_2=o_p(n^{-1/2}).
$$

Then, by (2), (3), treatment positivity (Assumption A2), and study positivity (Assumption A3)

$$
\left|\mathbb{E}\big[(\frac{g_1}{\hat{g}_1}-1)\frac{e_1^{(1)}}{\hat{e}_1^{(1)}}\hat{B}_1\Delta v\big]\right|\leq\left\|\frac{g_1}{\hat{g}_1}-1\right\|_2\cdot\left\|\frac{e_1^{(1)}}{\hat{e}_1^{(1)}}\hat{B}_1\Delta v\right\|_2\leq C\,\|\hat{g}_1-g_1\|_2\,\|\Delta v\|_2=o_p(n^{-1/2}),
$$

and

$$
\left|\mathbb{E}\big[g_0(\frac{e_1^{(1)}}{\hat{e}_1^{(1)}}-1)\hat{B}_1\Delta v\big]\right|\leq C\,\|\hat{e}_1^{(1)}-e_1^{(1)}\|_2\,\|\Delta v\|_2=o_p(n^{-1/2}).
$$

Finally, for (iv), there are two pieces. By (4) and (5)

$$
\left|\mathbb{E}\big[g_0(\hat{B}_2-B_2)\Delta w\big]\right|\leq\|\hat{B}_2-B_2\|_2\|\Delta w\|_2\leq C\big(\|\Delta v\|_2+\|\Delta w\|_2\big)\|\Delta w\|_2=o_p(n^{-1/2}),
$$

and, then using (7),

$$
\left|\mathbb{E}\big[g_0\hat{B}_2(A-\kappa)(\mu_0-\hat{\mu}_0)\big]\right|\leq C\|\hat{\mu}_0-\mu_0\|_2\big(\|\hat{e}_1^{(0)}-e_1^{(0)}\|_2+\|\hat{e}_0^{(0)}-e_0^{(0)}\|_2\big)=o_p(n^{-1/2}).
$$

Combining (i)–(iv) gives $|R_{2nd}| \leq C R_n^2 + o_p(n^{-1/2})$ with

$$R_n := \|\hat{\mu}_1 - \mu_1\|_2 + \|\hat{\mu}_0 - \mu_0\|_2 + \|\hat{e}_1 - e_1\|_2 + \|\hat{e}_0 - e_0\|_2 + \|\hat{g}_1 - g_1\|_2 = o_p(n^{-\frac{1}{4}}),$$

hence $R_{2nd} = o_p(n^{-1/2})$.

Assume: (a) bounded outcomes so $|\mu| \leq M$; (b) positivity, i.e., for some $\underline{c} \in (0, 1/2)$, $\underline{c} \leq e_t(\mathbf{X}, s), g_s(\mathbf{X}) \leq 1 - \underline{c}$ a.s.; (c) cross-fitting; (d) propensity normalization in the observational arm: $\hat{e}_0(\mathbf{X}, 0) = 1 - \hat{e}_1(\mathbf{X}, 0)$; (e) fixed $\alpha$.

# C   EIF Derivation

## C.1   Setup Recap

We start by reiterating our setup and relevant notation. Recall that we have the following variables,

- $\mathbf{X}$: A vector of pretreatment covariates.
- $S \in \{0, 1\}$: A binary variable indicating study assignment (RWD vs RCT data).
- $T \in \{0, 1\}$: A binary treatment indicator.
- $Y \in \mathbb{R}^+$: The outcome of interest.

The distribution of the population is denoted by $\mathcal{P}$ over $(\mathbf{X}, S, T, Y)$, which we assume admits a probability density function, denoted by $p(\mathbf{X}, S, T, Y)$.

We then defined the quantities (i) study selection score: $g_s(\mathbf{x}) = \mathbb{P}_{\mathcal{P}}(S = s \mid \mathbf{X} = \mathbf{x})$, (ii) treatment propensity score: $e_t(\mathbf{x}, s) = \mathbb{P}_{\mathcal{P}}(T = t \mid \mathbf{X} = \mathbf{x}, S = s)$, and (iii) expected outcome: $\mu(\mathbf{x}, s, t) = \mathbb{E}_{\mathcal{P}}[Y \mid \mathbf{X} = \mathbf{x}, S = s, T = t]$.

And, using our sensitivity parameters $\gamma$ and $\rho$ we constructed the following terms:

$$w(\mathbf{X}, t, \rho) := e_t(\mathbf{X}, 0)\mu(\mathbf{X}, 0, t) + \big(1 - e_t(\mathbf{X}, 0)\big)(1 + \rho)\mu(\mathbf{X}, 0, t),$$
$$v(\mathbf{X}, t, \gamma) := (1 + \gamma)\mu(\mathbf{X}, 1, t),$$
$$\lambda_1(\mathbf{X}, t, \rho, \gamma, \alpha) := \frac{\exp(\alpha v(\mathbf{X}, t, \gamma))}{\exp(\alpha v(\mathbf{X}, t, \gamma)) + \exp(\alpha w(\mathbf{X}, t, \rho))},$$
$$\lambda_2(\mathbf{X}, t, \rho, \gamma, \alpha) := \frac{\exp(\alpha w(\mathbf{X}, t, \rho))}{\exp(\alpha v(\mathbf{X}, t, \gamma)) + \exp(\alpha w(\mathbf{X}, t, \rho))}.$$

Our goal is to derive an efficient influence function for $\theta(t, \rho, \gamma, \alpha)$, where

$$\theta(t, \rho, \gamma, \alpha) =$$
$$\mathbb{E}_{\mathcal{P}}\left[g_1(\mathbf{X})\mu(\mathbf{X}, t, 1) + g_0(\mathbf{X})\left\{\lambda_1(\mathbf{X}, t, \rho, \gamma, \alpha)v(\mathbf{X}, t, \gamma) + \lambda_2(\mathbf{X}, t, \rho, \gamma, \alpha)w(\mathbf{X}, t, \rho)\right\}\right].$$

Note that we can write

$$\theta(t, \rho, \gamma, \alpha) = \theta_1(t) + \theta_2(t, \rho, \gamma, \alpha),$$

where:

$$\theta_1(t) = \mathbb{E}\left[g_1(\mathbf{X})\mu(\mathbf{X}, 1, t)\right], \text{ and}$$

$$\theta_2(t, \rho, \gamma, \alpha) = \mathbb{E}\left[g_0(\mathbf{X})\left\{\lambda_1(\mathbf{X}, t, \rho, \gamma, \alpha)v(\mathbf{X}, t, \gamma) + \lambda_2(\mathbf{X}, t, \rho, \gamma, \alpha)w(\mathbf{X}, t, \rho)\right\}\right].$$

## C.2   Efficient Influence Function

We use point mass contamination to derive a candidate EIF, $\phi(Z; t, \rho, \gamma, \alpha)$, for our estimand, $\theta(t, \rho, \gamma, \alpha)$. We proceed to verify that the candidate is indeed a valid influence function, and

subsequently the *efficient* influence function because we are in a fully-saturated model space, by confirming that for any generic score, $h$, we have

$$\nabla_h \theta(t, \rho, \gamma, \alpha) = \mathbb{E}_{\mathcal{P}}[\phi(Z; t, \rho, \gamma, \alpha)h].$$

We start by letting $\Phi$ be an EIF operator which "takes a parameter and returns its efficient influence function" (Schuler and van der Laan, 2024). In our case, $\Phi(\theta(t, \rho, \gamma, \alpha)) = \phi(Z; t, \rho, \gamma, \alpha)$.

Using this notation, we can break our estimand of interest into separate components for easier derivation. We start by noting that thanks to the linearity property of the EIF

$$\Phi(\theta(t, \rho, \gamma, \alpha)) = \Phi(\theta_1(t)) + \Phi(\theta_2(t, \rho, \gamma, \alpha)).$$

Before proceeding to derive $\Phi(\theta_1(t))$ and $\Phi(\theta_2(t, \rho, \gamma, \alpha))$, we first compute $\Phi(\cdot)$ for various sub-components that make up our estimand.

$$\Phi(p(\mathbf{x})) = \mathbb{I}_{\mathbf{x}}(\mathbf{X}) - p(\mathbf{x}),$$

$$\Phi(g_s(\mathbf{x})) = \frac{\mathbb{I}_{\mathbf{x}}(\mathbf{X})}{p(\mathbf{x})} \left[ \mathbb{I}_s(S) - g_s(\mathbf{x}) \right],$$

$$\Phi(e_t(\mathbf{x}, s)) = \frac{\mathbb{I}_{\mathbf{x}}(\mathbf{X})\mathbb{I}_s(S)}{p(\mathbf{x}, s)} \left[ \mathbb{I}_t(T) - e_t(\mathbf{x}, s) \right],$$

$$\Phi(\mu(\mathbf{x}, s, t)) = \frac{\mathbb{I}_{\mathbf{x}}(\mathbf{X})\mathbb{I}_s(S)\mathbb{I}_t(T)}{p(\mathbf{x}, s, t)} \left[ Y - \mu(\mathbf{x}, s, t) \right].$$

In the equations above, and throughout the remainder of this section, $\mathbb{I}_a(A)$ denotes an indicator function that equals 1 when the random variable $A$ takes the value $a$, and 0 otherwise.

### C.2.1   Candidate EIF for $\theta_1$

Taking the point mass contamination approach, we first rewrite $\theta_1(t)$ assuming all of our covariates are discrete. Namely,

$$\theta_1(t) = \sum_{\mathbf{x}} g_1(\mathbf{x})\mu(\mathbf{x}, 1, t)p(\mathbf{x}).$$

We apply the product rule and plug in values to calculate $\Phi(\theta_1(t))$ below.

$$\Phi(\theta_1(t)) = \sum_{\mathbf{x}} \Phi(g_1(\mathbf{x}))\mu(\mathbf{x}, 1, t)p(\mathbf{x}) + g_1(\mathbf{x})\Phi(\mu(\mathbf{x}, 1, t))p(\mathbf{x}) + g_1(\mathbf{x})\mu(\mathbf{x}, 1, t)\Phi(p(\mathbf{x}))$$

$$= \sum_{\mathbf{x}} \frac{\mathbb{I}_{\mathbf{x}}(\mathbf{X})}{p(\mathbf{x})} \left[ S - g_1(\mathbf{x}) \right] \mu(\mathbf{x}, 1, t)p(\mathbf{x})$$

$$+ \sum_{\mathbf{x}} \frac{S\mathbb{I}_{\mathbf{x}}(\mathbf{X})\mathbb{I}_t(T)}{p(\mathbf{x}, 1, t)} \left[ Y - \mu(\mathbf{x}, 1, t) \right] g_1(\mathbf{x})p(\mathbf{x}) + \left[ \mathbb{I}_{\mathbf{x}}(\mathbf{X}) - p(\mathbf{x}) \right] g_1(\mathbf{x})\mu(\mathbf{x}, 1, t)$$

$$= \sum_{\mathbf{x}} \frac{S\mathbb{I}_{\mathbf{x}}(\mathbf{X})\mathbb{I}_t(T)}{e_t(\mathbf{x}, 1)} \left[ Y - \mu(\mathbf{x}, 1, t) \right] + \mathbb{I}_{\mathbf{x}}(\mathbf{X})S\mu(\mathbf{x}, 1, t) - g_1(\mathbf{x})\mu(\mathbf{x}, 1, t)p(\mathbf{x})$$

$$= \frac{S\mathbb{I}_t(T)}{e_t(\mathbf{X}, 1)} \left[ Y - \mu(\mathbf{X}, 1, t) \right] + S\mu(\mathbf{X}, 1, t) - \theta_1(t).$$

In summary, the candidate EIF for $\theta_1(t)$ is:

$$\phi_{\theta_1} = \frac{S\mathbb{I}(T = t)}{e_t(\mathbf{X}, 1)} \Big[ Y - \mu(\mathbf{X}, 1, t) \Big] + S\mu(\mathbf{X}, 1, t) - \theta_1(t).$$

We drop the arguments for $\phi$ and simply denote the portion of EIF for $\theta_1(t)$ as $\phi_{\theta_1}$ for simplicity.

### C.2.2 Candidate EIF for $\theta_2$

Since $\gamma$, $\rho$, and $\alpha$ are predefined hyperparameters, while deriving the candidate EIF for $\theta_2$ we omit them as arguments to $w, v, \lambda_1$ and $\lambda_2$ for brevity. Importantly, these hyperparameters will become relevant when deriving the EIF, so they are still present in each function - just omitted for notational brevity.

We again start by rewriting $\theta_2$ assuming all of our covariates are discrete.

$$\theta_2 = \sum_{\mathbf{x}} g_0(\mathbf{x}) \left\{ \lambda_1(\mathbf{x})v(\mathbf{x}) + \lambda_2(\mathbf{x})w(\mathbf{x}) \right\} p(\mathbf{x}) = \sum_{\mathbf{x}} g_0(\mathbf{x}) \left\{ v(\mathbf{x}) + \lambda_2(\mathbf{x}) \left[ w(\mathbf{x}) - v(\mathbf{x}) \right] \right\} p(\mathbf{x}).$$

Then, we can apply the product rule to see that

$$\begin{aligned}
\Phi(\theta_2) = &\sum_{\mathbf{x}} \Phi(g_0(\mathbf{x}))p(\mathbf{x}) \left\{ v(\mathbf{x}) + \lambda_2(\mathbf{x}) \left[ w(\mathbf{x}) - v(\mathbf{x}) \right] \right\} \\
&+ \sum_{\mathbf{x}} \Phi(v(\mathbf{x}))g_0(\mathbf{x})p(\mathbf{x}) \left\{ 1 - \lambda_2(\mathbf{x}) \right\} \\
&+ \sum_{\mathbf{x}} \Phi(w(\mathbf{x}))g_0(\mathbf{x})p(\mathbf{x})\lambda_2(\mathbf{x}) \\
&+ \sum_{\mathbf{x}} \Phi(\lambda_2(\mathbf{x}))g_0(\mathbf{x})p(\mathbf{x}) \left\{ w(\mathbf{x}) - v(\mathbf{x}) \right\} \\
&+ \sum_{\mathbf{x}} \Phi(p(\mathbf{x}))g_0(\mathbf{x}) \left\{ v(\mathbf{x}) + \lambda_2(\mathbf{x}) \left[ w(\mathbf{x}) - v(\mathbf{x}) \right] \right\}.
\end{aligned}$$

We already know $\Phi(g_0(\mathbf{x}))$ and $\Phi(p(\mathbf{x}))$. But we need to calculate $\Phi(v(\mathbf{x}))$, $\Phi(w(\mathbf{x}))$, and $\Phi(\lambda_2(\mathbf{x}))$. First, note that we can rewrite

$$w(\mathbf{x}) = w(\mathbf{x}, t, \rho) = e_t(\mathbf{x}, 0)\mu(\mathbf{x}, 0, t) + e_{1-t}(\mathbf{x}, 0)(1+\rho)\mu(\mathbf{x}, 0, t) = \mu(\mathbf{x}, 0, t) \left[ 1 + \rho e_{1-t}(\mathbf{x}, 0) \right].$$

Then,

$$\begin{aligned}
\Phi(v(\mathbf{x})) =& \Phi(v(\mathbf{x}, t, \gamma)) \\
=& \Phi\left( (1+\gamma)\mu(\mathbf{x}, 1, t) \right) \\
=& (1+\gamma)\Phi\left( \mu(\mathbf{x}, 1, t) \right) \\
=& (1+\gamma)\frac{S\mathbb{I}_{\mathbf{x}}(\mathbf{X})\mathbb{I}_t(T)}{p(\mathbf{x}, 1, t)} \left[ Y - \mu(\mathbf{x}, 1, t) \right],
\end{aligned}$$

$$\begin{aligned}
\Phi(w(\mathbf{x})) =& w(\mathbf{x}, t, \rho) \\
=& \Phi\left( \mu(\mathbf{x}, 0, t) \left[ 1 + \rho e_{1-t}(\mathbf{x}, 0) \right] \right) \\
=& \Phi\left( \mu(\mathbf{x}, 0, t) \right) \left[ 1 + \rho e_{1-t}(\mathbf{x}, 0) \right] + \rho\mu(\mathbf{x}, 0, t)\Phi(e_{1-t}(\mathbf{x}, 0)) \\
=& \left[ 1 + \rho e_{1-t}(\mathbf{x}, 0) \right] \left( \frac{(1-S)\mathbb{I}_{\mathbf{x}}(\mathbf{X})\mathbb{I}_t(T)}{p(\mathbf{x}, 0, t)} \left[ Y - \mu(\mathbf{x}, 0, t) \right] \right) \\
&+ \rho\mu(\mathbf{x}, 0, t) \left( \frac{(1-S)\mathbb{I}_{\mathbf{x}}(\mathbf{X})}{p(\mathbf{x}, 0)} \left[ \mathbb{I}_{1-t}(T) - e_{1-t}(\mathbf{x}, 0) \right] \right),
\end{aligned}$$

and

$$\Phi(\lambda_2(\mathbf{x})) = \Phi\left(\frac{\exp(\alpha w(\mathbf{x}))}{\exp(\alpha v(\mathbf{x})) + \exp(\alpha w(\mathbf{x}))}\right)$$

$$
\begin{aligned}
&= \alpha \exp(\alpha w(\mathbf{x})) \left\{\exp(\alpha v(\mathbf{x})) + \exp(\alpha w(\mathbf{x}))\right\}^{-2} \\
&\quad \times \left\{\Phi(w(\mathbf{x}))\left[\exp(\alpha v(\mathbf{x})) + \exp(\alpha w(\mathbf{x}))\right] - \right. \\
&\qquad \left. \Phi(v(\mathbf{x}))\exp(\alpha v(\mathbf{x})) - \Phi(w(\mathbf{x}))\exp(\alpha w(\mathbf{x}))\right\} \\
&= \alpha \exp(\alpha w(\mathbf{x})) \left\{\exp(\alpha v(\mathbf{x})) + \exp(\alpha w(\mathbf{x}))\right\}^{-2} \\
&\quad \times \left\{\Phi(w(\mathbf{x}))\exp(\alpha v(\mathbf{x})) - \Phi(v(\mathbf{x}))\exp(\alpha v(\mathbf{x}))\right\} \\
&= \alpha\lambda_2(\mathbf{x})(1 - \lambda_2(\mathbf{x}))\left[\Phi(w(\mathbf{x})) - \Phi(v(\mathbf{x}))\right] \\
&= \alpha\lambda_1(\mathbf{x})\lambda_2(\mathbf{x})\left[\Phi(w(\mathbf{x})) - \Phi(v(\mathbf{x}))\right].
\end{aligned}
$$

Now, we plug $\Phi(\lambda_2(\mathbf{x}))$ in to the full $\Phi(\theta_2)$ and, because $\Phi(\lambda_2(\mathbf{x}))$ is composed of $\Phi(v(\mathbf{x}))$ and $\Phi(w(\mathbf{x}))$, combine like terms.

$$
\begin{aligned}
\Phi(\theta_2) =& \sum_{\mathbf{x}} \Phi(g_0(\mathbf{x}))p(\mathbf{x})\left\{v(\mathbf{x}) + \lambda_2(\mathbf{x})\left[w(\mathbf{x}) - v(\mathbf{x})\right]\right\} \\
&+ \sum_{\mathbf{x}} \Phi(v(\mathbf{x}))g_0(\mathbf{x})p(\mathbf{x})\left\{1 - \lambda_2(\mathbf{x})\right\} \\
&+ \sum_{\mathbf{x}} \Phi(w(\mathbf{x}))g_0(\mathbf{x})p(\mathbf{x})\lambda_2(\mathbf{x}) \\
&+ \sum_{\mathbf{x}} \left\{\alpha\lambda_1(\mathbf{x})\lambda_2(\mathbf{x})\left[\Phi(w(\mathbf{x})) - \Phi(v(\mathbf{x}))\right]\right\}g_0(\mathbf{x})p(\mathbf{x})\left\{w(\mathbf{x}) - v(\mathbf{x})\right\} \\
&+ \sum_{\mathbf{x}} \Phi(p(\mathbf{x}))g_0(\mathbf{x})\left\{v(\mathbf{x}) + \lambda_2(\mathbf{x})\left[w(\mathbf{x}) - v(\mathbf{x})\right]\right\} \\
=& \sum_{\mathbf{x}} \Phi(g_0(\mathbf{x}))p(\mathbf{x})\left\{v(\mathbf{x}) + \lambda_2(\mathbf{x})\left[w(\mathbf{x}) - v(\mathbf{x})\right]\right\} \\
&+ \sum_{\mathbf{x}} \Phi(v(\mathbf{x}))g_0(\mathbf{x})p(\mathbf{x})\left\{\lambda_1(\mathbf{x}) + \alpha\lambda_1(\mathbf{x})\lambda_2(\mathbf{x})\left[v(\mathbf{x}) - w(\mathbf{x})\right]\right\} \\
&+ \sum_{\mathbf{x}} \Phi(w(\mathbf{x}))g_0(\mathbf{x})p(\mathbf{x})\left\{\lambda_2(\mathbf{x}) + \alpha\lambda_1(\mathbf{x})\lambda_2(\mathbf{x})\left[w(\mathbf{x}) - v(\mathbf{x})\right]\right\} \\
&+ \sum_{\mathbf{x}} \Phi(p(\mathbf{x}))g_0(\mathbf{x})\left\{v(\mathbf{x}) + \lambda_2(\mathbf{x})\left[w(\mathbf{x}) - v(\mathbf{x})\right]\right\}.
\end{aligned}
$$

We now plug in the corresponding values for the remaining $\Phi(\cdot)$'s in $\Phi(\theta_2)$.

$$
\begin{aligned}
\Phi(\theta_2) =& \sum_{\mathbf{x}} \left\{\frac{\mathbb{I}_{\mathbf{x}}(\mathbf{X})}{p(\mathbf{x})}\left[1 - S - g_0(\mathbf{x})\right]\right\}p(\mathbf{x})\left\{v(\mathbf{x}) + \lambda_2(\mathbf{x})\left[w(\mathbf{x}) - v(\mathbf{x})\right]\right\} \\
&+ \sum_{\mathbf{x}} \left\{(1 + \gamma)\frac{S\mathbb{I}_{\mathbf{x}}(\mathbf{X})\mathbb{I}_t(T)}{p(\mathbf{x}, 1, t)}\left[Y - \mu(\mathbf{x}, 1, t)\right]\right\}g_0(\mathbf{x})p(\mathbf{x}) \\
&\qquad \times \left\{\lambda_1(\mathbf{x}) + \alpha\lambda_1(\mathbf{x})\lambda_2(\mathbf{x})\left[v(\mathbf{x}) - w(\mathbf{x})\right]\right\} \\
&+ \sum_{\mathbf{x}} \left\{\left[1 + \rho e_{1-t}(\mathbf{x}, 0)\right]\left(\frac{(1 - S)\mathbb{I}_{\mathbf{x}}(\mathbf{X})\mathbb{I}_t(T)}{p(\mathbf{x}, 0, t)}\left[Y - \mu(\mathbf{x}, 0, t)\right]\right)\right. \\
&\qquad \left. + \rho\mu(\mathbf{x}, 0, t)\left(\frac{(1 - S)\mathbb{I}_{\mathbf{x}}(\mathbf{X})}{p(\mathbf{x}, 0)}\left[\mathbb{I}_{1-t}(T) - e_{1-t}(\mathbf{x}, 0)\right]\right)\right\} \\
&\qquad \times g_0(\mathbf{x})p(\mathbf{x})\left\{\lambda_2(\mathbf{x}) + \alpha\lambda_1(\mathbf{x})\lambda_2(\mathbf{x})\left[w(\mathbf{x}) - v(\mathbf{x})\right]\right\} \\
&+ \sum_{\mathbf{x}} \left\{\mathbb{I}_{\mathbf{x}}(\mathbf{X}) - p(\mathbf{x})\right\}g_0(\mathbf{x})\left\{v(\mathbf{x}) + \lambda_2(\mathbf{x})\left[w(\mathbf{x}) - v(\mathbf{x})\right]\right\}.
\end{aligned}
$$

By applying the indicator function and simplifying, we get

$$
\begin{aligned}
\Phi(\theta_2) = & \left[1 - S - g_0(\mathbf{X})\right]\left\{v(\mathbf{X}) + \lambda_2(\mathbf{X})\left[w(\mathbf{X}) - v(\mathbf{X})\right]\right\} \\
& + (1 + \gamma)\frac{S\mathbb{I}_t(T)}{e_t(\mathbf{X}, 1)g_1(\mathbf{X})}\left[Y - \mu(\mathbf{X}, 1, t)\right]g_0(\mathbf{X}) \\
& \qquad \times \left\{\lambda_1(\mathbf{X}) + \alpha\lambda_1(\mathbf{X})\lambda_2(\mathbf{X})\left[v(\mathbf{X}) - w(\mathbf{X})\right]\right\} \\
& + (1 - S)\left\{\lambda_2(\mathbf{X}) + \alpha\lambda_1(\mathbf{X})\lambda_2(\mathbf{X})\left[w(\mathbf{X}) - v(\mathbf{X})\right]\right\} \\
& \qquad \times \left\{\frac{\mathbb{I}_t(T)}{e_t(\mathbf{X}, 0)}\left[Y - \mu(\mathbf{X}, 0, t)\right]\left[1 + \rho e_{1-t}(\mathbf{X}, 0)\right]\right. \\
& \qquad \qquad \left. + \rho\mu(\mathbf{X}, 0, t)\left[\mathbb{I}_{1-t}(T) - e_{1-t}(\mathbf{X}, 0)\right]\right\} \\
& + g_0(\mathbf{X})\left\{v(\mathbf{X}) + \lambda_2(\mathbf{X})\left[w(\mathbf{X}) - v(\mathbf{X})\right]\right\} \\
& - \theta_2.
\end{aligned}
$$

Then, reorganizing the order of terms and canceling, we arrive at the final form of a candidate EIF for $\theta_2$:

$$
\begin{aligned}
\Phi(\theta_2) = & S(1 + \gamma)\left\{\lambda_1(\mathbf{X}) + \alpha\lambda_1(\mathbf{X})\lambda_2(\mathbf{X})\left[v(\mathbf{X}) - w(\mathbf{X})\right]\right\} \\
& \qquad \times \left\{\frac{\mathbb{I}_t(T)}{e_t(\mathbf{X}, 1)g_1(\mathbf{X})}\left[Y - \mu(\mathbf{X}, 1, t)\right]g_0(\mathbf{X})\right\} \\
& + [1 - S]\left\{\lambda_1(\mathbf{X})v(\mathbf{X}) + \lambda_2(\mathbf{X})w(\mathbf{X})\right\} \\
& + [1 - S]\left\{\lambda_2(\mathbf{X}) + \alpha\lambda_1(\mathbf{X})\lambda_2(\mathbf{X})\left[w(\mathbf{X}) - v(\mathbf{X})\right]\right\} \\
& \qquad \times \left\{\frac{\mathbb{I}_t(T)}{e_t(\mathbf{X}, 0)}\left[Y - \mu(\mathbf{X}, 0, t)\right]\left[1 + \rho e_{1-t}(\mathbf{X}, 0)\right]\right. \\
& \qquad \qquad \left. + \rho\mu(\mathbf{X}, 0, t)\left[\mathbb{I}_{1-t}(T) - e_{1-t}(\mathbf{X}, 0)\right]\right\} \\
& - \theta_2.
\end{aligned}
$$

As we did for $\phi_{\theta_1}$, we drop the arguments and simply denote the EIF for $\theta_2$ as $\phi_{\theta_2}$ for simplicity.

### C.2.3  Checking Candidate EIF for $\theta_1$

We start by rewriting

$$
\begin{aligned}
\theta_1 &= \int_{\mathbf{x}} g_1(\mathbf{x})\mu(\mathbf{x}, 1, t)p(\mathbf{x})dx \\
&= \int_{\mathbf{x}}\int_y yp(y|\mathbf{x}, 1, t)dy\ p(1|\mathbf{x})p(\mathbf{x})dx,
\end{aligned}
$$

by replacing $\mu(\mathbf{x}, 1, t)$ with $\int_y yp(y|\mathbf{x}, 1, t)dy$ and $g_1(\mathbf{x})$ with $p(1|\mathbf{x})$.

Recall that we need to show that $\nabla_h\theta_1 = \mathbb{E}_{\mathcal{P}}[\phi_{\theta_1}h]$ for any generic score, $h$. To compute the directional derivative for $\theta_1$ and $h$, $\nabla_h\theta_1$, we introduce the notation $\tilde{p}_\epsilon = (1 + \epsilon h)p$. Then, for any generic $h$ we have that

$$
\nabla_h\theta_1 = \frac{\partial}{\partial\epsilon}\int_{\mathbf{x}}\int_y y\tilde{p}_\epsilon(y|\mathbf{x}, 1, t)dy\ \tilde{p}_\epsilon(1|\mathbf{x})\tilde{p}_\epsilon(\mathbf{x})dx\bigg|_{\epsilon=0}.
$$

For the expectation $\mathbb{E}_{\mathcal{P}}[\phi_{\theta_1}h]$ we forego denoting the $\mathcal{P}$ for the remainder of this section for brevity. Then, we take the following steps to rewrite this expectation.

$$\mathbb{E}[\phi_{\theta_1} h] = \mathbb{E}[(\phi_{\theta_1} + \theta_1)h] - \mathbb{E}[\theta_1 h]$$

$$= \int_{\mathbf{x}} \sum_s \sum_{t'} \int_y \left[ \frac{S\mathbb{I}(t' = t)}{e_t(\mathbf{x}, 1)} \left[ y - \mu(\mathbf{x}, 1, t) \right] + S\mu(\mathbf{x}, 1, t) \right]$$
$$\times hp(y|\mathbf{x}, s, t')dy \, p(t'|\mathbf{x}, s)p(s|\mathbf{x})p(\mathbf{x})dx$$

$$= \int_{\mathbf{x}} \int_y [y - \mu(\mathbf{x}, 1, t)] \, hp(y|\mathbf{x}, 1, t)dy \, p(1|\mathbf{x})p(\mathbf{x})dx$$

$$+ \int_{\mathbf{x}} \mu(\mathbf{x}, 1, t) \sum_{t'} \int_y hp(y|\mathbf{x}, 1, t')dy \, p(t'|\mathbf{x}, 1)p(1|\mathbf{x})p(\mathbf{x})dx.$$

In the above, we leveraged the fact that $\mathbb{E}[\theta_1 h] = 0$ for any valid score function $h$.

It is easier to prove this equality by decomposing $h = h_{Y|\mathbf{X},S,T} + h_{T|\mathbf{X},S} + h_{S|\mathbf{X}} + h_{\mathbf{X}}$ and showing that

$$\nabla_{h_{Y|\mathbf{X},S,T}} \theta_1 + \nabla_{h_{T|\mathbf{X},S}} \theta_1 + \nabla_{h_{S|\mathbf{X}}} \theta_1 + \nabla_{h_{\mathbf{X}}} \theta_1 =$$
$$\mathbb{E}[\phi_{\theta_1} h_{Y|\mathbf{X},S,T}] + \mathbb{E}[\phi_{\theta_1} h_{T|\mathbf{X},S}] + \mathbb{E}[\phi_{\theta_1} h_{S|\mathbf{X}}] + \mathbb{E}[\phi_{\theta_1} h_{\mathbf{X}}].$$

In particular, we will show that

$$\nabla_{h_{\mathbf{X}}} \theta_1 = \mathbb{E}[\phi_{\theta_1} h_{\mathbf{X}}],$$
$$\nabla_{h_{S|\mathbf{X}}} \theta_1 = \mathbb{E}[\phi_{\theta_1} h_{S|\mathbf{X}}],$$
$$\nabla_{h_{T|\mathbf{X},S}} \theta_1 = \mathbb{E}[\phi_{\theta_1} h_{T|\mathbf{X},S}],$$
$$\nabla_{h_{Y|\mathbf{X},S,T}} \theta_1 = \mathbb{E}[\phi_{\theta_1} h_{Y|\mathbf{X},S,T}].$$

We start with $\nabla_{h_{\mathbf{X}}} \theta_1 = \mathbb{E}[\phi_{\theta_1} h_{\mathbf{X}}]$. For $\nabla_{h_{\mathbf{X}}} \theta_1$ we replace the corresponding term in the factorized distribution function, $\tilde{p}_\epsilon(\mathbf{x}) = (1 + \epsilon h_{\mathbf{X}}(\mathbf{x}))p(\mathbf{x})$, and set the other conditional probability density functions to their normal $p$ form.

$$\nabla_{h_{\mathbf{X}}} \theta_1 = \frac{\partial}{\partial \epsilon} \int_{\mathbf{x}} \int_y yp(y|\mathbf{x}, 1, t)dy \, p(1|\mathbf{x})(1 + \epsilon h_{\mathbf{X}}(\mathbf{x}))p(\mathbf{x})dx \bigg|_{\epsilon=0}$$

$$= \int_{\mathbf{x}} \int_y yp(y|\mathbf{x}, 1, t)dy \, p(1|\mathbf{x})h_{\mathbf{X}}(\mathbf{x})p(\mathbf{x})dx$$

$$= \int_{\mathbf{x}} \mu(\mathbf{x}, 1, t) \, p(1|\mathbf{x})h_{\mathbf{X}}(\mathbf{x})p(\mathbf{x})dx$$

$$= \mathbb{E}[\mu(\mathbf{X}, 1, t)g_1(\mathbf{X})h_{\mathbf{X}}(\mathbf{X})].$$

Then we are left to show that $\mathbb{E}[\phi_{\theta_1} h_{\mathbf{X}}] = \mathbb{E}[\mu(\mathbf{X}, 1, t)g_1(\mathbf{X})h_{\mathbf{X}}(\mathbf{X})]$:

$$\mathbb{E}[\phi_{\theta_1} h_{\mathbf{X}}] = \int_{\mathbf{x}} \int_y [y - \mu(\mathbf{x}, 1, t)] \, h_{\mathbf{X}}(\mathbf{x})p(y|\mathbf{x}, 1, t)dy \, p(1|\mathbf{x})p(\mathbf{x})dx$$

$$+ \int_{\mathbf{x}} \mu(\mathbf{x}, 1, t) \sum_{t'} \int_y h_{\mathbf{X}}(\mathbf{x})p(y|\mathbf{x}, 1, t')dy \, p(t'|\mathbf{x}, 1)p(1|\mathbf{x})p(\mathbf{x})dx$$

$$= \int_{\mathbf{x}} h_{\mathbf{X}}(\mathbf{x}) \left( \int_y [y - \mu(\mathbf{x}, 1, t)] \, p(y|\mathbf{x}, 1, t)dy \right) p(1|\mathbf{x})p(\mathbf{x})dx$$

$$+ \int_{\mathbf{x}} h_{\mathbf{X}}(\mathbf{x})\mu(\mathbf{x}, 1, t) \left( \sum_{t'} \int_y p(y|\mathbf{x}, 1, t')dy \, p(t'|\mathbf{x}, 1) \right) p(1|\mathbf{x})p(\mathbf{x})dx$$

$$= 0 + \int_{\mathbf{x}} h_{\mathbf{X}}(\mathbf{x})\mu(\mathbf{x}, 1, t)(1)p(1|\mathbf{x})p(\mathbf{x})dx$$

$$= \mathbb{E}[\mu(\mathbf{X}, 1, t)g_1(\mathbf{X})h_{\mathbf{X}}(\mathbf{X})]$$

Similarly, for $\nabla_{h_{S|\mathbf{x}}}\theta_1 = \mathbb{E}[\phi_{\theta_1}h_{S|\mathbf{x}}]$, we first simplify $\nabla_{h_{S|\mathbf{x}}}\theta_1$ by replacing the corresponding term in the factorized distribution function, $\tilde{p}_\epsilon(1|\mathbf{x}) = (1 + \epsilon h_{S|\mathbf{x}}(1|\mathbf{x}))p(1|\mathbf{x})$, and set the other conditional probability density functions to their normal $p$ form.

$$
\begin{aligned}
\nabla_{h_{S|\mathbf{x}}}\theta_1 &= \frac{\partial}{\partial\epsilon}\int_{\mathbf{x}}\int_y yp(y|\mathbf{x},1,t)dy \ (1 + \epsilon h_{S|\mathbf{x}}(1|\mathbf{x}))p(1|\mathbf{x})p(\mathbf{x})dx\bigg|_{\epsilon=0}\\
&= \int_{\mathbf{x}}\int_y yp(y|\mathbf{x},1,t)dy \ h_{S|\mathbf{x}}(1|\mathbf{x})p(1|\mathbf{x})p(\mathbf{x})dx\\
&= \int_{\mathbf{x}}\mu(\mathbf{x},1,t)h_{S|\mathbf{x}}(1|\mathbf{x})p(1|\mathbf{x})p(\mathbf{x})dx\\
&= \mathbb{E}[\mu(\mathbf{X},1,t)g_1(\mathbf{X})h_{S|\mathbf{x}}(1|\mathbf{X})].
\end{aligned}
$$

We then show that $\mathbb{E}[\phi_{\theta_1}h_{S|\mathbf{X}}] = \mathbb{E}[\mu(\mathbf{X},1,t)g_1(\mathbf{X})h_{S|\mathbf{X}}(1|\mathbf{X})]$:

$$
\begin{aligned}
\mathbb{E}[\phi_{\theta_1}h_{S|\mathbf{X}}] &= \int_{\mathbf{x}}\int_y [y - \mu(\mathbf{x},1,t)] \, h_{S|\mathbf{X}}(s|\mathbf{x})p(y|\mathbf{x},1,t)dy \ p(1|\mathbf{x})p(\mathbf{x})dx\\
&\quad + \int_{\mathbf{x}}\mu(\mathbf{x},1,t)\sum_{t'}\int_y h_{S|\mathbf{X}}(s|\mathbf{x})p(y|\mathbf{x},1,t')dy \ p(t'|\mathbf{x},1)p(1|\mathbf{x})p(\mathbf{x})dx\\
&= \int_{\mathbf{x}}h_{S|\mathbf{X}}(s|\mathbf{x})\left(\int_y [y - \mu(\mathbf{x},1,t)]\,p(y|\mathbf{x},1,t)dy\right)p(1|\mathbf{x})p(\mathbf{x})dx\\
&\quad + \int_{\mathbf{x}}h_{S|\mathbf{X}}(s|\mathbf{x})\mu(\mathbf{x},1,t)\left(\sum_{t'}\int_y p(y|\mathbf{x},1,t')dy \ p(t'|\mathbf{x},1)\right)p(1|\mathbf{x})p(\mathbf{x})dx\\
&= 0 + \int_{\mathbf{x}}h_{S|\mathbf{X}}(s|\mathbf{x})\mu(\mathbf{x},1,t)\big(1\big)p(1|\mathbf{x})p(\mathbf{x})dx\\
&= \int_{\mathbf{x}}h_{S|\mathbf{X}}(1|\mathbf{x})\mu(\mathbf{x},1,t)p(1|\mathbf{x})p(\mathbf{x})dx\\
&= \mathbb{E}[\mu(\mathbf{X},1,t)g_1(\mathbf{X})h_{S|\mathbf{X}}(1|\mathbf{X})]
\end{aligned}
$$

Moving on to $\nabla_{h_{T|\mathbf{X},S}}\theta_1 = \mathbb{E}[\phi_{\theta_1}h_{T|\mathbf{X},S}]$, we start by simplifying $\nabla_{h_{T|\mathbf{X},S}}\theta_1$ by replacing the corresponding term in the factorized distribution function and setting the other conditional probability density functions to their normal $p$ form. We note here that there is no $\tilde{p}_\epsilon(t|\mathbf{x},s)$ in the integral, so this term simplifies to zero.

$$
\begin{aligned}
\nabla_{h_{T|\mathbf{X},S}}\theta_1 &= \frac{\partial}{\partial\epsilon}\int_{\mathbf{x}}\int_y yp(y|\mathbf{x},1,t)dy \ p(1|\mathbf{x})p(\mathbf{x})dx\bigg|_{\epsilon=0}\\
&= 0.
\end{aligned}
$$

We show too that $\mathbb{E}[\phi_{\theta_1}h_{T|\mathbf{X},S}]$ is equal to zero.

$$\mathbb{E}[\phi_{\theta_1} h_{T|\mathbf{X},S}] = \int_{\mathbf{x}} \int_y [y - \mu(\mathbf{x},1,t)]\, h_{T|\mathbf{X},S}(t|\mathbf{x},s)p(y|\mathbf{x},1,t)dy\, p(1|\mathbf{x})p(\mathbf{x})dx$$

$$+ \int_{\mathbf{x}} \mu(\mathbf{x},1,t) \sum_{t'} \int_y h_{T|\mathbf{X},S}(t'|\mathbf{x},s)p(y|\mathbf{x},1,t')dy\, p(t'|\mathbf{x},1)p(1|\mathbf{x})p(\mathbf{x})dx$$

$$= \int_{\mathbf{x}} h_{T|\mathbf{X},S}(t|\mathbf{x},s)\left( \int_y [y - \mu(\mathbf{x},1,t)]\, p(y|\mathbf{x},1,t)dy \right)p(1|\mathbf{x})p(\mathbf{x})dx$$

$$+ \int_{\mathbf{x}} \mu(\mathbf{x},1,t) \sum_{t'} h_{T|\mathbf{X},S}(t'|\mathbf{x},s)\left( \int_y p(y|\mathbf{x},1,t')dy \right)p(t'|\mathbf{x},1)p(1|\mathbf{x})p(\mathbf{x})dx$$

$$= 0 + \int_{\mathbf{x}} \mu(\mathbf{x},1,t)\left( \sum_{t'} h_{T|\mathbf{X},S}(t'|\mathbf{x},s)p(t'|\mathbf{x},1) \right)p(1|\mathbf{x})p(\mathbf{x})dx$$

$$= 0 + \int_{\mathbf{x}} \mu(\mathbf{x},1,t)\big(0\big)p(1|\mathbf{x})p(\mathbf{x})dx$$

$$= 0.$$

Similarly, for $\nabla_{h_{Y|\mathbf{X},S,T}}\theta_1 = \mathbb{E}[\phi_{\theta_1} h_{Y|\mathbf{X},S,T}]$, we simplify $\nabla_{h_{Y|\mathbf{X},S,T}}\theta_1$ by replacing the corresponding term in the factorized distribution function, $\tilde{p}_\epsilon(y|\mathbf{x},1,t) = (1 + \epsilon h_{Y|\mathbf{X},S,T}(y|\mathbf{x},1,t))p(y|\mathbf{x},1,t)$, and set the other conditional probability density functions to their normal $p$ form.

$$\nabla_{h_{Y|\mathbf{X},S,T}}\theta_1 = \frac{\partial}{\partial \epsilon} \int_{\mathbf{x}} \int_y y(1 + \epsilon h_{Y|\mathbf{X},S,T}(y|\mathbf{x},1,t))p(y|\mathbf{x},1,t)dy\, p(1|\mathbf{x})p(\mathbf{x})dx \Big|_{\epsilon=0}$$

$$= \int_{\mathbf{x}} \int_y y h_{Y|\mathbf{X},S,T}(y|\mathbf{x},1,t)p(y|\mathbf{x},1,t)dy\, p(1|\mathbf{x})p(\mathbf{x})dx$$

$$= \mathbb{E}[g_1(\mathbf{X})\mathbb{E}[Y h_{Y|\mathbf{X},S,T}(Y|\mathbf{X},1,t)|\mathbf{X},S=1,T=t]]$$

We finish validating the EIF for $\theta_1$ by showing that
$\mathbb{E}[\phi_{\theta_1} h_{Y|\mathbf{X},S,T}] = \mathbb{E}[g_1(\mathbf{X})\mathbb{E}[Y h_{Y|\mathbf{X},S,T}(Y|\mathbf{X},1,t)|\mathbf{X},S=1,T=t]]$.

$$\mathbb{E}[\phi_{\theta_1} h_{Y|\mathbf{X},S,T}] = \int_{\mathbf{x}} \int_y [y - \mu(\mathbf{x},1,t)]\, h_{Y|\mathbf{X},S,T}(y|\mathbf{x},s,t)p(y|\mathbf{x},1,t)dy\, p(1|\mathbf{x})p(\mathbf{x})dx$$

$$+ \int_{\mathbf{x}} \mu(\mathbf{x},1,t) \sum_{t'} \int_y h_{Y|\mathbf{X},S,T}(y|\mathbf{x},s,t')p(y|\mathbf{x},1,t')dy p(t'|\mathbf{x},1)p(1|\mathbf{x})p(\mathbf{x})dx$$

$$= \mathbb{E}\big[\mathbb{I}(S=1)\mathbb{E}[(Y - \mu(\mathbf{X},1,t))h_{Y|\mathbf{X},S,T}(Y|\mathbf{X},1,t)|\mathbf{X},S=1,T=t]\big]$$

$$+ \mathbb{E}\big[\mu(\mathbf{X},1,t)\mathbb{I}(S=1)\mathbb{E}[h_{Y|\mathbf{X},S,T}(Y|\mathbf{X},1,T)|\mathbf{X},S=1]\big]$$

$$= \mathbb{E}\big[\mathbb{I}(S=1)\mathbb{E}[Y h_{Y|\mathbf{X},S,T}(Y|\mathbf{X},1,t)|\mathbf{X},S=1,T=t]\big]$$

$$- \mathbb{E}\big[\mathbb{I}(S=1)\mu(\mathbf{X},1,t)\mathbb{E}[h_{Y|\mathbf{X},S,T}(Y|\mathbf{X},1,t)|\mathbf{X},S=1,T=t]\big]$$

$$+ \mathbb{E}\big[\mu(\mathbf{X},1,t)\mathbb{I}(S=1)\mathbb{E}[h_{Y|\mathbf{X},S,T}(Y|\mathbf{X},1,T)|\mathbf{X},S=1]\big]$$

$$= \mathbb{E}\big[\mathbb{I}(S=1)\mathbb{E}[Y h_{Y|\mathbf{X},S,T}(Y|\mathbf{X},1,t)|\mathbf{X},S=1,T=t]\big]$$

$$- \mathbb{E}\big[\mathbb{I}(S=1)\mu(\mathbf{X},1,t) \times 0\big]$$

$$+ \mathbb{E}\big[\mu(\mathbf{X},1,t)\mathbb{I}(S=1) \times 0\big]$$

$$= \mathbb{E}\big[g_1(\mathbf{X})\mathbb{E}[Y h_{Y|\mathbf{X},S,T}(Y|\mathbf{X},1,t)|\mathbf{X},S=1,T=t]\big]$$

### C.2.4 Checking Candidate EIF for $\theta_2$

We perform the same steps to check the candidate EIF for $\theta_2$. We first write $\theta_2$ as shown below:

$$\theta_2 = \int_{\mathbf{x}} g_0(\mathbf{x}) \left\{ \lambda_1(\mathbf{x})v(\mathbf{x}) + \lambda_2(\mathbf{x})w(\mathbf{x}) \right\} p(x)dx$$

$$= \int_{\mathbf{x}} \left[ \frac{\exp\left( \alpha(1+\gamma) \int_y yp(y|\mathbf{x},1,t)dy \right)}{\exp\left( \alpha(1+\gamma) \int_y yp(y|\mathbf{x},1,t)dy \right) + \exp\left( \alpha \int_y yp(y|\mathbf{x},0,t)dy[1 + \rho p(1-t|\mathbf{x},0)] \right)} \right.$$

$$\times (1+\gamma) \int_y yp(y|\mathbf{x},1,t)dy \Bigg] p(0|\mathbf{x})p(x)dx$$

$$+ \int_{\mathbf{x}} \left[ \frac{\exp\left( \alpha \int_y yp(y|\mathbf{x},0,t)dy[1 + \rho p(1-t|\mathbf{x},0)] \right)}{\exp\left( \alpha(1+\gamma) \int_y yp(y|\mathbf{x},1,t)dy \right) + \exp\left( \alpha \int_y yp(y|\mathbf{x},0,t)dy[1 + \rho p(1-t|\mathbf{x},0)] \right)} \right.$$

$$\times \int_y yp(y|\mathbf{x},0,t)dy[1 + \rho p(1-t|\mathbf{x},0)] \Bigg] p(0|\mathbf{x})p(x)dx,$$

where we replace $g_0(\mathbf{x})$ with $p(0|\mathbf{x})$, $\mu(\mathbf{x},1,t)$ with $\int_y yp(y|\mathbf{x},1,t)dy$, $\mu(\mathbf{x},0,t)$ with $\int_y yp(y|\mathbf{x},0,t)dy$, $e_t(\mathbf{x},0)$ with $p(t|\mathbf{x},0)$ and $e_{1-t}(\mathbf{x},0)$ with $p(1-t|\mathbf{x},0)$.

We then have that for any generic $h$ that $\nabla_h \theta_2$ can be written as

$$\nabla_h \theta_2 =$$

$$\frac{\partial}{\partial \epsilon} \int_{\mathbf{x}} \left[ \frac{\exp\left( \alpha(1+\gamma) \int_y y\tilde{p}_\epsilon(y|\mathbf{x},1,t)dy \right)}{\exp\left( \alpha(1+\gamma) \int_y y\tilde{p}_\epsilon(y|\mathbf{x},1,t)dy \right) + \exp\left( \alpha \int_y y\tilde{p}_\epsilon(y|\mathbf{x},0,t)dy[1 + \rho\tilde{p}_\epsilon(1-t|\mathbf{x},0)] \right)} \right.$$

$$\times (1+\gamma) \int_y y\tilde{p}_\epsilon(y|\mathbf{x},1,t)dy \Bigg] \tilde{p}_\epsilon(0|\mathbf{x})\tilde{p}_\epsilon(x)dx \Bigg|_{\epsilon=0}$$

$$+ \frac{\partial}{\partial \epsilon} \int_{\mathbf{x}} \left[ \frac{\exp\left( \alpha \int_y y\tilde{p}_\epsilon(y|\mathbf{x},0,t)dy[1 + \rho\tilde{p}_\epsilon(1-t|\mathbf{x},0)] \right)}{\exp\left( \alpha(1+\gamma) \int_y y\tilde{p}_\epsilon(y|\mathbf{x},1,t)dy \right) + \exp\left( \alpha \int_y y\tilde{p}_\epsilon(y|\mathbf{x},0,t)dy[1 + \rho\tilde{p}_\epsilon(1-t|\mathbf{x},0)] \right)} \right.$$

$$\times \int_y y\tilde{p}_\epsilon(y|\mathbf{x},0,t)dy[1 + \rho\tilde{p}_\epsilon(1-t|\mathbf{x},0)] \Bigg] \tilde{p}_\epsilon(0|\mathbf{x})\tilde{p}_\epsilon(x)dx \Bigg|_{\epsilon=0}$$

and $\mathbb{E}[\phi_{\theta_2} h]$ can be written as

$$\mathbb{E}[\phi_{\theta_2} h] = \mathbb{E}[(\phi_{\theta_2} + \theta_2)h] - \mathbb{E}[\theta_2 h]$$

$$= \int_{\mathbf{x}} \sum_s \sum_{t'} \int_y \Bigg\{$$

$$s(1+\gamma) \left\{ \lambda_1(\mathbf{x}) + \alpha\lambda_1(\mathbf{x})\lambda_2(\mathbf{x}) \left[ v(\mathbf{x}) - w(\mathbf{x}) \right] \right\}$$

$$\times \left\{ \frac{\mathbb{I}_t(t')}{e_t(\mathbf{x},1)g_1(\mathbf{x})} \left[ y - \mu(\mathbf{x},1,t) \right] g_0(\mathbf{x}) \right\}$$

$$+ [1-s] \left\{ \lambda_1(\mathbf{x})v(\mathbf{x}) + \lambda_2(\mathbf{x})w(\mathbf{x}) \right\}$$

$$+ [1-s] \left\{ \lambda_2(\mathbf{x}) + \alpha\lambda_1(\mathbf{x})\lambda_2(\mathbf{x}) \left[ w(\mathbf{x}) - v(\mathbf{x}) \right] \right\}$$

$$\times \left\{ \frac{\mathbb{I}_t(t')}{e_t(\mathbf{x},0)} \left[ y - \mu(\mathbf{x},0,t) \right] \left[ 1 + \rho e_{1-t}(\mathbf{x},0) \right] + \rho\mu(\mathbf{x},0,t) \left[ \mathbb{I}_{1-t}(t') - e_{1-t}(\mathbf{x},0) \right] \right\}$$

$$\Bigg\} \times hp(y|\mathbf{x},s,t')p(t'|\mathbf{x},s)p(s|\mathbf{x})p(\mathbf{x})dydx.$$

We can distribute terms to write this as

$$\mathbb{E}[\phi_{\theta_2}h] = \int_{\mathbf{x}}\int_y (1+\gamma)\left\{\lambda_1(\mathbf{x}) + \alpha\lambda_1(\mathbf{x})\lambda_2(\mathbf{x})\left[v(\mathbf{x}) - w(\mathbf{x})\right]\right\}$$
$$\times \left[y - \mu(\mathbf{x}, 1, t)\right]hp(y|\mathbf{x}, 1, t)p(0|\mathbf{x})p(\mathbf{x})dydx$$
$$+ \int_{\mathbf{x}}\sum_{t'}\int_y \left\{\lambda_1(\mathbf{x})v(\mathbf{x}) + \lambda_2(\mathbf{x})w(\mathbf{x})\right\}hp(y|\mathbf{x}, 0, t')p(t'|\mathbf{x}, 0)p(0|\mathbf{x})p(\mathbf{x})dydx$$
$$+ \int_{\mathbf{x}}\int_y \left\{\lambda_2(\mathbf{x}) + \alpha\lambda_1(\mathbf{x})\lambda_2(\mathbf{x})\left[w(\mathbf{x}) - v(\mathbf{x})\right]\right\}\left[y - \mu(\mathbf{x}, 0, t)\right]\left[1 + \rho e_{1-t}(\mathbf{x}, 0)\right]$$
$$\times hp(y|\mathbf{x}, 0, t)p(0|\mathbf{x})p(\mathbf{x})dydx$$
$$+ \int_{\mathbf{x}}\sum_{t'}\int_y \left\{\lambda_2(\mathbf{x}) + \alpha\lambda_1(\mathbf{x})\lambda_2(\mathbf{x})\left[w(\mathbf{x}) - v(\mathbf{x})\right]\right\}\rho\mu(\mathbf{x}, 0, t)\left[\mathbb{I}_{1-t}(t') - e_{1-t}(\mathbf{x}, 0)\right]$$
$$\times hp(y|\mathbf{x}, 0, t')p(t'|\mathbf{x}, 0)p(0|\mathbf{x})p(\mathbf{x})dydx.$$

And then write in expectation notation as

$$\mathbb{E}[\phi_{\theta_2}h] = (1+\gamma)\mathbb{E}\Big[g_0(\mathbf{X})\left\{\lambda_1(\mathbf{X}) + \alpha\lambda_1(\mathbf{X})\lambda_2(\mathbf{X})\left[v(\mathbf{X}) - w(\mathbf{X})\right]\right\}$$
$$\times \mathbb{E}[(Y - \mu(\mathbf{X}, 1, t))h|\mathbf{X}, S=1, T=t]\Big]$$
$$+ \mathbb{E}\Big[g_0(\mathbf{X})\left\{\lambda_1(\mathbf{X})v(\mathbf{X}) + \lambda_2(\mathbf{X})w(\mathbf{X})\right\}\mathbb{E}[h|\mathbf{X}, S=0]\Big]$$
$$+ \mathbb{E}\Big[g_0(\mathbf{X})\left[1 + \rho e_{1-t}(\mathbf{X}, 0)\right]\left\{\lambda_2(\mathbf{X}) + \alpha\lambda_1(\mathbf{X})\lambda_2(\mathbf{X})\left[w(\mathbf{X}) - v(\mathbf{X})\right]\right\}$$
$$\times \mathbb{E}[(Y - \mu(\mathbf{X}, 0, t))h|\mathbf{X}, S=0, T=t]\Big]$$
$$+ \mathbb{E}\Big[g_0(\mathbf{X})\rho\mu(\mathbf{X}, 0, t)\left\{\lambda_2(\mathbf{X}) + \alpha\lambda_1(\mathbf{X})\lambda_2(\mathbf{X})\left[w(\mathbf{X}) - v(\mathbf{X})\right]\right\}$$
$$\times \mathbb{E}[[\mathbb{I}_{1-t}(T) - e_{1-t}(\mathbf{X}, 0)]h|\mathbf{X}, S=0]\Big].$$

We now show equality of the separate decomposed components of the generic score. In particular, just like we did for $\theta_1$, we will show that

$$\nabla_{h_{\mathbf{X}}}\theta_2 = \mathbb{E}[\phi_{\theta_2}h_{\mathbf{X}}],$$
$$\nabla_{h_{S|\mathbf{X}}}\theta_2 = \mathbb{E}[\phi_{\theta_2}h_{S|\mathbf{X}}],$$
$$\nabla_{h_{T|\mathbf{X},S}}\theta_2 = \mathbb{E}[\phi_{\theta_2}h_{T|\mathbf{X},S}],$$
$$\nabla_{h_{Y|\mathbf{X},S,T}}\theta_2 = \mathbb{E}[\phi_{\theta_2}h_{Y|\mathbf{X},S,T}].$$

Starting with $\nabla_{h_{\mathbf{X}}}\theta_2 = \mathbb{E}[\phi_{\theta_2}h_{\mathbf{X}}]$, we first simplify $\nabla_{h_{\mathbf{X}}}\theta_2$ by replacing the corresponding term in the factorized distribution function, $\tilde{p}_\epsilon(\mathbf{x}) = (1 + \epsilon h_{\mathbf{X}}(\mathbf{x}))p(\mathbf{x})$, and set the other conditional probability density functions to their normal $p$ form.

$$\nabla_{h_\mathbf{x}} \theta_2 = \frac{\partial}{\partial \epsilon} \int_\mathbf{x} \left[ \frac{\exp\left(\alpha(1+\gamma)\int_y yp(y|\mathbf{x},1,t)dy\right)}{\exp\left(\alpha(1+\gamma)\int_y yp(y|\mathbf{x},1,t)dy\right) + \exp\left(\alpha\int_y yp(y|\mathbf{x},0,t)dy[1+\rho p(1-t|\mathbf{x},0)]\right)} \right.$$

$$\left. \times (1+\gamma)\int_y yp(y|\mathbf{x},1,t)dy \right] p(0|\mathbf{x})(1+\epsilon h_\mathbf{X}(\mathbf{x}))p(x)dx \Bigg|_{\epsilon=0}$$

$$+ \frac{\partial}{\partial \epsilon} \int_\mathbf{x} \left[ \frac{\exp\left(\alpha\int_y yp(y|\mathbf{x},0,t)dy[1+\rho p(1-t|\mathbf{x},0)]\right)}{\exp\left(\alpha(1+\gamma)\int_y yp(y|\mathbf{x},1,t)dy\right) + \exp\left(\alpha\int_y yp(y|\mathbf{x},0,t)dy[1+\rho p(1-t|\mathbf{x},0)]\right)} \right.$$

$$\left. \times \int_y yp(y|\mathbf{x},0,t)dy[1+\rho p(1-t|\mathbf{x},0)] \right] p(0|\mathbf{x})(1+\epsilon h_\mathbf{X}(\mathbf{x}))p(x)dx \Bigg|_{\epsilon=0}$$

$$= \int_\mathbf{x} \left\{ \lambda_1(\mathbf{X})v(\mathbf{X}) + \lambda_2(\mathbf{X})w(\mathbf{X}) \right\} p(0|\mathbf{x})h_\mathbf{X}(\mathbf{x})p(x)dx$$

$$= \mathbb{E}\left[ g_0(\mathbf{X})\left\{ \lambda_1(\mathbf{X})v(\mathbf{X}) + \lambda_2(\mathbf{X})w(\mathbf{X}) \right\} h_\mathbf{X}(\mathbf{X}) \right].$$

We now show that $\mathbb{E}[\phi_{\theta_2} h_\mathbf{X}] = \mathbb{E}\left[ g_0(\mathbf{X})\left\{ \lambda_1(\mathbf{X})v(\mathbf{X}) + \lambda_2(\mathbf{X})w(\mathbf{X}) \right\} h_\mathbf{X}(\mathbf{X}) \right]$:

$$\mathbb{E}[\phi_{\theta_2} h_\mathbf{X}] = (1+\gamma)\mathbb{E}\Big[ g_0(\mathbf{X})\left\{ \lambda_1(\mathbf{X}) + \alpha\lambda_1(\mathbf{X})\lambda_2(\mathbf{X})\left[ v(\mathbf{X}) - w(\mathbf{X}) \right] \right\}$$

$$\times \mathbb{E}[(Y - \mu(\mathbf{X},1,t))h_\mathbf{X}(\mathbf{X})|\mathbf{X}, S=1, T=t] \Big]$$

$$+ \mathbb{E}\Big[ g_0(\mathbf{X})\left\{ \lambda_1(\mathbf{X})v(\mathbf{X}) + \lambda_2(\mathbf{X})w(\mathbf{X}) \right\} \mathbb{E}[h_\mathbf{X}(\mathbf{X})|\mathbf{X}, S=0] \Big]$$

$$+ \mathbb{E}\Big[ g_0(\mathbf{X})[1+\rho e_{1-t}(\mathbf{X},0)]\left\{ \lambda_2(\mathbf{X}) + \alpha\lambda_1(\mathbf{X})\lambda_2(\mathbf{X})\left[ w(\mathbf{X}) - v(\mathbf{X}) \right] \right\}$$

$$\times \mathbb{E}[(Y - \mu(\mathbf{X},0,t))h_\mathbf{X}(\mathbf{X})|\mathbf{X}, S=0, T=t] \Big]$$

$$+ \mathbb{E}\Big[ g_0(\mathbf{X})\rho\mu(\mathbf{X},0,t)\left\{ \lambda_2(\mathbf{X}) + \alpha\lambda_1(\mathbf{X})\lambda_2(\mathbf{X})\left[ w(\mathbf{X}) - v(\mathbf{X}) \right] \right\}$$

$$\times \mathbb{E}[[\mathbb{I}_{1-t}(T) - e_{1-t}(\mathbf{X},0)]h_\mathbf{X}(\mathbf{X})|\mathbf{X}, S=0] \Big].$$

We can pull the $h_\mathbf{X}$ outside of all of the inner expectations to write as

$$\mathbb{E}[\phi_{\theta_2} h_\mathbf{X}] = (1+\gamma)\mathbb{E}\Big[ g_0(\mathbf{X})\left\{ \lambda_1(\mathbf{X}) + \alpha\lambda_1(\mathbf{X})\lambda_2(\mathbf{X})\left[ v(\mathbf{X}) - w(\mathbf{X}) \right] \right\}$$

$$\times h_\mathbf{X}(\mathbf{X})\mathbb{E}[(Y - \mu(\mathbf{X},1,t))|\mathbf{X}, S=1, T=t] \Big]$$

$$+ \mathbb{E}\Big[ g_0(\mathbf{X})\left\{ \lambda_1(\mathbf{X})v(\mathbf{X}) + \lambda_2(\mathbf{X})w(\mathbf{X}) \right\} h_\mathbf{X}(\mathbf{X}) \Big]$$

$$+ \mathbb{E}\Big[ g_0(\mathbf{X})[1+\rho e_{1-t}(\mathbf{X},0)]\left\{ \lambda_2(\mathbf{X}) + \alpha\lambda_1(\mathbf{X})\lambda_2(\mathbf{X})\left[ w(\mathbf{X}) - v(\mathbf{X}) \right] \right\}$$

$$\times h_\mathbf{X}(\mathbf{X})\mathbb{E}[(Y - \mu(\mathbf{X},0,t))|\mathbf{X}, S=0, T=t] \Big]$$

$$+ \mathbb{E}\Big[ g_0(\mathbf{X})\rho\mu(\mathbf{X},0,t)\left\{ \lambda_2(\mathbf{X}) + \alpha\lambda_1(\mathbf{X})\lambda_2(\mathbf{X})\left[ w(\mathbf{X}) - v(\mathbf{X}) \right] \right\}$$

$$\times h_\mathbf{X}(\mathbf{X})\mathbb{E}[[\mathbb{I}_{1-t}(T) - e_{1-t}(\mathbf{X},0)]|\mathbf{X}, S=0] \Big].$$

And then, applying the inner conditional expectations and cancelling like terms, we get

$$\mathbb{E}[\phi_{\theta_2} h_{\mathbf{X}}] = (1+\gamma)\mathbb{E}\Big[g_0(\mathbf{X})\left\{\lambda_1(\mathbf{X}) + \alpha\lambda_1(\mathbf{X})\lambda_2(\mathbf{X})\left[v(\mathbf{X}) - w(\mathbf{X})\right]\right\}$$
$$\times\, h_{\mathbf{X}}(\mathbf{X})(\mu(\mathbf{X},1,t) - \mu(\mathbf{X},1,t))\Big]$$
$$+ \mathbb{E}\Big[g_0(\mathbf{X})\left\{\lambda_1(\mathbf{X})v(\mathbf{X}) + \lambda_2(\mathbf{X})w(\mathbf{X})\right\}h_{\mathbf{X}}(\mathbf{X})\Big]$$
$$+ \mathbb{E}\Big[g_0(\mathbf{X})\left[1 + \rho e_{1-t}(\mathbf{X},0)\right]\left\{\lambda_2(\mathbf{X}) + \alpha\lambda_1(\mathbf{X})\lambda_2(\mathbf{X})\left[w(\mathbf{X}) - v(\mathbf{X})\right]\right\}$$
$$\times\, h_{\mathbf{X}}(\mathbf{X})(\mu(\mathbf{X},0,t) - \mu(\mathbf{X},0,t))\Big]$$
$$+ \mathbb{E}\Big[g_0(\mathbf{X})\rho\mu(\mathbf{X},0,t)\left\{\lambda_2(\mathbf{X}) + \alpha\lambda_1(\mathbf{X})\lambda_2(\mathbf{X})\left[w(\mathbf{X}) - v(\mathbf{X})\right]\right\}$$
$$\times\, h_{\mathbf{X}}(\mathbf{X})\left[e_{1-t}(\mathbf{X},0) - e_{1-t}(\mathbf{X},0)\right]\Big]$$
$$= \mathbb{E}\Big[g_0(\mathbf{X})\left\{\lambda_1(\mathbf{X})v(\mathbf{X}) + \lambda_2(\mathbf{X})w(\mathbf{X})\right\}h_{\mathbf{X}}(\mathbf{X})\Big].$$

Next, we show $\nabla_{h_{S|\mathbf{x}}}\theta_2 = \mathbb{E}[\phi_{\theta_2} h_{S|\mathbf{X}}]$. We start this by simplifying $\nabla_{h_{S|\mathbf{x}}}\theta_2$ by replacing the corresponding term in the factorized distribution function, $\tilde{p}_\epsilon(0|\mathbf{x}) = (1 + \epsilon h_{S|\mathbf{X}}(0|\mathbf{x}))p(0|\mathbf{x})$, and set the other conditional probability density functions to their normal $p$ form.

$$\nabla_{h_{S|\mathbf{x}}}\theta_2 = \frac{\partial}{\partial\epsilon}\int_{\mathbf{x}}\left[\frac{\exp\left(\alpha(1+\gamma)\int_y yp(y|\mathbf{x},1,t)dy\right)}{\exp\left(\alpha(1+\gamma)\int_y yp(y|\mathbf{x},1,t)dy\right) + \exp\left(\alpha\int_y yp(y|\mathbf{x},0,t)dy[1 + \rho p(1-t|\mathbf{x},0)]\right)}\right.$$
$$\left.\times (1+\gamma)\int_y yp(y|\mathbf{x},1,t)dy\right](1 + \epsilon h_{S|\mathbf{X}}(0|\mathbf{x}))p(0|\mathbf{x})p(\mathbf{x})d\mathbf{x}\Bigg|_{\epsilon=0}$$
$$+ \frac{\partial}{\partial\epsilon}\int_{\mathbf{x}}\left[\frac{\exp\left(\alpha\int_y yp(y|\mathbf{x},0,t)dy[1 + \rho p(1-t|\mathbf{x},0)]\right)}{\exp\left(\alpha(1+\gamma)\int_y yp(y|\mathbf{x},1,t)dy\right) + \exp\left(\alpha\int_y yp(y|\mathbf{x},0,t)dy[1 + \rho p(1-t|\mathbf{x},0)]\right)}\right.$$
$$\left.\times \int_y yp(y|\mathbf{x},0,t)dy[1 + \rho p(1-t|\mathbf{x},0)]\right](1 + \epsilon h_{S|\mathbf{X}}(0|\mathbf{x}))p(0|\mathbf{x})p(\mathbf{x})d\mathbf{x}\Bigg|_{\epsilon=0}$$
$$= \int_{\mathbf{x}}\left\{\lambda_1(\mathbf{X})v(\mathbf{X}) + \lambda_2(\mathbf{X})w(\mathbf{X})\right\}h_{S|\mathbf{X}}(0|\mathbf{x})p(0|\mathbf{x})p(\mathbf{x})d\mathbf{x}$$
$$= \mathbb{E}\Big[g_0(\mathbf{X})\left\{\lambda_1(\mathbf{X})v(\mathbf{X}) + \lambda_2(\mathbf{X})w(\mathbf{X})\right\}h_{S|\mathbf{X}}(0|\mathbf{X})\Big].$$

We proceed to show that $\mathbb{E}[\phi_{\theta_2} h_{S|\mathbf{X}}] = \mathbb{E}\Big[g_0(\mathbf{X})\left\{\lambda_1(\mathbf{X})v(\mathbf{X}) + \lambda_2(\mathbf{X})w(\mathbf{X})\right\}h_{S|\mathbf{X}}(0|\mathbf{X})\Big]$. First, plugging in $h_{S|\mathbf{X}}$,

$$\mathbb{E}[\phi_{\theta_2} h_{S|\mathbf{X}}] = (1+\gamma)\mathbb{E}\Big[g_0(\mathbf{X})\left\{\lambda_1(\mathbf{X}) + \alpha\lambda_1(\mathbf{X})\lambda_2(\mathbf{X})\left[v(\mathbf{X}) - w(\mathbf{X})\right]\right\}$$
$$\times\, \mathbb{E}[(Y - \mu(\mathbf{X},1,t))h_{S|\mathbf{X}}(S|\mathbf{X})|\mathbf{X}, S=1, T=t]\Big]$$
$$+ \mathbb{E}\Big[g_0(\mathbf{X})\left\{\lambda_1(\mathbf{X})v(\mathbf{X}) + \lambda_2(\mathbf{X})w(\mathbf{X})\right\}\mathbb{E}[h_{S|\mathbf{X}}(S|\mathbf{X})|\mathbf{X}, S=0]\Big]$$
$$+ \mathbb{E}\Big[g_0(\mathbf{X})\left[1 + \rho e_{1-t}(\mathbf{X},0)\right]\left\{\lambda_2(\mathbf{X}) + \alpha\lambda_1(\mathbf{X})\lambda_2(\mathbf{X})\left[w(\mathbf{X}) - v(\mathbf{X})\right]\right\}$$
$$\times\, \mathbb{E}[(Y - \mu(\mathbf{X},0,t))h_{S|\mathbf{X}}(S|\mathbf{X})|\mathbf{X}, S=0, T=t]\Big]$$
$$+ \mathbb{E}\Big[g_0(\mathbf{X})\rho\mu(\mathbf{X},0,t)\left\{\lambda_2(\mathbf{X}) + \alpha\lambda_1(\mathbf{X})\lambda_2(\mathbf{X})\left[w(\mathbf{X}) - v(\mathbf{X})\right]\right\}$$
$$\times\, \mathbb{E}[[\mathbb{1}_{1-t}(T) - e_{1-t}(\mathbf{X},0)]h_{S|\mathbf{X}}(S|\mathbf{X})|\mathbf{X}, S=0]\Big].$$

Then, pulling $h_{S|\mathbf{X}}$ out of the inner expectation where possible

$$
\begin{aligned}
\mathbb{E}[\phi_{\theta_2} h_{S|\mathbf{X}}] =&(1+\gamma)\mathbb{E}\Big[g_0(\mathbf{X})\left\{\lambda_1(\mathbf{X})+\alpha\lambda_1(\mathbf{X})\lambda_2(\mathbf{X})\left[v(\mathbf{X})-w(\mathbf{X})\right]\right\} \\
&\qquad\times h_{S|\mathbf{X}}(1|\mathbf{X})\mathbb{E}[(Y-\mu(\mathbf{X},1,t))|\mathbf{X},S=1,T=t]\Big] \\
&+\mathbb{E}\Big[g_0(\mathbf{X})\left\{\lambda_1(\mathbf{X})v(\mathbf{X})+\lambda_2(\mathbf{X})w(\mathbf{X})\right\}h_{S|\mathbf{X}}(0|\mathbf{X})\Big] \\
&+\mathbb{E}\Big[g_0(\mathbf{X})\left[1+\rho e_{1-t}(\mathbf{X},0)\right]\left\{\lambda_2(\mathbf{X})+\alpha\lambda_1(\mathbf{X})\lambda_2(\mathbf{X})\left[w(\mathbf{X})-v(\mathbf{X})\right]\right\} \\
&\qquad\times h_{S|\mathbf{X}}(0|\mathbf{X})\mathbb{E}[(Y-\mu(\mathbf{X},0,t))|\mathbf{X},S=0,T=t]\Big] \\
&+\mathbb{E}\Big[g_0(\mathbf{X})\rho\mu(\mathbf{X},0,t)\left\{\lambda_2(\mathbf{X})+\alpha\lambda_1(\mathbf{X})\lambda_2(\mathbf{X})\left[w(\mathbf{X})-v(\mathbf{X})\right]\right\} \\
&\qquad\times h_{S|\mathbf{X}}(0|\mathbf{X})\mathbb{E}[[\mathbb{I}_{1-t}(T)-e_{1-t}(\mathbf{X},0)]|\mathbf{X},S=0]\Big].
\end{aligned}
$$

And then applying the inner expectations, cancelling terms, and simplifying,

$$
\begin{aligned}
\mathbb{E}[\phi_{\theta_2} h_{S|\mathbf{X}}] =&(1+\gamma)\mathbb{E}\Big[g_0(\mathbf{X})\left\{\lambda_1(\mathbf{X})+\alpha\lambda_1(\mathbf{X})\lambda_2(\mathbf{X})\left[v(\mathbf{X})-w(\mathbf{X})\right]\right\} \\
&\qquad\times h_{S|\mathbf{X}}(1|\mathbf{X})(\mu(\mathbf{X},1,t)-\mu(\mathbf{X},1,t))\Big] \\
&+\mathbb{E}\Big[g_0(\mathbf{X})\left\{\lambda_1(\mathbf{X})v(\mathbf{X})+\lambda_2(\mathbf{X})w(\mathbf{X})\right\}h_{S|\mathbf{X}}(0|\mathbf{X})\Big] \\
&+\mathbb{E}\Big[g_0(\mathbf{X})\left[1+\rho e_{1-t}(\mathbf{X},0)\right]\left\{\lambda_2(\mathbf{X})+\alpha\lambda_1(\mathbf{X})\lambda_2(\mathbf{X})\left[w(\mathbf{X})-v(\mathbf{X})\right]\right\} \\
&\qquad\times h_{S|\mathbf{X}}(0|\mathbf{X})(\mu(\mathbf{X},0,t)-\mu(\mathbf{X},0,t))\Big] \\
&+\mathbb{E}\Big[g_0(\mathbf{X})\rho\mu(\mathbf{X},0,t)\left\{\lambda_2(\mathbf{X})+\alpha\lambda_1(\mathbf{X})\lambda_2(\mathbf{X})\left[w(\mathbf{X})-v(\mathbf{X})\right]\right\} \\
&\qquad\times h_{S|\mathbf{X}}(0|\mathbf{X})\left[e_{1-t}(\mathbf{X},0)-e_{1-t}(\mathbf{X},0)\right]\Big] \\
=&\mathbb{E}\Big[g_0(\mathbf{X})\left\{\lambda_1(\mathbf{X})v(\mathbf{X})+\lambda_2(\mathbf{X})w(\mathbf{X})\right\}h_{S|\mathbf{X}}(0|\mathbf{X})\Big].
\end{aligned}
$$

Nearly there, we now show $\nabla_{h_{T|\mathbf{X},S}}\theta_2 = \mathbb{E}[\phi_{\theta_2} h_{T|\mathbf{X},S}]$. We simplify $\nabla_{h_{T|\mathbf{X},S}}\theta_2$ by replacing the corresponding term in the factorized distribution function, $\tilde{p}_\epsilon(1-t|\mathbf{x},0) = (1+\epsilon h_{T|\mathbf{X},S}(1-t|\mathbf{x},0))p(1-t|\mathbf{x},0)$, and set the other conditional probability density functions to their normal $p$ form. We have to format $\nabla_{h_{T|\mathbf{X},S}}\theta_2$ slightly differently to allow it to stay within the page margins.

$$\nabla_{h_{T|\mathbf{x},S}}\theta_2 = \frac{\partial}{\partial\epsilon}\int_{\mathbf{x}}\exp\left(\alpha(1+\gamma)\int_y yp(y|\mathbf{x},1,t)dy\right)$$

$$\times\left\{\exp\left(\alpha\int_y yp(y|\mathbf{x},0,t)dy[1+\rho(1+\epsilon h_{T|\mathbf{x},S}(1-t|\mathbf{x},0))p(1-t|\mathbf{x},0)]\right)\right.$$

$$\left.+\exp\left(\alpha(1+\gamma)\int_y yp(y|\mathbf{x},1,t)dy\right)\right\}^{-1}$$

$$\times\left.\left((1+\gamma)\int_y yp(y|\mathbf{x},1,t)dy\right)p(0|\mathbf{x})p(x)dx\right|_{\epsilon=0}$$

$$+\frac{\partial}{\partial\epsilon}\int_{\mathbf{x}}\exp\left(\alpha\int_y yp(y|\mathbf{x},0,t)dy[1+\rho(1+\epsilon h_{T|\mathbf{x},S}(1-t|\mathbf{x},0))p(1-t|\mathbf{x},0)]\right)$$

$$\times\left\{\exp\left(\alpha\int_y yp(y|\mathbf{x},0,t)dy[1+\rho(1+\epsilon h_{T|\mathbf{x},S}(1-t|\mathbf{x},0))p(1-t|\mathbf{x},0)]\right)\right.$$

$$\left.+\exp\left(\alpha(1+\gamma)\int_y yp(y|\mathbf{x},1,t)dy\right)\right\}^{-1}$$

$$\times\left(\int_y yp(y|\mathbf{x},0,t)dy[1+\rho(1+\epsilon h_{T|\mathbf{x},S}(1-t|\mathbf{x},0))p(1-t|\mathbf{x},0)]\right)$$

$$\times p(0|\mathbf{x})p(x)dx\Big|_{\epsilon=0}$$

To simplify, we plug in $\exp\left(\alpha v(\mathbf{x})\right)$ for $\exp\left(\alpha(1+\gamma)\int_y yp(y|\mathbf{x},1,t)dy\right)$ and $\mu(\mathbf{x},0,t)$ for $\int_y yp(y|\mathbf{x},0,t)dy$. We also replace $p(1-t|\mathbf{x},0)$ with $e_{1-t}(\mathbf{x},0)$ simply for clarity in the following steps. Then, $\nabla_{h_{T|\mathbf{x},S}}\theta_2$ is equal to

$$\frac{\partial}{\partial\epsilon}\int_{\mathbf{x}}\frac{\exp\left(\alpha v(\mathbf{x})\right)}{\exp\left(\alpha v(\mathbf{x})\right)+\exp\left(\alpha\mu(\mathbf{x},0,t)[1+\rho(1+\epsilon h_{T|\mathbf{x},S}(1-t|\mathbf{x},0))e_{1-t}(\mathbf{x},0)]\right)}$$
$$\times v(\mathbf{x})p(0|\mathbf{x})p(x)dx|_{\epsilon=0}$$
$$+\frac{\partial}{\partial\epsilon}\int_{\mathbf{x}}\frac{\exp\left(\alpha\mu(\mathbf{x},0,t)[1+\rho(1+\epsilon h_{T|\mathbf{x},S}(1-t|\mathbf{x},0))e_{1-t}(\mathbf{x},0)]\right)}{\exp\left(\alpha v(\mathbf{x})\right)+\exp\left(\alpha\mu(\mathbf{x},0,t)[1+\rho(1+\epsilon h_{T|\mathbf{x},S}(1-t|\mathbf{x},0))e_{1-t}(\mathbf{x},0)]\right)}$$
$$\times\mu(\mathbf{x},0,t)[1+\rho(1+\epsilon h_{T|\mathbf{x},S}(1-t|\mathbf{x},0))e_{1-t}(\mathbf{x},0)]p(0|\mathbf{x})p(x)dx|_{\epsilon=0}.$$

This is a rather complex partial derivative. To proceed, we will let

$$w_{\epsilon_T}(\mathbf{x})=\mu(\mathbf{x},0,t)[1+\rho(1+\epsilon h_{T|\mathbf{x},S}(1-t|\mathbf{x},0))e_{1-t}(\mathbf{x},0)]$$

and rewrite the above as

$$=\int_{\mathbf{x}}\frac{\partial}{\partial\epsilon}\left[\frac{v(\mathbf{x})\exp\left(\alpha v(\mathbf{x})\right)}{\exp\left(\alpha v(\mathbf{x})\right)+\exp\left(\alpha w_{\epsilon_T}(\mathbf{x})\right)}+\frac{w_{\epsilon_T}(\mathbf{x})\exp\left(\alpha w_{\epsilon_T}(\mathbf{x})\right)}{\exp\left(\alpha v(\mathbf{x})\right)+\exp\left(\alpha w_{\epsilon_T}(\mathbf{x})\right)}\right]\Big|_{\epsilon=0}p(0|\mathbf{x})p(x)dx.$$

Note that

$$\frac{\partial}{\partial\epsilon}w_{\epsilon_T}(\mathbf{x})=h_{T|\mathbf{x},S}(1-t|\mathbf{x},0)\rho\mu(\mathbf{x},0,t)e_{1-t}(\mathbf{x},0)$$

and $w_{\epsilon_T}(\mathbf{x})|_{\epsilon=0} = w(\mathbf{x})$. From here, we evaluate the partial derivative piece by piece. First the left fraction:

$$\frac{\partial}{\partial\epsilon} \frac{v(\mathbf{x})\exp\left(\alpha v(\mathbf{x})\right)}{\exp\left(\alpha v(\mathbf{x})\right) + \exp\left(\alpha w_{\epsilon_T}(\mathbf{x})\right)}\bigg|_{\epsilon=0} = \frac{-v(\mathbf{x})\exp\left(\alpha v(\mathbf{x})\right)\exp\left(\alpha w_{\epsilon_T}(\mathbf{x})\right) \times \alpha\frac{\partial}{\partial\epsilon}w_{\epsilon_T}(\mathbf{x})}{\left[\exp\left(\alpha v(\mathbf{x})\right) + \exp\left(\alpha w_{\epsilon_T}(\mathbf{x})\right)\right]^2}\bigg|_{\epsilon=0}$$

$$= \frac{-\alpha v(\mathbf{x})\frac{\partial}{\partial\epsilon}w_{\epsilon_T}(\mathbf{x})\exp\left(\alpha v(\mathbf{x})\right)\exp\left(\alpha w(\mathbf{x})\right)}{\left[\exp\left(\alpha v(\mathbf{x})\right) + \exp\left(\alpha w(\mathbf{x})\right)\right]^2}$$

Second the right fraction:

$$\frac{\partial}{\partial\epsilon}\left(\frac{w_{\epsilon_T}(\mathbf{x})\exp\left(\alpha w_{\epsilon_T}(\mathbf{x})\right)}{\exp\left(\alpha v(\mathbf{x})\right) + \exp\left(\alpha w_{\epsilon_T}(\mathbf{x})\right)}\right)\bigg|_{\epsilon=0}$$

$$= \Bigg\{\left[\exp\left(\alpha v(\mathbf{x})\right) + \exp\left(\alpha w_{\epsilon_T}(\mathbf{x})\right)\right]$$

$$\times \left[\frac{\partial w_{\epsilon_T}(\mathbf{x})}{\partial\epsilon}\exp\left(\alpha w_{\epsilon_T}(\mathbf{x})\right) + w_{\epsilon_T}(\mathbf{x})\exp\left(\alpha w_{\epsilon_T}(\mathbf{x})\right) \times \alpha\frac{\partial w_{\epsilon_T}(\mathbf{x})}{\partial\epsilon}\right]$$

$$- w_{\epsilon_T}(\mathbf{x})\exp\left(2\alpha w_{\epsilon_T}(\mathbf{x})\right) \times \alpha\frac{\partial w_{\epsilon_T}(\mathbf{x})}{\partial\epsilon}\Bigg\}\left[\exp\left(\alpha v(\mathbf{x})\right) + \exp\left(\alpha w_{\epsilon_T}(\mathbf{x})\right)\right]^{-2}\bigg|_{\epsilon=0}$$

$$= \frac{\frac{\partial w_{\epsilon_T}(\mathbf{x})}{\partial\epsilon} \times \left[\exp\left(\alpha v(\mathbf{x})\right)\exp\left(\alpha w(\mathbf{x})\right) + \alpha w(\mathbf{x})\exp\left(\alpha v(\mathbf{x})\right)\exp\left(\alpha w(\mathbf{x})\right) + \exp\left(2\alpha w(\mathbf{x})\right)\right]}{\left[\exp\left(\alpha v(\mathbf{x})\right) + \exp\left(\alpha w(\mathbf{x})\right)\right]^2}$$

Putting these two terms together, we have

$$\frac{\partial w_{\epsilon_T}(\mathbf{x})}{\partial\epsilon} \times \Bigg[\frac{\exp\left(\alpha v(\mathbf{x})\right)\exp\left(\alpha w(\mathbf{x})\right) + \exp\left(2\alpha w(\mathbf{x})\right)}{\left[\exp\left(\alpha v(\mathbf{x})\right) + \exp\left(\alpha w(\mathbf{x})\right)\right]^2} +$$

$$\frac{\alpha w(\mathbf{x})\exp\left(\alpha v(\mathbf{x})\right)\exp\left(\alpha w(\mathbf{x})\right) - \alpha v(\mathbf{x})\exp\left(\alpha v(\mathbf{x})\right)\exp\left(\alpha w(\mathbf{x})\right)}{\left[\exp\left(\alpha v(\mathbf{x})\right) + \exp\left(\alpha w(\mathbf{x})\right)\right]^2}\Bigg]$$

$$=$$

$$\frac{\partial w_{\epsilon_T}(\mathbf{x})}{\partial\epsilon} \times \left[\lambda_2(\mathbf{x}) + \alpha\lambda_1(\mathbf{x})\lambda_2(\mathbf{x})\left[w(\mathbf{x}) - v(\mathbf{x})\right]\right]$$

$$=$$

$$h_{T|\mathbf{X},S}(1-t|\mathbf{x},0)\rho\mu(\mathbf{x},0,t)e_{1-t}(\mathbf{x},0)\left[\lambda_2(\mathbf{x}) + \alpha\lambda_1(\mathbf{x})\lambda_2(\mathbf{x})\left[w(\mathbf{x}) - v(\mathbf{x})\right]\right]$$

Now, we can plug this partial derivative evaluated at $\epsilon = 0$ back into our integral, to get that

$$\nabla_{h_{T|\mathbf{X},S}}\theta_2 = \int_{\mathbf{x}} h_{T|\mathbf{X},S}(1-t|\mathbf{x},0)\rho\mu(\mathbf{x},0,t)e_{1-t}(\mathbf{x},0)$$

$$\times \left[\lambda_2(\mathbf{x}) + \alpha\lambda_1(\mathbf{x})\lambda_2(\mathbf{x})\left[w(\mathbf{x}) - v(\mathbf{x})\right]\right]p(0|\mathbf{x})p(x)dx$$

$$= \int_{\mathbf{x}} p(0|\mathbf{x})\rho\mu(\mathbf{x},0,t)\left\{\lambda_2(\mathbf{x}) + \alpha\lambda_1(\mathbf{x})\lambda_2(\mathbf{x})\left[w(\mathbf{x}) - v(\mathbf{x})\right]\right\}$$

$$\times e_{1-t}(\mathbf{x},0)h_{T|\mathbf{X},S}(1-t|\mathbf{x},0)p(x)dx$$

$$= \mathbb{E}\Big[g_0(\mathbf{X})\rho\mu(\mathbf{X},0,t)\left\{\lambda_2(\mathbf{X}) + \alpha\lambda_1(\mathbf{X})\lambda_2(\mathbf{X})\left[w(\mathbf{X}) - v(\mathbf{X})\right]\right\}$$

$$\times e_{1-t}(\mathbf{X},0)h_{T|\mathbf{X},S}(1-t|\mathbf{X},0)\Big].$$

We proceed to show that $\mathbb{E}[\phi_{\theta_2} h_{T|\mathbf{X},S}]$ equals the above value. First, plugging in $h_{T|\mathbf{X},S}$,

$$
\begin{aligned}
\mathbb{E}[\phi_{\theta_2} h_{T|\mathbf{X},S}] =&(1+\gamma)\mathbb{E}[g_0(\mathbf{X})\{\lambda_1(\mathbf{X})+\alpha\lambda_1(\mathbf{X})\lambda_2(\mathbf{X})[v(\mathbf{X})-w(\mathbf{X})]\}\\
&\times \mathbb{E}[(Y-\mu(\mathbf{X},1,t))h_{T|\mathbf{X},S}(T|\mathbf{X},S)|\mathbf{X},S=1,T=t]]\\
&+\mathbb{E}[g_0(\mathbf{X})\{\lambda_1(\mathbf{X})v(\mathbf{X})+\lambda_2(\mathbf{X})w(\mathbf{X})\}\mathbb{E}[h_{T|\mathbf{X},S}(T|\mathbf{X},S)|\mathbf{X},S=0]]\\
&+\mathbb{E}[g_0(\mathbf{X})[1+\rho e_{1-t}(\mathbf{X},0)]\{\lambda_2(\mathbf{X})+\alpha\lambda_1(\mathbf{X})\lambda_2(\mathbf{X})[w(\mathbf{X})-v(\mathbf{X})]\}\\
&\times \mathbb{E}[(Y-\mu(\mathbf{X},0,t))h_{T|\mathbf{X},S}(T|\mathbf{X},S)|\mathbf{X},S=0,T=t]]\\
&+\mathbb{E}[g_0(\mathbf{X})\rho\mu(\mathbf{X},0,t)\{\lambda_2(\mathbf{X})+\alpha\lambda_1(\mathbf{X})\lambda_2(\mathbf{X})[w(\mathbf{X})-v(\mathbf{X})]\}\\
&\times \mathbb{E}[[\mathbb{I}_{1-t}(T)-e_{1-t}(\mathbf{X},0)]h_{T|\mathbf{X},S}(T|\mathbf{X},S)|\mathbf{X},S=0]].
\end{aligned}
$$

Then we pull $h_{T|\mathbf{X},S}$ out of the inner expectation where possible

$$
\begin{aligned}
\mathbb{E}[\phi_{\theta_2} h_{T|\mathbf{X},S}] =&(1+\gamma)\mathbb{E}[g_0(\mathbf{X})\{\lambda_1(\mathbf{X})+\alpha\lambda_1(\mathbf{X})\lambda_2(\mathbf{X})[v(\mathbf{X})-w(\mathbf{X})]\}\\
&\times h_{T|\mathbf{X},S}(t|\mathbf{X},1)\mathbb{E}[(Y-\mu(\mathbf{X},1,t))|\mathbf{X},S=1,T=t]]\\
&+\mathbb{E}[g_0(\mathbf{X})\{\lambda_1(\mathbf{X})v(\mathbf{X})+\lambda_2(\mathbf{X})w(\mathbf{X})\}\mathbb{E}[h_{T|\mathbf{X},S}(T|\mathbf{X},S)|\mathbf{X},S=0]]\\
&+\mathbb{E}[g_0(\mathbf{X})[1+\rho e_{1-t}(\mathbf{X},0)]\{\lambda_2(\mathbf{X})+\alpha\lambda_1(\mathbf{X})\lambda_2(\mathbf{X})[w(\mathbf{X})-v(\mathbf{X})]\}\\
&\times h_{T|\mathbf{X},S}(t|\mathbf{X},0)\mathbb{E}[(Y-\mu(\mathbf{X},0,t))|\mathbf{X},S=0,T=t]]\\
&+\mathbb{E}\Big[g_0(\mathbf{X})\rho\mu(\mathbf{X},0,t)\{\lambda_2(\mathbf{X})+\alpha\lambda_1(\mathbf{X})\lambda_2(\mathbf{X})[w(\mathbf{X})-v(\mathbf{X})]\}\\
&\times \Big(\mathbb{E}[\mathbb{I}_{1-t}(T)h_{T|\mathbf{X},S}(T|\mathbf{X},S)|\mathbf{X},S=0]\\
&\qquad - e_{1-t}(\mathbf{X},0)\mathbb{E}[h_{T|\mathbf{X},S}(T|\mathbf{X},S)|\mathbf{X},S=0]\Big)\Big],
\end{aligned}
$$

and use the mean zero property of the score function to set all terms multiplied by $\mathbb{E}[h_{T|\mathbf{X},S}(T|\mathbf{X},S)|\mathbf{X},S=0]$ equal to zero:

$$
\begin{aligned}
\mathbb{E}[\phi_{\theta_2} h_{T|\mathbf{X},S}] =&(1+\gamma)\mathbb{E}[g_0(\mathbf{X})\{\lambda_1(\mathbf{X})+\alpha\lambda_1(\mathbf{X})\lambda_2(\mathbf{X})[v(\mathbf{X})-w(\mathbf{X})]\}\\
&\times h_{T|\mathbf{X},S}(t|\mathbf{X},1)\mathbb{E}[(Y-\mu(\mathbf{X},1,t))|\mathbf{X},S=1,T=t]]\\
&+\mathbb{E}[g_0(\mathbf{X})[1+\rho e_{1-t}(\mathbf{X},0)]\{\lambda_2(\mathbf{X})+\alpha\lambda_1(\mathbf{X})\lambda_2(\mathbf{X})[w(\mathbf{X})-v(\mathbf{X})]\}\\
&\times h_{T|\mathbf{X},S}(t|\mathbf{X},0)\mathbb{E}[(Y-\mu(\mathbf{X},0,t))|\mathbf{X},S=0,T=t]]\\
&+\mathbb{E}[g_0(\mathbf{X})\rho\mu(\mathbf{X},0,t)\{\lambda_2(\mathbf{X})+\alpha\lambda_1(\mathbf{X})\lambda_2(\mathbf{X})[w(\mathbf{X})-v(\mathbf{X})]\}\\
&\times \mathbb{E}[\mathbb{I}_{1-t}(T)h_{T|\mathbf{X},S}(T|\mathbf{X},S)|\mathbf{X},S=0]].
\end{aligned}
$$

Then applying the inner expectations and canceling like terms we get

$$
\begin{aligned}
\mathbb{E}[\phi_{\theta_2} h_{T|\mathbf{X},S}] =&(1+\gamma)\mathbb{E}[g_0(\mathbf{X})\{\lambda_1(\mathbf{X})+\alpha\lambda_1(\mathbf{X})\lambda_2(\mathbf{X})[v(\mathbf{X})-w(\mathbf{X})]\}\\
&\times h_{T|\mathbf{X},S}(t|\mathbf{X},1)(\mu(\mathbf{X},1,t)-\mu(\mathbf{X},1,t))]\\
&+\mathbb{E}[g_0(\mathbf{X})[1+\rho e_{1-t}(\mathbf{X},0)]\{\lambda_2(\mathbf{X})+\alpha\lambda_1(\mathbf{X})\lambda_2(\mathbf{X})[w(\mathbf{X})-v(\mathbf{X})]\}\\
&\times h_{T|\mathbf{X},S}(t|\mathbf{X},0)(\mu(\mathbf{X},0,t)-\mu(\mathbf{X},0,t))]\\
&+\mathbb{E}[g_0(\mathbf{X})\rho\mu(\mathbf{X},0,t)\{\lambda_2(\mathbf{X})+\alpha\lambda_1(\mathbf{X})\lambda_2(\mathbf{X})[w(\mathbf{X})-v(\mathbf{X})]\}\\
&\times \mathbb{E}[\mathbb{I}_{1-t}(T)h_{T|\mathbf{X},S}(T|\mathbf{X},S)|\mathbf{X},S=0]]\\
=&\mathbb{E}[g_0(\mathbf{X})\rho\mu(\mathbf{X},0,t)\{\lambda_2(\mathbf{X})+\alpha\lambda_1(\mathbf{X})\lambda_2(\mathbf{X})[w(\mathbf{X})-v(\mathbf{X})]\}\\
&\times \mathbb{E}[\mathbb{I}_{1-t}(T)h_{T|\mathbf{X},S}(T|\mathbf{X},S)|\mathbf{X},S=0]].
\end{aligned}
$$

Now, notice that

$$\mathbb{E}[\mathbb{I}_{1-t}(T)h_{T|\mathbf{X},S}(T|\mathbf{X},S)|\mathbf{X},S=0] = \mathbb{E}\left[\sum_{t'\in\{t,1-t\}}\mathbb{I}_{1-t}(t')h_{T|\mathbf{X},S}(t'|\mathbf{X},0)e_{t'}(\mathbf{X},0)\right]$$

$$= \mathbb{E}\big[e_{1-t}(\mathbf{X},0)h_{T|\mathbf{X},S}(1-t|\mathbf{X},0)\big].$$

Therefore,

$$\mathbb{E}[\phi_{\theta_2}h_{T|\mathbf{X},S}] = \mathbb{E}\Big[g_0(\mathbf{X})\rho\mu(\mathbf{X},0,t)\{\lambda_2(\mathbf{X}) + \alpha\lambda_1(\mathbf{X})\lambda_2(\mathbf{X})[w(\mathbf{X})-v(\mathbf{X})]\}$$

$$\times e_{1-t}(\mathbf{X},0)h_{T|\mathbf{X},S}(1-t|\mathbf{X},0)\Big],$$

and we have shown that $\nabla_{h_{T|\mathbf{X},S}}\theta_2 = \mathbb{E}[\phi_{\theta_2}h_{T|\mathbf{X},S}]$

Finally, we show that $\nabla_{h_{Y|\mathbf{X},S,T}}\theta_2 = \mathbb{E}[\phi_{\theta_2}h_{Y|\mathbf{X},S,T}]$. As always, we start by simplifying $\nabla_{h_{Y|\mathbf{X},S,T}}\theta_2$ by replacing the corresponding terms in the factorized distribution function. This time we set $\tilde{p}_\epsilon(y|\mathbf{x},1,t) = (1 + \epsilon h_{Y|\mathbf{X},S,T}(y|\mathbf{x},1,t))p(y|\mathbf{x},1,t)$ and $\tilde{p}_\epsilon(y|\mathbf{x},0,t) = (1 + \epsilon h_{Y|\mathbf{X},S,T}(y|\mathbf{x},0,t))p(y|\mathbf{x},0,t)$. We set the other conditional probability density functions to their normal $p$ form. We again have to format $\nabla_{h_{Y|\mathbf{X},S,T}}\theta_2$ slightly differently to allow it to stay within the page margins.

$$\nabla_{h_{T|\mathbf{X},S}}\theta_2 = \frac{\partial}{\partial\epsilon}\int_{\mathbf{x}}\exp\left(\alpha(1+\gamma)\int_y y(1+\epsilon h_{Y|\mathbf{X},S,T}(y|\mathbf{x},1,t))p(y|\mathbf{x},1,t)dy\right)$$

$$\times\left\{\exp\left(\alpha\int_y y(1+\epsilon h_{Y|\mathbf{X},S,T}(y|\mathbf{x},0,t))p(y|\mathbf{x},0,t)dy[1+\rho p(1-t|\mathbf{x},0)]\right)\right.$$

$$\left.+\exp\left(\alpha(1+\gamma)\int_y y(1+\epsilon h_{Y|\mathbf{X},S,T}(y|\mathbf{x},1,t))p(y|\mathbf{x},1,t)dy\right)\right\}^{-1}$$

$$\times\left((1+\gamma)\int_y y(1+\epsilon h_{Y|\mathbf{X},S,T}(y|\mathbf{x},1,t))p(y|\mathbf{x},1,t)dy\right)p(0|\mathbf{x})p(x)dx\bigg|_{\epsilon=0}$$

$$+\frac{\partial}{\partial\epsilon}\int_{\mathbf{x}}\exp\left(\alpha\int_y y(1+\epsilon h_{Y|\mathbf{X},S,T}(y|\mathbf{x},0,t))p(y|\mathbf{x},0,t)dy[1+\rho p(1-t|\mathbf{x},0)]\right)$$

$$\times\left\{\exp\left(\alpha\int_y y(1+\epsilon h_{Y|\mathbf{X},S,T}(y|\mathbf{x},0,t))p(y|\mathbf{x},0,t)dy[1+\rho p(1-t|\mathbf{x},0)]\right)\right.$$

$$\left.+\exp\left(\alpha(1+\gamma)\int_y y(1+\epsilon h_{Y|\mathbf{X},S,T}(y|\mathbf{x},1,t))p(y|\mathbf{x},1,t)dy\right)\right\}^{-1}$$

$$\times\left(\int_y y(1+\epsilon h_{Y|\mathbf{X},S,T}(y|\mathbf{x},0,t))p(y|\mathbf{x},0,t)dy[1+\rho p(1-t|\mathbf{x},0)]\right)$$

$$\times p(0|\mathbf{x})p(x)dx\bigg|_{\epsilon=0}.$$

We define the following terms to substitute in above:

$$v_{\epsilon_Y}(\mathbf{x}) = (1+\gamma)\int_y y(1+\epsilon h_{Y|\mathbf{X},S,T}(y|\mathbf{x},1,t))p(y|\mathbf{x},1,t)dy$$

$$= v(\mathbf{x}) + \epsilon(1+\gamma)\int_y y h_{Y|\mathbf{X},S,T}(y|\mathbf{x},1,t)p(y|\mathbf{x},1,t)dy, \text{ and}$$

$$w_{\epsilon_Y}(\mathbf{x}) = \int_y y(1+\epsilon h_{Y|\mathbf{X},S,T}(y|\mathbf{x},0,t))p(y|\mathbf{x},0,t)dy[1+\rho p(1-t|\mathbf{x},0)]$$

$$= w(x) + \epsilon[1+\rho e_{1-t}(\mathbf{x},0)]\int_y y h_{Y|\mathbf{X},S,T}(y|\mathbf{x},0,t)p(y|\mathbf{x},0,t)dy.$$

Note that

$$\frac{\partial v_{\epsilon_Y}(\mathbf{x})}{\partial \epsilon} = (1+\gamma)\int_y y h_{Y|\mathbf{X},S,T}(y|\mathbf{x},1,t)p(y|\mathbf{x},1,t)dy,$$

$$\frac{\partial w_{\epsilon_Y}(\mathbf{x})}{\partial \epsilon} = [1+\rho e_{1-t}(\mathbf{x},0)]\int_y y h_{Y|\mathbf{X},S,T}(y|\mathbf{x},0,t)p(y|\mathbf{x},0,t)dy,$$

and $v_{\epsilon_Y}(\mathbf{x})|_{\epsilon=0} = v(\mathbf{x})$, $w_{\epsilon_Y}(\mathbf{x})|_{\epsilon=0} = w(\mathbf{x})$. Then,

$$\nabla_{h_{Y|\mathbf{X},S,T}}\theta_2 = \int_{\mathbf{x}}\frac{\partial}{\partial \epsilon}\left[\frac{v_{\epsilon_Y}(\mathbf{x})\exp\left(\alpha v_{\epsilon_Y}(\mathbf{x})\right)+w_{\epsilon_Y}(\mathbf{x})\exp\left(\alpha w_{\epsilon_Y}(\mathbf{x})\right)}{\exp\left(\alpha v_{\epsilon_Y}(\mathbf{x})\right)+\exp\left(\alpha w_{\epsilon_Y}(\mathbf{x})\right)}\right]\Bigg|_{\epsilon=0}p(0|\mathbf{x})p(x)dx.$$

Like above, we evaluate the partial derivative piece by piece for clarity. First, the left term of the fraction:

$$\frac{\partial}{\partial \epsilon}\frac{v_{\epsilon_Y}(\mathbf{x})\exp\left(\alpha v_{\epsilon_Y}(\mathbf{x})\right)}{\exp\left(\alpha v_{\epsilon_Y}(\mathbf{x})\right)+\exp\left(\alpha w_{\epsilon_Y}(\mathbf{x})\right)}\Bigg|_{\epsilon=0}$$

$$= \Bigg\{ \left[\exp\left(\alpha v_{\epsilon_Y}(\mathbf{x})\right)+\exp\left(\alpha w_{\epsilon_Y}(\mathbf{x})\right)\right]$$

$$\times \left[\frac{\partial v_{\epsilon_Y}(\mathbf{x})}{\partial \epsilon}\exp\left(\alpha v_{\epsilon_Y}(\mathbf{x})\right)+v_{\epsilon_Y}(\mathbf{x})\exp\left(\alpha v_{\epsilon_Y}(\mathbf{x})\right)\times\alpha\frac{\partial v_{\epsilon_Y}(\mathbf{x})}{\partial \epsilon}\right]$$

$$- v_{\epsilon_Y}(\mathbf{x})\exp\left(\alpha v_{\epsilon_Y}(\mathbf{x})\right)\left[\exp\left(\alpha v_{\epsilon_Y}(\mathbf{x})\right)\times\alpha\frac{\partial v_{\epsilon_Y}(\mathbf{x})}{\partial \epsilon}+\exp\left(\alpha w_{\epsilon_Y}(\mathbf{x})\right)\times\alpha\frac{\partial w_{\epsilon_Y}(\mathbf{x})}{\partial \epsilon}\right]\Bigg\}$$

$$\times \left[\exp\left(\alpha v_{\epsilon_Y}(\mathbf{x})\right)+\exp\left(\alpha w_{\epsilon_Y}(\mathbf{x})\right)\right]^{-2}\Bigg|_{\epsilon=0}$$

$$= \frac{\frac{\partial v_{\epsilon_Y}(\mathbf{x})}{\partial \epsilon}\times\left[\exp\left(\alpha v(\mathbf{x})\right)\exp\left(\alpha w(\mathbf{x})\right)+\alpha v(\mathbf{x})\exp\left(\alpha v(\mathbf{x})\right)\exp\left(\alpha w(\mathbf{x})\right)+\exp\left(2\alpha v(\mathbf{x})\right)\right]}{\left[\exp\left(\alpha v(\mathbf{x})\right)+\exp\left(\alpha w(\mathbf{x})\right)\right]^2}$$

$$- \frac{\frac{\partial w_{\epsilon_Y}(\mathbf{x})}{\partial \epsilon}\times\alpha v(\mathbf{x})\exp\left(\alpha v(\mathbf{x})\right)\exp\left(\alpha w(\mathbf{x})\right)}{\left[\exp\left(\alpha v(\mathbf{x})\right)+\exp\left(\alpha w(\mathbf{x})\right)\right]^2}$$

The right fraction partial derivative looks similar:

$$\frac{\partial}{\partial \epsilon} \frac{w_{\epsilon_Y}(\mathbf{x}) \exp\left(\alpha w_{\epsilon_Y}(\mathbf{x})\right)}{\exp\left(\alpha v_{\epsilon_Y}(\mathbf{x})\right) + \exp\left(\alpha w_{\epsilon_Y}(\mathbf{x})\right)}\Bigg|_{\epsilon=0}$$

$$= \Bigg\{ \left[\exp\left(\alpha v_{\epsilon_Y}(\mathbf{x})\right) + \exp\left(\alpha w_{\epsilon_Y}(\mathbf{x})\right)\right]$$

$$\times \left[\frac{\partial w_{\epsilon_Y}(\mathbf{x})}{\partial \epsilon} \exp\left(\alpha w_{\epsilon_Y}(\mathbf{x})\right) + w_{\epsilon_Y}(\mathbf{x}) \exp\left(\alpha w_{\epsilon_Y}(\mathbf{x})\right) \times \alpha \frac{\partial w_{\epsilon_Y}(\mathbf{x})}{\partial \epsilon}\right]$$

$$- w_{\epsilon_Y}(\mathbf{x}) \exp\left(\alpha w_{\epsilon_Y}(\mathbf{x})\right) \left[\exp\left(\alpha v_{\epsilon_Y}(\mathbf{x})\right) \times \alpha \frac{\partial v_{\epsilon_Y}(\mathbf{x})}{\partial \epsilon} + \exp\left(\alpha w_{\epsilon_Y}(\mathbf{x})\right) \times \alpha \frac{\partial w_{\epsilon_Y}(\mathbf{x})}{\partial \epsilon}\right] \Bigg\}$$

$$\times \left[\exp\left(\alpha v_{\epsilon_Y}(\mathbf{x})\right) + \exp\left(\alpha w_{\epsilon_Y}(\mathbf{x})\right)\right]^{-2}\Bigg|_{\epsilon=0}$$

$$= \frac{\frac{\partial w_{\epsilon_Y}(\mathbf{x})}{\partial \epsilon} \times \left[\exp\left(\alpha v(\mathbf{x})\right) \exp\left(\alpha w(\mathbf{x})\right) + \alpha w(\mathbf{x}) \exp\left(\alpha v(\mathbf{x})\right) \exp\left(\alpha w(\mathbf{x})\right) + \exp\left(2\alpha w(\mathbf{x})\right)\right]}{\left[\exp\left(\alpha v(\mathbf{x})\right) + \exp\left(\alpha w(\mathbf{x})\right)\right]^2}$$

$$- \frac{\frac{\partial v_{\epsilon_Y}(\mathbf{x})}{\partial \epsilon} \times \alpha w(\mathbf{x}) \exp\left(\alpha v(\mathbf{x})\right) \exp\left(\alpha w(\mathbf{x})\right)}{\left[\exp\left(\alpha v(\mathbf{x})\right) + \exp\left(\alpha w(\mathbf{x})\right)\right]^2}$$

Now, we combine terms and group by those multiplied to $\frac{\partial v_{\epsilon_Y}(\mathbf{x})}{\partial \epsilon}$ and $\frac{\partial w_{\epsilon_Y}(\mathbf{x})}{\partial \epsilon}$. First, $\frac{\partial v_{\epsilon_Y}(\mathbf{x})}{\partial \epsilon}$:

$$\frac{\partial v_{\epsilon_Y}(\mathbf{x})}{\partial \epsilon} \times \left[\frac{\exp\left(\alpha v(\mathbf{x})\right) \exp\left(\alpha w(\mathbf{x})\right) + \exp\left(2\alpha v(\mathbf{x})\right)}{\left[\exp\left(\alpha v(\mathbf{x})\right) + \exp\left(\alpha w(\mathbf{x})\right)\right]^2} + \right.$$

$$\left. \frac{\alpha v(\mathbf{x}) \exp\left(\alpha v(\mathbf{x})\right) \exp\left(\alpha w(\mathbf{x})\right) - \alpha w(\mathbf{x}) \exp\left(\alpha v(\mathbf{x})\right) \exp\left(\alpha w(\mathbf{x})\right)}{\left[\exp\left(\alpha v(\mathbf{x})\right) + \exp\left(\alpha w(\mathbf{x})\right)\right]^2}\right]$$

$$=$$

$$\frac{\partial v_{\epsilon_Y}(\mathbf{x})}{\partial \epsilon} \times \left[\lambda_1(\mathbf{x}) + \alpha \lambda_1(\mathbf{x}) \lambda_2(\mathbf{x}) \left[v(\mathbf{x}) - w(\mathbf{x})\right]\right]$$

$$=$$

$$(1 + \gamma) \int_y y h_{Y|\mathbf{X},S,T}(y|\mathbf{x}, 1, t) p(y|\mathbf{x}, 1, t) dy \times \left[\lambda_1(\mathbf{x}) + \alpha \lambda_1(\mathbf{x}) \lambda_2(\mathbf{x}) \left[v(\mathbf{x}) - w(\mathbf{x})\right]\right].$$

Next, $\frac{\partial w_{\epsilon_Y}(\mathbf{x})}{\partial \epsilon}$:

$$\frac{\partial w_{\epsilon_Y}(\mathbf{x})}{\partial \epsilon} \times \left[\frac{\exp\left(\alpha v(\mathbf{x})\right) \exp\left(\alpha w(\mathbf{x})\right) + \exp\left(2\alpha w(\mathbf{x})\right)}{\left[\exp\left(\alpha v(\mathbf{x})\right) + \exp\left(\alpha w(\mathbf{x})\right)\right]^2} + \right.$$

$$\left. \frac{\alpha w(\mathbf{x}) \exp\left(\alpha v(\mathbf{x})\right) \exp\left(\alpha w(\mathbf{x})\right) - \alpha v(\mathbf{x}) \exp\left(\alpha v(\mathbf{x})\right) \exp\left(\alpha w(\mathbf{x})\right)}{\left[\exp\left(\alpha v(\mathbf{x})\right) + \exp\left(\alpha w(\mathbf{x})\right)\right]^2}\right]$$

$$=$$

$$\frac{\partial w_{\epsilon_Y}(\mathbf{x})}{\partial \epsilon} \times \left[\lambda_2(\mathbf{x}) + \alpha \lambda_1(\mathbf{x}) \lambda_2(\mathbf{x}) \left[w(\mathbf{x}) - v(\mathbf{x})\right]\right]$$

$$=$$

$$[1 + \rho e_{1-t}(\mathbf{x}, 0)] \int_y y h_{Y|\mathbf{X},S,T}(y|\mathbf{x}, 0, t) p(y|\mathbf{x}, 0, t) dy \left[\lambda_2(\mathbf{x}) + \alpha \lambda_1(\mathbf{x}) \lambda_2(\mathbf{x}) \left[w(\mathbf{x}) - v(\mathbf{x})\right]\right].$$

We plug both of these components that make up the partial derivative evaluated at $\epsilon = 0$ back into our integral:

$$\nabla_{h_{Y|\mathbf{X},S,T}}\theta_2 = \int_{\mathbf{x}}(1+\gamma)\int_y yh_{Y|\mathbf{X},S,T}(y|\mathbf{x},1,t)p(y|\mathbf{x},1,t)dy$$

$$\times \left[\lambda_1(\mathbf{x})+\alpha\lambda_1(\mathbf{x})\lambda_2(\mathbf{x})\left[v(\mathbf{x})-w(\mathbf{x})\right]\right]p(0|\mathbf{x})p(x)dx$$

$$+\int_{\mathbf{x}}[1+\rho e_{1-t}(\mathbf{x},0)]\int_y yh_{Y|\mathbf{X},S,T}(y|\mathbf{x},0,t)p(y|\mathbf{x},0,t)dy$$

$$\times \left[\lambda_2(\mathbf{x})+\alpha\lambda_1(\mathbf{x})\lambda_2(\mathbf{x})\left[w(\mathbf{x})-v(\mathbf{x})\right]\right]p(0|\mathbf{x})p(x)dx.$$

Reorganizing terms and writing in expectation form, we get

$$\nabla_{h_{Y|\mathbf{X},S,T}}\theta_2 = (1+\gamma)\mathbb{E}\bigg[g_0(\mathbf{X})\left\{\lambda_1(\mathbf{X})+\alpha\lambda_1(\mathbf{X})\lambda_2(\mathbf{X})\left[v(\mathbf{X})-w(\mathbf{X})\right]\right\}$$

$$\times \mathbb{E}[Yh_{Y|\mathbf{X},S,T}(Y|\mathbf{X},S,T)|\mathbf{X},S=1,T=t]\bigg]$$

$$+\mathbb{E}\bigg[g_0(\mathbf{X})[1+\rho e_{1-t}(\mathbf{X},0)]\left\{\lambda_2(\mathbf{X})+\alpha\lambda_1(\mathbf{X})\lambda_2(\mathbf{X})\left[w(\mathbf{X})-v(\mathbf{X})\right]\right\}$$

$$\times \mathbb{E}[Yh_{Y|\mathbf{X},S,T}(Y|\mathbf{X},S,T)|\mathbf{X},S=0,T=t]\bigg].$$

We finish by showing that this equals $\mathbb{E}[\phi_{\theta_2}h_{Y|\mathbf{X},S,T}]$. First, setting it up,

$$\mathbb{E}[\phi_{\theta_2}h_{Y|\mathbf{X},S,T}] = (1+\gamma)\mathbb{E}[g_0(\mathbf{X})\left\{\lambda_1(\mathbf{X})+\alpha\lambda_1(\mathbf{X})\lambda_2(\mathbf{X})\left[v(\mathbf{X})-w(\mathbf{X})\right]\right\}$$

$$\times \mathbb{E}[(Y-\mu(\mathbf{X},1,t))h_{Y|\mathbf{X},S,T}(Y|\mathbf{X},S,T)|\mathbf{X},S=1,T=t]]$$

$$+\mathbb{E}[g_0(\mathbf{X})\left\{\lambda_1(\mathbf{X})v(\mathbf{X})+\lambda_2(\mathbf{X})w(\mathbf{X})\right\}\mathbb{E}[h_{Y|\mathbf{X},S,T}(Y|\mathbf{X},S,T)|\mathbf{X},S=0]]$$

$$+\mathbb{E}[g_0(\mathbf{X})[1+\rho e_{1-t}(\mathbf{X},0)]\left\{\lambda_2(\mathbf{X})+\alpha\lambda_1(\mathbf{X})\lambda_2(\mathbf{X})\left[w(\mathbf{X})-v(\mathbf{X})\right]\right\}$$

$$\times \mathbb{E}[(Y-\mu(\mathbf{X},0,t))h_{Y|\mathbf{X},S,T}(Y|\mathbf{X},S,T)|\mathbf{X},S=0,T=t]]$$

$$+\mathbb{E}[g_0(\mathbf{X})\rho\mu(\mathbf{X},0,t)\left\{\lambda_2(\mathbf{X})+\alpha\lambda_1(\mathbf{X})\lambda_2(\mathbf{X})\left[w(\mathbf{X})-v(\mathbf{X})\right]\right\}$$

$$\times \mathbb{E}[[1-T-e_{1-t}(\mathbf{X},0)]h_{Y|\mathbf{X},S,T}(Y|\mathbf{X},S,T)|\mathbf{X},S=0]].$$

Then, we use linearity of expectation to separate terms,

$$\mathbb{E}[\phi_{\theta_2}h_{Y|\mathbf{X},S,T}] = (1+\gamma)\mathbb{E}\bigg[g_0(\mathbf{X})\left\{\lambda_1(\mathbf{X})+\alpha\lambda_1(\mathbf{X})\lambda_2(\mathbf{X})\left[v(\mathbf{X})-w(\mathbf{X})\right]\right\}$$

$$\times \mathbb{E}[Yh_{Y|\mathbf{X},S,T}(Y|\mathbf{X},S,T)|\mathbf{X},S=1,T=t]$$

$$-\mu(\mathbf{X},1,t)\mathbb{E}[h_{Y|\mathbf{X},S,T}(Y|\mathbf{X},S,T)|\mathbf{X},S=1,T=t]\bigg]]$$

$$+\mathbb{E}[g_0(\mathbf{X})\left\{\lambda_1(\mathbf{X})v(\mathbf{X})+\lambda_2(\mathbf{X})w(\mathbf{X})\right\}\mathbb{E}[h_{Y|\mathbf{X},S,T}(Y|\mathbf{X},S,T)|\mathbf{X},S=0]]$$

$$+\mathbb{E}\bigg[g_0(\mathbf{X})[1+\rho e_{1-t}(\mathbf{X},0)]\left\{\lambda_2(\mathbf{X})+\alpha\lambda_1(\mathbf{X})\lambda_2(\mathbf{X})\left[w(\mathbf{X})-v(\mathbf{X})\right]\right\}$$

$$\times \mathbb{E}[Yh_{Y|\mathbf{X},S,T}(Y|\mathbf{X},S,T)|\mathbf{X},S=0,T=t]$$

$$-\mu(\mathbf{X},0,t)\mathbb{E}[h_{Y|\mathbf{X},S,T}(Y|\mathbf{X},S,T)|\mathbf{X},S=0,T=t]\bigg]$$

$$+\mathbb{E}[g_0(\mathbf{X})\rho\mu(\mathbf{X},0,t)\left\{\lambda_2(\mathbf{X})+\alpha\lambda_1(\mathbf{X})\lambda_2(\mathbf{X})\left[w(\mathbf{X})-v(\mathbf{X})\right]\right\}$$

$$\times \mathbb{E}[[1-T-e_{1-t}(\mathbf{X},0)]h_{Y|\mathbf{X},S,T}(Y|\mathbf{X},S,T)|\mathbf{X},S=0]].$$

Now, we use the mean zero property of the score function to remove several of the terms,

$$\mathbb{E}[\phi_{\theta_2} h_{Y|\mathbf{X},S,T}] = (1+\gamma)\mathbb{E}\Big[g_0(\mathbf{X})\left\{\lambda_1(\mathbf{X}) + \alpha\lambda_1(\mathbf{X})\lambda_2(\mathbf{X})\left[v(\mathbf{X}) - w(\mathbf{X})\right]\right\}$$
$$\times \mathbb{E}[Y h_{Y|\mathbf{X},S,T}(Y|\mathbf{X},S,T)|\mathbf{X},S=1,T=t]\Big]$$
$$+ \mathbb{E}\Big[g_0(\mathbf{X})\left[1 + \rho e_{1-t}(\mathbf{X},0)\right]\left\{\lambda_2(\mathbf{X}) + \alpha\lambda_1(\mathbf{X})\lambda_2(\mathbf{X})\left[w(\mathbf{X}) - v(\mathbf{X})\right]\right\}$$
$$\times \mathbb{E}[Y h_{Y|\mathbf{X},S,T}(Y|\mathbf{X},S,T)|\mathbf{X},S=0,T=t]\Big].$$

This confirms that $\nabla_{h_{Y|\mathbf{X},S,T}}\theta_2 = \mathbb{E}[\phi_{\theta_2} h_{Y|\mathbf{X},S,T}]$. The verification of the candidate EIF for $\theta_2$ is complete.

### C.3 EIF for $\theta(t,\rho,\gamma,\alpha)$

Having derived and validated the candidate EIFs for $\theta_1$ and $\theta_2$, we simply use the linearity property of EIFs to write the full form the EIF for $\theta(t,\rho,\gamma,\alpha)$ as:

$$\phi(Z;t,\rho,\gamma,\alpha) = \frac{S\mathbb{I}(T=t)}{e_t(\mathbf{X},1)}\Big[Y - \mu(\mathbf{X},1,t)\Big] + S\mu(\mathbf{X},1,t)+$$
$$+ S(1+\gamma)\left\{\lambda_1(\mathbf{X}) + \alpha\lambda_1(\mathbf{X})\lambda_2(\mathbf{X})\left[v(\mathbf{X}) - w(\mathbf{X})\right]\right\}$$
$$\times \left\{\frac{\mathbb{I}_t(T)}{e_t(\mathbf{X},1)g_1(\mathbf{X})}\left[Y - \mu(\mathbf{X},1,t)\right]g_0(\mathbf{X})\right\}$$
$$+ [1-S]\left\{\lambda_1(\mathbf{X})v(\mathbf{X}) + \lambda_2(\mathbf{X})w(\mathbf{X})\right\}$$
$$+ [1-S]\left\{\lambda_2(\mathbf{X}) + \alpha\lambda_1(\mathbf{X})\lambda_2(\mathbf{X})\left[w(\mathbf{X}) - v(\mathbf{X})\right]\right\}$$
$$\times \Bigg\{\frac{\mathbb{I}_t(T)}{e_t(\mathbf{X},0)}\left[Y - \mu(\mathbf{X},0,t)\right]\left[1 + \rho e_{1-t}(\mathbf{X},0)\right]$$
$$+ \rho\mu(\mathbf{X},0,t)\left[\mathbb{I}_{1-t}(T) - e_{1-t}(\mathbf{X},0)\right]\Bigg\}$$
$$- \theta(t,\rho,\gamma,\alpha).$$

## D (In)compatible $\rho$ and $\gamma$

In Remark 1, we briefly discussed the reasoning behind the Lemma 1 and Theorem 1 conditions that $v(\mathbf{x},t,-\gamma) \le w(\mathbf{x},t,\rho)$ and $w(\mathbf{x},t,-\rho) \le v(\mathbf{x},t,\gamma)$. We referred to $(\rho,\gamma)$ pairs that lead to violations in these conditions as incompatible, which we visually represented as one of the four regions in our breakdown frontier plots. In this Appendix section, we aim to provide more context on this concept and discuss how we estimate the (in)compatibility of a given $(\rho,\gamma)$ in practice.

### D.1 Further Discussion of Incompatibility

We define a pair $(\rho,\gamma)$ as incompatible if they do not sufficiently relax the assumptions to allow overlap between the bounds produced by $\gamma$ and $\rho$. We elaborate on this idea in this subsection.

Incompatibility stems from the fact that there must be some source of unmeasured confounding affecting either study selection or treatment assignment if the observed potential outcomes in the experimental and observational studies are different. In other words, as discussed in Remark 1, if $\exists(\mathbf{x},t)$ such that $|\mathbb{E}_{\mathcal{P}}[Y \mid \mathbf{X} = \mathbf{x}, S = 1, T = t] - \mathbb{E}_{\mathcal{P}}[Y \mid \mathbf{X} = \mathbf{x}, S = 0, T = t]| = \Delta(t) > 0$, then Assumption A6 and/or Assumption A5 must be violated.

Recall that in our partial identification framework

$$v(\mathbf{x},t,\gamma) := (1+\gamma)\mu(\mathbf{x},1,t), \quad w(\mathbf{x},t,\rho) := e_t(\mathbf{x},0)\mu(\mathbf{x},0,t) + e_{1-t}(\mathbf{x},0)(1+\rho)\mu(\mathbf{x},0,t),$$

serve as two upper bounds on $\mathbb{E}_{\mathcal{P}}[Y(t) \mid \mathbf{X} = \mathbf{x}, S = 0]$. Analogously, $v(\mathbf{x}, t, -\gamma)$ and $w(\mathbf{x}, t, -\rho)$ serve as two lower bounds. In particular, we showed in the proof of Lemma 1 in Appendix B.1 that

$$\mathbb{E}_{\mathcal{P}}[Y(t) \mid \mathbf{X} = \mathbf{x}, S = 0] \in [v(\mathbf{x}, t, -\gamma), v(\mathbf{x}, t, \gamma)], \text{ and}$$
$$\mathbb{E}_{\mathcal{P}}[Y(t) \mid \mathbf{X} = \mathbf{x}, S = 0] \in [w(\mathbf{x}, t, -\rho), w(\mathbf{x}, t, \rho)].$$

We then took the $\max$ over the two lower bounds and the $\min$ over the two upper bounds to get the tightest possible bounds on $\mathbb{E}_{\mathcal{P}}[Y(t) \mid \mathbf{X} = \mathbf{x}, S = 0]$ that we subsequently use to upper bound $\mathbb{E}_{\mathcal{P}}[Y(t) \mid \mathbf{X} = \mathbf{x}]$.

Consider the case that $\rho$ and $\gamma$ are both set to zero and $\Delta(t) > 0$. In this case, we have that the two intervals bounding $\mathbb{E}_{\mathcal{P}}[Y(t) \mid \mathbf{X} = \mathbf{x}, S = 0]$ are

$$\mathbb{E}_{\mathcal{P}}[Y(t) \mid \mathbf{X} = \mathbf{x}, S = 0] \in [v(\mathbf{x}, t, -0), v(\mathbf{x}, t, 0)] = \mu(\mathbf{x}, 1, t), \text{ and}$$
$$\mathbb{E}_{\mathcal{P}}[Y(t) \mid \mathbf{X} = \mathbf{x}, S = 0] \in [w(\mathbf{x}, t, -0), w(\mathbf{x}, t, 0)] = \mu(\mathbf{x}, 0, t).$$

But, as we established earlier, if $\Delta(t) > 0$, then $\mu(\mathbf{x}, 1, t) \neq \mu(\mathbf{x}, 0, t)$. This leads to a contradiction as the bounds based on $\rho = 0$ and $\gamma = 0$ assume no differences in potential outcomes across studies or treatment groups. But if $\mu(\mathbf{x}, 1, t) \neq \mu(\mathbf{x}, 0, t)$ is observed, then some violation of assumptions must be present. Hence, $(\rho, \gamma) = (0, 0)$ is incompatible with the data.

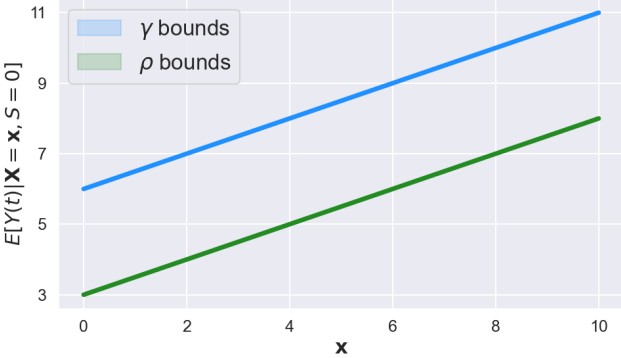

Figure 4: Bounds on $\mathbb{E}_{\mathcal{P}}[Y(t) \mid \mathbf{X} = \mathbf{x}, S = 0]$ from $[v(\mathbf{x}, t, -0), v(\mathbf{x}, t, 0)]$ and $[w(\mathbf{x}, t, -0), w(\mathbf{x}, t, 0)]$ when $\mu(\mathbf{x}, 1, t) \neq \mu(\mathbf{x}, 0, t)$.

Towards building up an understanding through a series of visuals, we depict this scenario in a toy example using Figure 4, where $\mu(\mathbf{x}, 1, t) \neq \mu(\mathbf{x}, 0, t)$. Here, it is clear that the bounds on $\mathbb{E}_{\mathcal{P}}[Y(t) \mid \mathbf{X} = \mathbf{x}, S = 0]$ do not intersect—leading us to call this an incompatible choice for $\rho$ and $\gamma$. Conversely, consider the scenario where we keep $\rho = 0$ but we increase $\gamma$ sufficiently so that $v(\mathbf{x}, t, -\gamma) = (1 - \gamma)\mu(\mathbf{x}, 1, t) < w(\mathbf{x}, t, 0) = \mu(\mathbf{x}, 0, t)$. This scenario is depicted in Figure 5. Here, we see that the bounds on $\mathbb{E}_{\mathcal{P}}[Y(t) \mid \mathbf{X} = \mathbf{x}, S = 0]$ do intersect—leading us to call this choice of $\rho$ and $\gamma$ a compatible pair of parameters.

Connecting this back to the conditions in Lemma 1 and Theorem 1, in Figure 4 we violated the condition that $v(\mathbf{x}, t, -\gamma) \leq w(\mathbf{x}, t, \rho)$ whereas in Figure 5 both $v(\mathbf{x}, t, -\gamma) \leq w(\mathbf{x}, t, \rho)$ and $w(\mathbf{x}, t, -\rho) \leq v(\mathbf{x}, t, \gamma)$.

When $\gamma$ and $\rho$ are both greater than zero, they can still be incompatible if either $v(\mathbf{x}, t, -\gamma) > w(\mathbf{x}, t, \rho)$ or $w(\mathbf{x}, t, -\rho) > v(\mathbf{x}, t, \gamma)$. Continuing with our example, consider the case depicted in Figure 6(a) where the lower bound on the $\gamma$ bound is larger than the upper bound on the $\rho$ bound, i.e. $v(\mathbf{x}, t, -\gamma) > w(\mathbf{x}, t, \rho)$. This is an example where $\Delta(t) > 0$ and the values of $\rho$ and $\gamma$ are not sufficiently large to explain this discrepency. Conversely, in Figure 6(b), the values of $\rho$ and $\gamma$ are increased, making $v(\mathbf{x}, t, -\gamma) \leq w(\mathbf{x}, t, \rho)$, and leading to compatible parameter values. In this plot, we also show how we take the $\max$ of the two lower bounds and the $\min$ of the two upper bounds to get the tightest bounds—corresponding to how we constructed the tightest bounds in Lemma 1 and Theorem 1.

## D.2 Estimating Incompatibility

In practice, we do not observe the true values of $v(\mathbf{x}, t, \gamma)w(\mathbf{x}, t, \rho)$ and $w(\mathbf{x}, t, \rho)$ and must estimate them from data. As a result, determining whether a given $(\rho, \gamma)$ pair is incompatible must account for estimation uncertainty.

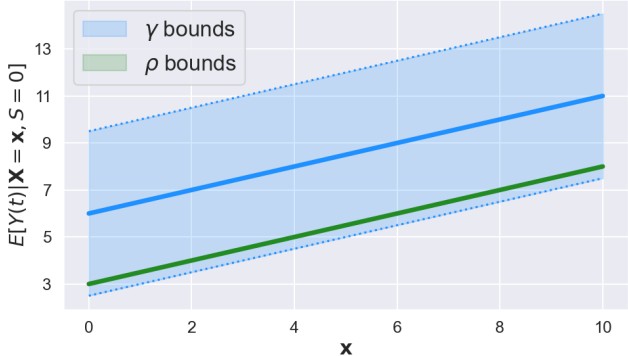

Figure 5: Bounds on $\mathbb{E}_{\mathcal{P}}[Y(t) \mid \mathbf{X} = \mathbf{x}, S = 0]$ from $[v(\mathbf{x}, t, -\gamma), v(\mathbf{x}, t, \gamma)]$ and $[w(\mathbf{x}, t, -0), w(\mathbf{x}, t, 0)]$ when $\mu(\mathbf{x}, 1, t) \neq \mu(\mathbf{x}, 0, t)$ and $\gamma$ is large enough that $v(\mathbf{x}, t, -\gamma) < w(\mathbf{x}, t, 0)$.

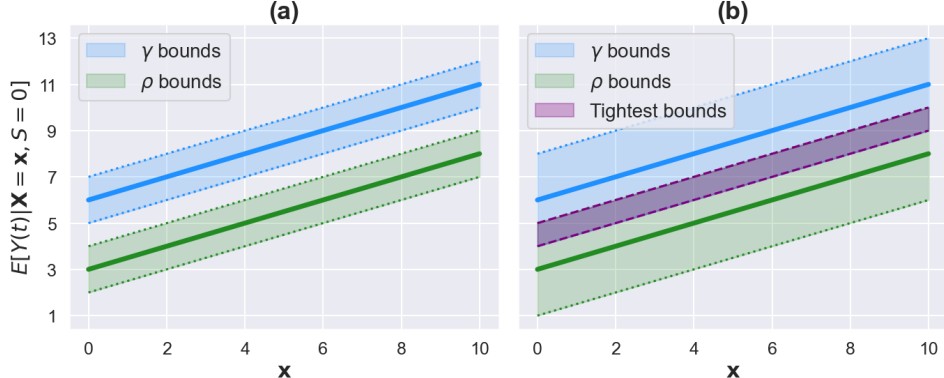

Figure 6: Bounds on $\mathbb{E}_{\mathcal{P}}[Y(t) \mid \mathbf{X} = \mathbf{x}, S = 0]$ from $[v(\mathbf{x}, t, -\gamma), v(\mathbf{x}, t, \gamma)]$ and $[w(\mathbf{x}, t, -\rho), w(\mathbf{x}, t, \rho)]$ when $\mu(\mathbf{x}, 1, t) \neq \mu(\mathbf{x}, 0, t)$. In **(a)**, $\rho$ and $\gamma$ are not large enough for the bounds to intersect. Whereas, in **(b)**, $\rho$ and $\gamma$ are made large enough for the bounds to intersect.

Ideally, for any given $(\rho, \gamma)$, we would test at each $(\mathbf{x}, t)$ whether the estimated bounds based on $\rho$ and $\gamma$ intersect, indicating compatibility. However, this is intractable with finite samples in high-dimensional covariate spaces. To address this challenge, we assess compatibility at the level of the target estimand, rather than at individual covariate profiles. Specifically, we evaluate whether the bounds intersect on average over $\mathbf{X}$, separately for each treatment arm $t \in \{0, 1\}$. When focusing on the CATE, this expectation is taken over the subpopulation of interest, whereas for the ATE, this expectation is taken over the full population.

As shown in the proof of Lemma 1 in Appendix B.1, the condition $\max\{v(\mathbf{x}, t, -\gamma), w(\mathbf{x}, t, -\rho)\} \leq \min\{v(\mathbf{x}, t, \gamma), w(\mathbf{x}, t, \rho)\}$ is equivalent to checking that both $v(\mathbf{x}, t, -\gamma) \leq w(\mathbf{x}, t, \rho)$ and $w(\mathbf{x}, t, -\rho) \leq v(\mathbf{x}, t, \gamma)$. Both conditions ensure that the $\rho$ and $\gamma$ intervals overlap. To make this condition testable, we assess whether the average size of the overlap region is nonnegative. Specifically, we check whether the expected difference of $\min\{v(\mathbf{x}, t, \gamma), w(\mathbf{x}, t, \rho)\} - \max\{v(\mathbf{x}, t, -\gamma), w(\mathbf{x}, t, -\rho)\} \geq 0$. We define $G(t) = \mathbb{E}[\min\{v(\mathbf{x}, t, \gamma), w(\mathbf{x}, t, \rho)\} - \max\{v(\mathbf{x}, t, -\gamma), w(\mathbf{x}, t, -\rho)\}]$. This expectation is taken over the relevant covariate distribution and is evaluated separately for each treatment arm $t \in \{0, 1\}$.

Because generating resamples that both satisfy the null and re-estimate the nuisance functions is non-trivial, we adopt a simplified resampling approach that treats these functions as fixed. We describe the full procedure below and return to its limitations at the end of this section.

For each treatment arm $t$, we conduct a one-sided hypothesis test where the null hypothesis is that the average size of the overlap region is nonnegative, $H_0 : G(t) \geq 0$, and the alternative is that the average size is negative, $H_A : G(t) < 0$. Rejecting the null corresponds to concluding that the

specified parameter pair is incompatible. Given a dataset $\mathcal{D}_n$, parameter values $(\rho', \gamma')$ and treatment arm $t'$, we construct a resampling-based test as follows. We first compute the observed test statistic as the sample mean of overlap sizes:

$$T_{obs} = \frac{1}{n} \sum_{i=1}^{n} \min\{v(\mathbf{x}_i, t', \gamma'), w(\mathbf{x}_i, t', \rho')\} - \max\{v(\mathbf{x}_i, t', -\gamma'), w(\mathbf{x}_i, t', -\rho')\}.$$

To simulate the null distribution where $G(t') = 0$, we subtract $T_{obs}$ from each individual overlap value, producing a centered dataset that has mean zero. We denote these centered values as $o_i = \min\{v(\mathbf{x}_i, t', \gamma'), w(\mathbf{x}_i, t', \rho')\} - \max\{v(\mathbf{x}_i, t', -\gamma'), w(\mathbf{x}_i, t', -\rho')\} - T_{obs}$. We then perform $R$ resampling iterations. In each iteration, $r$, we randomly draw $n$ values with replacement from the centered overlap size set $\{o_i, ..., o_n\}$, and compute the mean of the resampled values $T_r = \frac{1}{n} \sum_{j \in J_r} o_j$, where $J_r$ is the set of indices sampled in iteration $r$. Finally, we compute the one-sided p-value as the proportion of $T_r$ values that are greater than or equal to $T_{obs}$: $p = \frac{1}{R} \sum_{r=1}^{R} \mathbb{I}[T_r \geq T_{obs}]$.

We write this testing procedure as an algorithm in Appendix E.3. We also note two primary limitations of this approach.

First, as stated above, we do not test this condition at each $(\mathbf{x}, t)$ pair. This is primarily a practical limitation, as checking compatibility pointwise across the full covariate-treatment domain is infeasible in high dimensions or with continuous covariates. However, in settings with only discrete covariates and sufficiently large datasets, such a pointwise test could be applied to each covariate profile.

Second, as briefly mentioned above, our resampling-based procedure operates on estimated overlap values rather than resampling at the dataset level. This differs from a formal bootstrap test, which would involve resampling full observations and re-estimating the functions $e_t(\mathbf{x}, 0)$ and $\mu(\mathbf{x}, s, t)$ each time, propagating the uncertainty in estimating these quantities. In our setting it is extremely difficult to generate new datasets that satisfy the null $G(t') = 0$ while preserving the original data generation process. We therefore center the empirical distribution of the overlap values to satisfy the null and perform hypothesis testing relative to that.

While our approach captures part of the sampling uncertainty—namely, the variance in the derived overlap values—it treats the estimates of $e_t(\mathbf{x}, 0)$ and $\mu(\mathbf{x}, s, t)$ for $s \in \{0, 1\}$ as fixed. Ignoring the uncertainty in these nuisance functions understates the true variability of $G(t)$. Consequently, the reference distribution of $G(t)$ under the null is too narrow, causing the test to reject the null, and classify $(\rho, \gamma)$ pairs as incompatible, more often than a fully bootstrap-based procedure that re-estimates $e_t(\mathbf{x}, 0)$ and $\mu(\mathbf{x}, s, t)$ in every resample. For our sensitivity analysis framework, this potential for an inflated incompatible region is an acceptable (albeit not desirable) behavior. We prefer to label $(\rho, \gamma)$ pairs as incompatible that may, in fact, be compatible rather than risk retaining ones that do not sufficiently explain differences between the study types. Nevertheless, developing a testing scheme that fully accounts for nuisance-estimation uncertainty is an important direction for future work and would strengthen the overall framework.

# E  Algorithms

In this section, we include algorithms for the various components of our paper. We begin with the bias corrected estimator from Section 5, followed be procedures for constructing breakdown frontier plots, like those in Section 6, and for estimating the (in)compatibility of parameter pairs, described in Appendix D. For each algorithm, we also discuss hyperparameter choice considerations and note relevant computational considerations.

## E.1  Bias corrected Estimator

Algorithm 1 outlines the cross-fitting procedure used to implement the bias-corrected estimators defined in Section 5. When estimating population-level ATE, the algorithm is applied to the full dataset. For CATE estimation, the dataset is first filtered to include only those samples whose covariate profiles $\mathbf{X}_i$ fall within the subgroup of interest.

There are two hyperparameters beyond our sensitivity parameters $\rho$ and $\gamma$, namely $\alpha$ and the number of folds $k$. Larger values of $\alpha$ will more closely approximate the hard minimum and maximum operators. However, especially in small sample sizes, large $\alpha$ values can lead to instability in the

bias-correction term. This occurs in regions where the maximum or minimum function switches between its arguments, as the Boltzmann operator exhibits steep gradients at these transitions. For this paper, we use a moderate value of $\alpha = 10$, which balances stability and approximation quality. This choice may be increased in larger datasets.

The value of $k$ controls the number of splits to use for the cross-fitting procedure. Larger $k$ values provide more training data per fold but also increase computational demand. A suitable choice should consider the type of model used to estimate the nuisance functions, the dataset size, and the available computational resources.

Lastly, the choice of model used to estimate the nuisance functions $\hat{\eta}^{(j)}$ is another implicit hyper-parameter. Flexible machine learning models are commonly used, though choices should consider sample size and computational constraints.

In Algorithm 1, we omit the specifics of calculating the uncentered EIF value $\hat{\phi}_{i,t',\rho',\gamma',\alpha'}$ for a given sample $i$, treatment indicator $t'$ and parameter values $\rho', \gamma'$ and $\alpha'$ given the length and complexity of the term. We include that full form in Equation 8, where we dropped the arguments from $\hat{\lambda}_1^{(j)}$, $\hat{\lambda}_2^{(j)}$, $\hat{v}^{(j)}$, and $\hat{w}^{(j)}$ for brevity, but note that they correspond to those functions evaluated at $\mathbf{x}_i$ and parameter values $(t', \rho', \gamma', \alpha')$.

$$
\begin{aligned}
\hat{\phi}_{i,t',\rho',\gamma',\alpha'} =\ & \frac{s_i \cdot \mathbb{I}(t_i = t')}{\hat{e}_{t'}^{(j)}(\mathbf{x}_i, 1)} \left[ y_i - \hat{\mu}^{(j)}(\mathbf{x}_i, 1, t') \right] + s_i \cdot \hat{\mu}^{(j)}(\mathbf{x}_i, 1, t') \\
& + s_i(1 + \gamma') \left\{ \hat{\lambda}_1^{(j)} + \alpha' \hat{\lambda}_1^{(j)} \hat{\lambda}_2^{(j)} \left[ \hat{v}^{(j)} - \hat{w}^{(j)} \right] \right\} \\
& \times \left\{ \frac{\mathbb{I}(t_i = t')}{\hat{e}_{t'}^{(j)}(\mathbf{x}_i, 1) \hat{g}_1^{(j)}(\mathbf{x}_i)} \left[ y_i - \hat{\mu}^{(j)}(\mathbf{x}_i, 1, t') \right] \hat{g}_0^{(j)}(\mathbf{x}_i) \right\} \\
& + (1 - s_i) \left\{ \hat{\lambda}_1^{(j)} \hat{v}^{(j)} + \hat{\lambda}_2^{(j)} \hat{w}^{(j)} \right\} \\
& + (1 - s_i) \left\{ \hat{\lambda}_2^{(j)} + \alpha' \hat{\lambda}_1^{(j)} \hat{\lambda}_2^{(j)} \left[ \hat{w}^{(j)} - \hat{v}^{(j)} \right] \right\} \\
& \times \left\{ \frac{\mathbb{I}(t_i = t')}{\hat{e}_{t'}^{(j)}(\mathbf{x}_i, 0)} \left[ y_i - \hat{\mu}^{(j)}(\mathbf{x}_i, 0, t') \right] \left[ 1 + \rho' \cdot \hat{e}_{1-t'}^{(j)}(\mathbf{x}_i, 0) \right] \right. \\
& \left. + \rho' \cdot \hat{\mu}^{(j)}(\mathbf{x}_i, 0, t') \left[ \mathbb{I}(t_i = 1 - t') - \hat{e}_{1-t'}^{(j)}(\mathbf{x}_i, 0) \right] \right\}
\end{aligned}
\tag{8}
$$

### E.2 Breakdown Fontier Plot

Algorithm 2 outlines the procedure for constructing a breakdown frontier plot like those in Section 6. As with the algorithm for the bias-corrected estimator, when estimating population-level ATE, the algorithm is applied to the full dataset. Whereas, for CATE estimation, the dataset is first filtered to include only those samples whose covariate profiles $\mathbf{X}_i$ fall within the subgroup of interest.

The hyperparameters used are as follows: $\alpha$ and $k$ are inputs to the bias-corrected estimator, $R$ specifies the number of resampling iterations in the incompatibility test, and $c$ denotes the confidence level used throughout the algorithm. The most critical design choice is the grid of $(\rho, \gamma)$ values. Large grids that include many $(\rho, \gamma)$ pairs yield more detailed breakdown frontiers, but increase computational cost. A reasonable approach is to select maximum values of $\rho$ and $\gamma$ based on what a domain expert deems plausible, and then construct a grid of evenly spaced pairs of $(\rho, \gamma)$ between $(0, 0)$ and those maximum values. In our experiments, we consistently cap $\rho$ and $\gamma$ at 0.2, corresponding to a $20\%$ relative violation. While this choice reflects a substantial but interpretable level of assumption violation, selecting a plausible range involves subjective judgment based on the context and domain knowledge.

---

**Algorithm 2** Breakdown Frontier Plot Construction

---

**Require:** Dataset $\mathcal{D}_n$, grid of $(\rho, \gamma)$ values, confidence level $c \in (0, 1)$, Boltzmann smoothing parameter $\alpha \geq 0$, number of cross-fitting folds $k$, number of resampling iterations $R$

**Ensure:** Heatmap assigning each $(\rho, \gamma)$ pair to one of four regions

1: **Estimate nuisance functions once** via cross-fitting (see Algorithm 1, Steps 1–7) to obtain $\hat{\eta} = (\hat{g}, \hat{e}_t, \hat{\mu})$

2: **for** each $(\rho, \gamma)$ in the grid **do**

3:      Compute bias-corrected bounds $(\hat{\theta}^{bc}_{LB}, \hat{\theta}^{bc}_{UB})$ and variances $(\hat{\sigma}^2_{LB}, \hat{\sigma}^2_{UB})$ using Algorithm 1

4:      Construct $100(1-c)\%$ confidence intervals:

$$\mathrm{CI}_{LB} = \left[ \hat{\theta}^{bc}_{LB} - z_{1-c/2} \cdot \frac{\hat{\sigma}_{LB}}{\sqrt{n}}, \ \hat{\theta}^{bc}_{LB} + z_{1-c/2} \cdot \frac{\hat{\sigma}_{LB}}{\sqrt{n}} \right]$$

$$\mathrm{CI}_{UB} = \left[ \hat{\theta}^{bc}_{UB} - z_{1-c/2} \cdot \frac{\hat{\sigma}_{UB}}{\sqrt{n}}, \ \hat{\theta}^{bc}_{UB} + z_{1-c/2} \cdot \frac{\hat{\sigma}_{UB}}{\sqrt{n}} \right]$$

5:      Run incompatibility test (Algorithm 3) for $t = 0$ and $t = 1$ using $R$ resamples

6:      **if** p-value $< c$ for either $t$ **then**

7:          Assign region: `Incompatible`

8:      **else if** both bounds are strictly $> 0$ or strictly $< 0$, and both intervals exclude 0 **then**

9:          Assign region: `Conclusive`

10:     **else if** both point estimates have the same sign, but at least one CI includes 0 **then**

11:         Assign region: `Tentative`

12:     **else if** point estimates have opposite signs **then**

13:         Assign region: `Inconclusive`

14:     **end if**

15: **end for**

16: Render heatmap over the grid with regions color-coded

---

### E.3  (In)compatible $\rho$ and $\gamma$ Test

Finally, we include include Algorithm 3 to outline the steps of the (in)compatibility test. We refer to Appendix D for a detailed explanation and discussion of this procedure.

---

**Algorithm 3** Resampling-based Test for Parameter Compatibility

---

**Require:** Dataset $\mathcal{D}_n = \{(X_i, T_i, S_i, Y_i)\}_{i=1}^n$, parameter values $(\rho', \gamma')$, treatment arm $t'$, number of resamples $R$

**Ensure:** p-value for testing $H_0 : G(t') \geq 0$ vs. $H_A : G(t') < 0$

1: **Compute observed test statistic:**

$$T_{\mathrm{obs}} = \frac{1}{n} \sum_{i=1}^n \left[ \min\{v(X_i, t', \gamma'), w(X_i, t', \rho')\} - \max\{v(X_i, t', -\gamma'), w(X_i, t', -\rho')\} \right]$$

2: **Center overlap values to simulate null:**

$$o_i = \min\{v(X_i, t', \gamma'), w(X_i, t', \rho')\} - \max\{v(X_i, t', -\gamma'), w(X_i, t', -\rho')\} - T_{\mathrm{obs}} \quad \text{for } i = 1, \ldots, n$$

3: **for** $k = 1$ to $R$ **do**

4:      Sample $n$ indices $J_r = \{j_1, \ldots, j_n\}$ with replacement from $\{1, \ldots, n\}$

5:      Compute resampled statistic:

$$T_r = \frac{1}{n} \sum_{j \in J_r} o_j$$

6: **end for**

7: **Compute p-value:**

$$p = \frac{1}{R} \sum_{r=1}^R \mathbb{1}[T_r \geq T_{\mathrm{obs}}]$$

8: **return** $p$

---

# F   Simulation Setup

Each synthetic dataset in Section 6.1 is composed of 2500 i.i.d. samples where for each sample $i$, we start by generating three observable covariates and one unobserved confounder,

$$X_{i,1}, X_{i,2}, X_{i,3} \sim \mathcal{N}(1,1),$$
$$U_i \sim \mathcal{N}(1,1).$$

We then define $C_i = (1 - \beta)X_{i,1} + \beta U_i$, where $\beta$ is a hyperparameter passed to the data generating process (DGP). This parameter controls the level of unobserved confounding, with larger values of $\beta$ corresponding to more unobserved confounding. Using this, we generate the study and treatment indicators as

$$S_i \sim \text{Bernoulli}\Big(\text{expit}(-C_i)\Big), \text{ and}$$

$$T_i \sim \text{Bernoulli}\Big(S_i \times 0.5 + (1 - S_i) \times \text{expit}(C_i)\Big),$$

where expit is the logistic sigmoid: $\text{expit}(x) = \frac{1}{1+e^{-x}}$. The potential outcomes and observed outcome are then generated as

$$Y_i(0) = 100 + X_{i,2},$$
$$Y_i(1) = Y_i(0) + 12C_i - 10X_{i,3} + \tau, \text{ and}$$
$$Y_i = T_i \times Y_i(1) + (1 - T_i) \times Y_i(0) + \epsilon_i,$$

where $\epsilon_i \sim \mathcal{N}(0,1)$ and $\tau$ is another hyperparameter passed to the DGP that controls the size of the constant treatment effect. Therefore, a larger $\tau$ corresponds to a larger treatment effect.

There are five different breakdown frontier plots in Section 6.1. Each plot is generated from a different dataset that is created from the above DGP by varying the two hyperparameters, $\beta$ and $\tau$. In particular,

- Base: $\beta = 0.4$ and $\tau = 5$.
- Larger $\tau$: $\beta = 0.4$ and $\tau = 8$.
- Smaller $\tau$: $\beta = 0.4$ and $\tau = 2$.
- Larger $U$: $\beta = 0.6$ and $\tau = 5$.
- Smaller $U$: $\beta = 0.2$ and $\tau = 5$.

Note that in the plots, "Larger $U$" and "Smaller $U$" refer to higher a lower levels of unobserved confounding induced by the hyperparameter $\beta$, not to the values of the unobserved confounder $U$ itself. With full knowledge of the DGP, more accurate labels would be "Larger $\beta$" and "Smaller $\beta$". But given that the full DGP was not introduced in the main text, we use "Larger $U$" and "Smaller $U$" for simplicity and accessibility.

# G   Additional Simulation Results

We extend the analysis in Section 6.1 by presenting breakdown frontier plots under a variety of data generating scenarios. Specifically, we highlight

- Sample size performance at $n = 250, 500, 1000, 5000,$ and $10000$.
- Sensitivity to the $\alpha$ parameter in the Boltzmann estimator, examining how its effects vary with sample size.
- Robustness to moderate violations of model assumptions.

**Sample Size** Figure 7 presents breakdown frontier plots across these sample sizes. The DGP corresponds to the *Base* case described in Section F, with $\beta = 0.4$ and $\tau = 5$.

The results show the instability of the estimator at smaller sample sizes, reflected in wider and inconsistent tentative regions. However, the general takeaways from the breakdown frontier plot remain consistent, and estimator performance stabilizes quickly as the sample size increases.

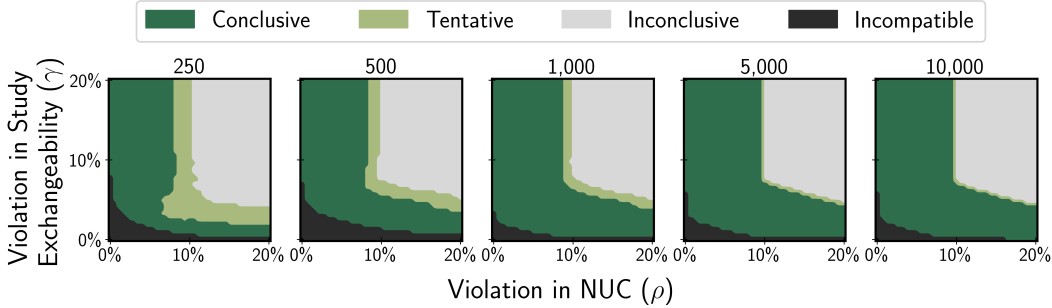

Figure 7: Breakdown frontier plots across various sample sizes ($n$). Figure titles indicate the number of samples in the dataset. All plots are generated under the *Base* DGP settings. Conclusive and tentative regions are distinguished using 95% confidence intervals, computed from the sample variance of the efficient influence function.

$\alpha$ **values** Figure 8 presents breakdown frontier plots across different sample sizes and choices of the $\alpha$ parameter in the Boltzmann estimator. Again, the DGP corresponds to the *Base* case described in Section F, with $\beta = 0.4$ and $\tau = 5$.

The results show that larger values of $\alpha$ tend to produce wider conclusive regions, which is consistent with the Boltzmann operator more closely approximating the max and min operators as $\alpha$ increases. The estimates stabilize for $\alpha > 10$, suggesting this as a generally reliable default choice.

**Model misspecification** To evaluate the robustness of our methodology when model assumptions are violated, we consider a nonlinear DGP but continue to estimate the nuisance functions using linear models. Specifically, we modify the data-generating process in Section **??** as follows:

$$X_{i,1}, X_{i,2}, X_{i,3} \sim \mathcal{N}(0.5, 1.5),$$
$$U_i \sim \mathcal{N}(0.5, 1.5),$$
$$C_i = (1 - \beta)X_{i,1} ** 2 + \beta U_i ** 2 - 1,$$
$$S_i \sim \text{Bernoulli}\Big(\text{expit}(-C_i)\Big), \text{ and}$$
$$T_i \sim \text{Bernoulli}\Big(S_i \times 0.5 + (1 - S_i) \times \text{expit}(C_i)\Big), Y_i(0) = 100 + X_{i,2} ** 2,$$
$$Y_i(1) = Y_i(0) - C_i + X_{i,3} ** 2 + \tau, \text{ and}$$
$$Y_i = T_i \times Y_i(1) + (1 - T_i) \times Y_i(0) + \epsilon_i, \quad \epsilon_i \sim \mathcal{N}(0, 1),$$

and, as before, we vary $\beta$ and $\tau$ to generate five variants of this DGP.

Figure 9 displays breakdown frontier plots under this misspecified setting. As expected, the frontiers are less stable near the boundaries, indicating increased noise due to model misspecification. Nonetheless, the plots still recover informative structure as they delineate regions of compatible and incompatible confounding levels and preserve the qualitative shape of the conclusive and inconclusive regions. This demonstrates that the breakdown frontier can remain informative even when the nuisance models are misspecified.

Figure 10 presents the same designs but with nuisance functions estimated using nonparametric machine learning models (specifically, random forests). In this case, the boundaries are noticeably more stable and the conclusive region expands, consistent with improved approximation of the nonlinear nuisance components. These results suggest that flexible models are advantageous when model assumptions are uncertain and sufficient sample size is available. However, when the analyst is willing to impose parametric structure and sample sizes are limited, parametric models may be preferred due to improved efficiency.

In summary, our findings highlight that the breakdown frontier approach is robust to moderate misspecification and that the choice of nuisance estimator provides a practical trade-off between robustness and efficiency.

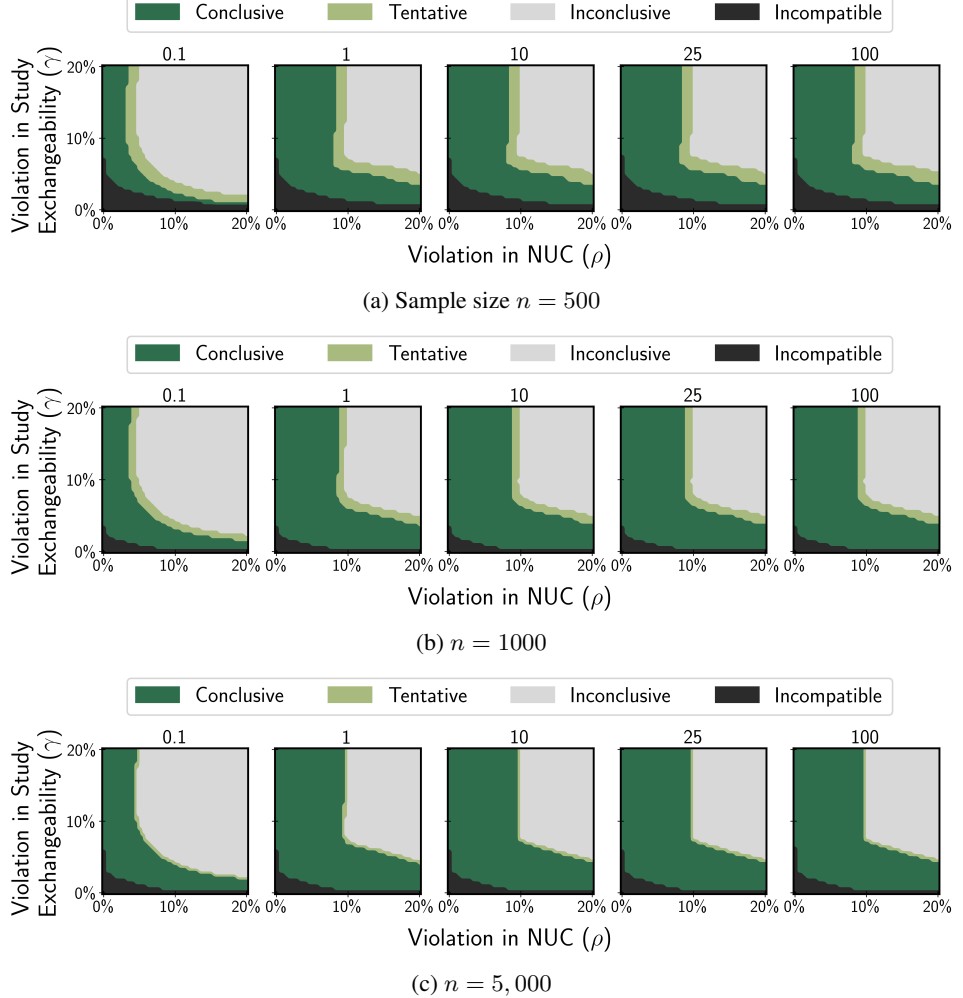

Figure 8: Breakdown frontier plots across various sample sizes ($n$) and $\alpha$ values. Subplot captions indicate the number of samples and individual figure titles indicate the value of $\alpha$ used to generate that plot. All plots are generated under the *Base* DGP settings. Conclusive and tentative regions are distinguished using 95% confidence intervals, computed from the sample variance of the efficient influence function.

# H   Experimental Details

This section provides implementation specifics for the results presented in Section 6. We outline the hyperparameters and other relevant experimental settings used to generate each of the breakdown frontier plots shown in Section 6. Appendix E includes the algorithm for constructing breakdown frontier plots (Algorithm 2), as well as algorithms for the double machine learning estimator (Algorithm 1) and the procedure for determining (in)compatible sensitivity parameters (Algorithm 3). These components together form the full procedure used to construct a breakdown frontier plot.

All experiments were conducted in Python. We reference relevant packages and classes were necessary. The source code can be found in our GitHub repo: `https://github.com/harsh-parikh/Partial-Identification-Data-Fusion`.

**Datasets.** For details on the data-generating process used to create the synthetic datasets in Section 6.1, see Appendix F. For the Project STAR dataset, the outcome of interest was defined as the average score across standardized tests on math, reading, language, and social science. The objective

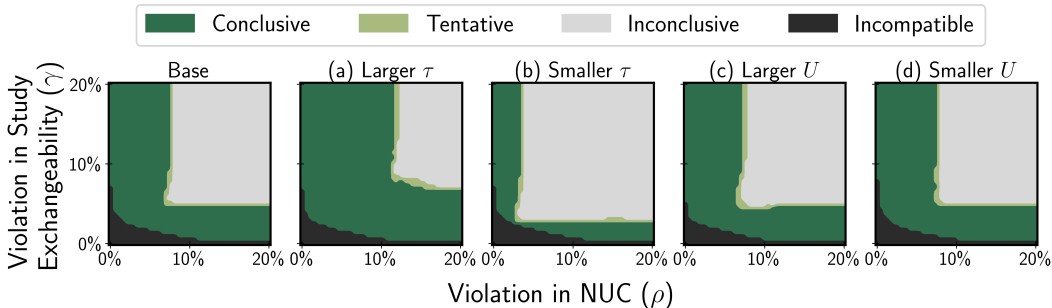

Figure 9: Breakdown frontier plots for various non-linear synthetic datasets generated using linear models to estimate the nuisance functions. Figure titles indicate the relation between the data used to generate that plot to the data used to generate the *Base* plot. Conclusive and tentative regions are distinguished using 95% confidence intervals, computed from the sample variance of the efficient influence function.

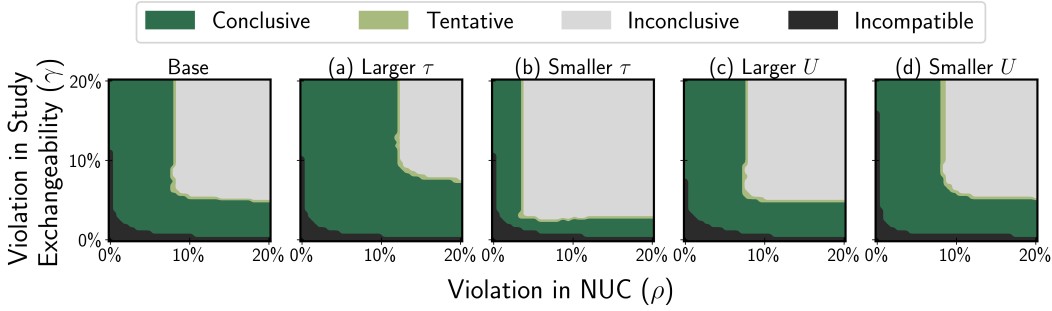

Figure 10: Breakdown frontier plots for various non-linear synthetic datasets generated using nonparametric models to estimate the nuisance functions. Figure titles indicate the relation between the data used to generate that plot to the data used to generate the *Base* plot. Conclusive and tentative regions are distinguished using 95% confidence intervals, computed from the sample variance of the efficient influence function.

was to assess the effect of small class sizes from kindergarten through third grade on this outcome. Measured covariates included gender, race, and age at the start of kindergarten.

For the ATE plot in Figure 3(a), the analysis was conducted on the full dataset. For the CATE plots in Figure 3(b), we restricted the dataset to two subgroups: students who began kindergarten before age six, and students who were at least six years old at the start of kindergarten.

Further details on the Project STAR dataset, including the raw dataset files and cleaning scripts, can be found in the accompanying code.

**Hyperparameter Settings.** Several hyperparameters are held constant across all breakdown frontier plots. Specifically, the following settings are used throughout:

- Grid of $(\rho, \gamma)$: We construct a grid over the $(\rho, \gamma)$ parameter space by taking all pairwise combinations of values sampled uniformly from the intervals $[0, 0.2]$. Specifically, we define:
$$\gamma \in \mathrm{linspace}(0, 0.2, 50), \quad \rho \in \mathrm{linspace}(0, 0.2, 50)$$
where $\mathrm{linspace}(a, b, n)$ denotes a sequence of $n$ evenly spaced values from $a$ to $b$, inclusive. The resulting grid contains $50 \times 50 = 2500$ $(\rho, \gamma)$ pairs.

- Confidence level $c$: 0.95

- Boltzmann smoothing parameter $\alpha$: 10

- Models used to estimate the nuisance functions:

  - $\hat{g}_s^{(j)}(\mathbf{x})$: `sklearn.linear_model.LogisticRegressionCV(n_jobs=1)`

- $\hat{e}_t^{(j)}(\mathbf{x}, s)$: `sklearn.linear_model.LogisticRegressionCV(n_jobs=1)`
- $\hat{\mu}^{(j)}(\mathbf{x}, s, t)$: `sklearn.linear_model.RidgeCV()`

The remaining hyperparameters—namely the number of cross-fitting folds to use for the double machine learning estimators, $k$, and the number of resampling iterations to use for the (in)compatible test, $R$—are set according to the data type. For all breakdown frontier plots based on simulated (Section 6.1), we use $k = 2$ and $R = 100$. For plots based on the Project STAR data (Section 6.2), we use $k = 5$ and $R = 1000$. The larger values for the Project STAR dataset were chosen to generate more precise results.

**Uncertainty Estimation.**    As discussed at the end of Section 5 and in the double machine learning estimator Algorithm 1, the variance of the bounds can be estimated either using the sample variance of the estimated efficient influence functions (EIF) or through resampling methods. For the simulated datasets, we used the sample variance of the EIFs. For the Project STAR dataset, however, we used a bootstrap resampling procedure with 1000 iterations.

The bootstrap approach was chosen for Project STAR due to the relatively small dataset size, specifically in the observational study arm, and the extreme nuisance function estimates it produced—which led to large and unstable variance estimates when relying solely on the EIFs. In contrast, the bootstrap procedure produced more consistent and stable variance estimates.

**Computational Setting and Details.**    All experiments were run on a Slurm-managed cluster using VMware virtual machines, each equipped with an Intel(R) Xeon(R) CPU E5-2699 v4 @ 2.20GHz. No GPU or specialized hardware was used.

For the simulated datasets, we constructed breakdown frontier results by submitting a single Slurm job with 1 CPU core and 32 GB of RAM. Each breakdown frontier plot dataset was generated in a just a few minutes, and compatible region tests (Algorithm 3) were run within the same job.

For the Project STAR dataset, where we estimated uncertainty using 1000 bootstrap resamples, we distributed the work across 20 Slurm jobs, each allocated the same compute resources as the simulated dataset setup. Each job ran 50 iterations with distinct random seeds. The compatible region tests for Project STAR were run in a separate Slurm job using the same resources.

Each job completed in approximately 1-2 minutes, and the overall compute cost was low. The primary limiting factor was the number of bootstrap iterations. All scripts can be run on a local machine with sufficient memory to store the breakdown frontier grids.

---

**Algorithm 1** Bias-corrected estimators for lower and upper bounds

---

**Require:** Dataset $\mathcal{D}_n$ with $n$ samples, sensitivity parameters $\rho, \gamma \geq 0$, Boltzmann operator hyperparameter $\alpha \geq 0$, number of folds $k \in \mathbb{N}$, with $(2 \leq k \leq n)$.

1: Split data into $k$ folds $\{\mathcal{E}_j\}_{j=1}^k$, where $\mathcal{E}_j$ is the set of indices for the samples in the hold-out set for fold $j$, and $\mathcal{T}_j = \{1, \ldots, n\} \setminus \mathcal{E}_j$ is the corresponding training set indices.

2: **for** $j = 1, \ldots, k$ **do**

3:     Estimate the set of nuisance function, $\hat{\eta}^{(j)} = (\hat{g}^{(j)}, \hat{e}_t^{(j)}, \hat{\mu}^{(j)})$, using training data $\mathcal{T}_j$:

4:         $\hat{g}_s^{(j)}(\mathbf{x})$: study propensity function

5:         $\hat{e}_t^{(j)}(\mathbf{x}, s)$: treatment propensity functions for $s \in \{0, 1\}$

6:         $\hat{\mu}^{(j)}(\mathbf{x}, s, t)$: expected outcome functions for $(s, t) \in \{0, 1\}^2$

7:     **for** $i \in \mathcal{E}_j$ **do**

8:         **for** $(t', \rho', \gamma', \alpha') \in \{(1, -\rho, -\gamma, \alpha), (0, \rho, \gamma, -\alpha), (1, \rho, \gamma, -\alpha), (0, -\rho, -\gamma, \alpha)\}$ **do**

9:             Calculate the following values at $\mathbf{x}_i$:

$$\hat{v}^{(j)}(\mathbf{x}_i, t', \gamma') = (1 + \gamma')\hat{\mu}^{(j)}(\mathbf{x}_i, 1, t'),$$

$$\hat{w}^{(j)}(\mathbf{x}_i, t', \rho') = \hat{e}_{t'}^{(j)}(\mathbf{x}, 0)\hat{\mu}^{(j)}(\mathbf{x}, 0, t') + (1 - \hat{e}_{t'}^{(j)}(\mathbf{x}, 0))(1 + \rho')\hat{\mu}^{(j)}(\mathbf{x}, 0, t'),$$

$$\hat{\lambda}_1^{(j)}(\mathbf{x}_i, t', \rho', \gamma', \alpha') = \frac{\exp(\alpha'\hat{v}^{(j)}(\mathbf{x}_i, t', \gamma'))}{\exp(\alpha'\hat{v}^{(j)}(\mathbf{x}_i, t', \gamma')) + \exp(\alpha'\hat{w}^{(j)}(\mathbf{x}_i, t', \rho'))},$$

$$\hat{\lambda}_2^{(j)}(\mathbf{x}_i, t', \rho', \gamma', \alpha') = \frac{\exp(\alpha'\hat{w}^{(j)}(\mathbf{x}_i, t', \rho'))}{\exp(\alpha'\hat{v}^{(j)}(\mathbf{x}_i, t', \gamma')) + \exp(\alpha'\hat{w}^{(j)}(\mathbf{x}_i, t', \rho'))}.$$

10:             Compute and store the plug-in term value, $\hat{b}_{i,t',\rho',\gamma',\alpha'} = \hat{g}_1^{(j)}(\mathbf{x}_i)\hat{\mu}^{(j)}(\mathbf{x}_i, 1, t) + \hat{g}_0^{(j)}(\mathbf{x}_i)\left\{\hat{\lambda}_1^{(j)}(\mathbf{x}_i, t', \rho', \gamma', \alpha')\hat{v}^{(j)}(\mathbf{x}_i, t', \gamma') + \hat{\lambda}_2^{(j)}(\mathbf{x}_i, t', \rho', \gamma', \alpha')\hat{w}^{(j)}(\mathbf{x}_i, t', \rho')\right\}$.

11:             Compute and store the (uncentered) EIF value $\hat{\phi}_{i,t',\rho',\gamma',\alpha'}$ (see Equation 8).

12:         **end for**

13:     **end for**

14: **end for**

15: Compute the lower and upper bound plug-in estimates:

$$\hat{\theta}_{LB}^{plugin}(\rho, \gamma, \alpha; \hat{\eta}) = \frac{1}{n}\sum_i \hat{b}_{i,1,-\rho,-\gamma,\alpha} - \hat{b}_{i,0,\rho,\gamma,-\alpha}, \quad \hat{\theta}_{UB}^{plugin}(\rho, \gamma, \alpha; \hat{\eta}) = \hat{b}_{i,1,\rho,\gamma,-\alpha} - \hat{b}_{i,0,-\rho,-\gamma,\alpha}.$$

16: Compute and store the lower and upper bound (centered) EIF values for each unit $i$:

$$\hat{\phi}_{i,LB,\rho,\gamma,\alpha} = \left(\hat{\phi}_{i,1,-\rho,-\gamma,\alpha} - \hat{\phi}_{i,0,\rho,\gamma,-\alpha}\right) - \hat{\theta}_{LB}^{plugin}(\rho, \gamma, \alpha; \hat{\eta}),$$

$$\hat{\phi}_{i,UB,\rho,\gamma,\alpha} = \left(\hat{\phi}_{i,1,\rho,\gamma,-\alpha} - \hat{\phi}_{i,0,-\rho,-\gamma,\alpha}\right) - \hat{\theta}_{UB}^{plugin}(\rho, \gamma, \alpha; \hat{\eta}).$$

17: Compute the lower and upper bound bias corrected estimates:

$$\hat{\theta}_{LB}^{bc}(\rho, \gamma, \alpha; \hat{\eta}) = \hat{\theta}_{LB}^{plugin}(\rho, \gamma, \alpha; \hat{\eta}) + \frac{1}{n}\sum_i^n \hat{\phi}_{i,LB,\rho,\gamma,\alpha},$$

$$\hat{\theta}_{UB}^{bc}(\rho, \gamma, \alpha; \hat{\eta}) = \hat{\theta}_{UB}^{plugin}(\rho, \gamma, \alpha; \hat{\eta}) + \frac{1}{n}\sum_i^n \hat{\phi}_{i,UB,\rho,\gamma,\alpha}$$

18: Compute bound variance estimates via the sample variance of the EIFs:

$$\hat{\sigma}_{LB}^2 = \frac{1}{n}\sum_i^n \left[\hat{\phi}_{i,LB,\rho,\gamma,\alpha} - \frac{1}{n}\sum_j^n \hat{\phi}_{j,LB,\rho,\gamma,\alpha}\right]^2,$$

$$\hat{\sigma}_{UB}^2 = \frac{1}{n}\sum_i^n \left[\hat{\phi}_{i,UB,\rho,\gamma,\alpha} - \frac{1}{n}\sum_j^n \hat{\phi}_{j,UB,\rho,\gamma,\alpha}\right]^2.$$

19: **return** $\hat{\theta}_{LB}^{bc}(\rho, \gamma, \alpha; \hat{\eta}), \hat{\theta}_{UB}^{bc}(\rho, \gamma, \alpha; \hat{\eta}), \hat{\sigma}_{LB}^2, \hat{\sigma}_{UB}^2$.

---

