# OpenReview forum: "Data Fusion for Partial Identification of Causal Effects"
_NeurIPS.cc/2025/Conference — NeurIPS 2025 poster_

### Official Review · Reviewer_Faet · 2025-06-30

**Clarity:** 3
**Significance:** 3
**Originality:** 3
**Rating:** 5
**Confidence:** 3

**Summary:**

The authors consider a setting where we have collected both randomized and observational data related to some target population for which we would like to estimate a treatment effect. Here, the observational data may have unmeasured confounding and there is also a question whether transportability holds between the populations underlying both the two datasets and the target population. The authors propose in this case an approach to estimate bounds for the ATE and CATE when two sensitivity parameters controls the violations of the no unmeasured confounding assumption and the transportability assumption. They further propose an efficient estimator for the bounds of the ATE. They evaluate their estimator on a set of simulations and demonstrate its use on a real-world dataset.

**Questions:**

- I would like to ask if they can reply to my comments on the study design from my second bullet point under "Weaknesses"?
- Given the claims on line 242-243, can you outline the similarities/differences in applying the same steps as in section 5 to deriving efficient estimators for the CATE? Somehow I would expect that estimating the CATE compared to the ATE should be significantly more difficult, especially if we want to obtain e.g. confidence intervals as you get in Theorem 2.
- I might have missed this point in the paper, but how should we go about selecting the parameter $\alpha$ in practice for the Boltzmann operator and have you looked into how sensitive this choice is for your estimator's performance?

**Ethical Concerns:**

["NO or VERY MINOR ethics concerns only"]

**Final Justification:**

Having read the response from the authors, especially their clarification on the study designs I mentioned under "Weaknesses" I feel confident about increasing the score from 4 to 5. The authors also addressed my other concerns (see rebuttal) and I think this is a solid paper.

**Limitations:**

The authors seem to address all important limitations.

**Paper Formatting Concerns:**

None found.

**Quality:**

3

**Strengths And Weaknesses:**

### Strengths
- The paper is overall well-written. I found most things easy to understand, especially Section 1 to 4.
- I find the approach well-motivated, especially this setting (partial identification in data fusion problems) is definitely overlooked in my opinion and it is nice to see a method like this being proposed. The underlying idea to come up with the bound is relatively simple which I also appreciate.
- I can not judge on the correctness of the proof for deriving the efficient estimator, but assuming that they are correct, I think these are all important properties to have for the estimator and it is nice to see them being derived theoretically.

### Weaknesses
- The simulations are relatively simple, focussing only on how the bounds change across various DGPs. It would have also been interesting better understand some of the statistical properties of the proposed estimator such as sample size efficiency, the robustness of the asymptotic properties in Theorem 2, or its sensitivity to choosing the scale in the Boltzmann estimator.
- There is one thing in the initial setup of the problem which I don't understand, which I think makes it hard for me to fully appreciate in what kind of study designs this method is applicable. In particular, if I interpret things correctly, we have three populations: the population underlying the RCT ($S=1$), the one underlying the observational study $S=0$, and then the target population.  On one hand, the authors say on lines 44-46 that we can get the ATE if transportability holds **or** if there is no unmeasured confounding in the observational study. This makes me think that the population underlying the observational cohort and the target population are the same, because then we would not need transportability if the observational study is unconfounded. But on the other hand, when reading section 2, I interpret it as being that the assumption A5 (which to me is the transportability condition) is needed together jointly with assumption 4 (NUC) if we would identify and estimate the ATE in the target population using the observational data. There seems to be some sort of contradiction here, I would like to hear the authors' thought on this, and in particular if they could explain, for instance using the STAR dataset as an example, how the experimental/observational populations are related to the target population.

Overall, I like this paper. The reason for giving a borderline accept are some of my confusion on the second bullet point under Weaknesses. I'd be happy to re-evaluate my score if the authors are willing to address this point.

---

> ### Author Rebuttal · Authors · 2025-07-30
>
> Thank you for the thoughtful and constructive review. We’re glad to hear you enjoyed the paper and appreciate your openness to revisiting your score. Below, we address your main point of confusion and clarify the intended interpretation of our setup and assumptions. We also respond to your additional comments and outline how we will strengthen the final version of the paper.
>
> **Problem Setup and Population Relationships**
>
> Thank you for raising this important clarification. Our setup involves just two source populations: the RCT cohort (S=1) and the observational cohort (S=0). Our target estimand is defined over the combined population (the mixture of these two), weighted by their respective covariate distributions. However, Our framework is mathematically flexible to accommodate an alternative estimand that chooses either observational (S=0), RCT (S=1) or a separate third population (S=2) as target population. The adaptation requires modifying the marginal covariate distribution in our estimand from $P(\mathbf{X})$ to $P(\mathbf{X}|S=s)$ and adjusting the breakdown frontier calculations accordingly. This maintains all the sensitivity analysis benefits while targeting the population highlighted in your comment. In the revised manuscript, we explicitly present both estimand options (combined populations and specific target population), discuss their appropriate use cases, and demonstrate how our sensitivity analysis framework applies to each. This will better highlight the flexibility of our approach while addressing the valid concern you've raised about target population selection.
>
> Using Project STAR as a concrete example: both cohorts were drawn from Tennessee school districts—the RCT population consisting of students randomized to different class sizes, and the observational population consisting of students assigned non-randomly. Our target population are all students, not just within one cohort.
>
> With this clarification, we can explain the statement in lines 44-46: The ATE over this combined population is point-identified if either:
> 1. The observational data satisfies no unmeasured confounding (NUC), or
> 2. The experimental and observational samples satisfy study exchangeability
>
> This is because we can decompose the CATE as:
> $\mathbb{E}[Y(1) - Y(0) \mid \mathbf{X} = \mathbf{x}] = \sum_{s \in \{0,1\}} \left[ \mathbb{E}[Y(1) \mid \mathbf{X} = \mathbf{x}, S = s] - \mathbb{E}[Y(0) \mid \mathbf{X} = \mathbf{x}, S = s] \right] P(S = s \mid \mathbf{X} = \mathbf{x})$
>
> The only non-identifiable components are the potential outcomes for $S=0$. Either NUC or transportability suffices to identify these terms, which is why we develop sensitivity analysis for scenarios where neither assumption fully holds.
>
> **CATE vs. ATE Estimation Differences**
>
> You are right that precise estimation of bounds for CATE is harder compared to ATE. Regarding lines 242-243 and CATE estimation: The same efficient influence function (EIF) derivation approach applies to both ATE and CATE, but with key differences in implementation. For CATE estimation (for a prespecified $\mathbf{x}$):
>
> 1. We must estimate conditional expectation functions at the covariate value $\mathbf{x}$ of interest rather than averaging across the entire population
> 2. This introduces additional estimation challenges, particularly for high-dimensional $\mathbf{X}$. In particular, it reduces the resultant effective sample size and just inflates the confidence intervals.
>
> While our theoretical framework extends to CATE estimation, practical implementation would rely on stronger smoothness assumptions for tighter uncertainty quantification.
>
> **Selecting $\alpha$ for the Boltzmann Operator**
>
> You've raised an important practical question. We discuss this briefly in Appendix E, but note that it is not referenced clearly in the main text. In our implementation, we found $\alpha$=10 provides a good balance between approximation quality and numerical stability across datasets of moderate size (n = 1000 to 5000). For larger datasets (n>10,000), higher values ($\alpha$=25+) can be used without numerical instability concerns.
>
> Practically, we recommend:
> - Starting with $\alpha$=10 for most applications
> - Testing sensitivity to $\alpha$ by comparing results with $\alpha$=1 and $\alpha$=50
> - For very large datasets, increasing to $\alpha$=50+
> - For small datasets (n<500), using $\alpha$=0.1-1 to prioritize numerical stability
>
> For most practical purposes, we found that our performance is not overly sensitive to moderate changes in $\alpha$, but extremely large values and/or small sample sizes can introduce numerical instability due to the steep gradients in the smooth approximation. In the revision, we will bring this guidance to the main text or add a more explicit reference to Appendix E to ensure it’s easy to find.
>
> **Simulation Study Extensions**
>
> We appreciate your suggestion to expand our simulations. While we focused on demonstrating bound behavior across DGPs, we agree that examining additional statistical properties would strengthen the paper. In the revision, we will add:
>
> 1. Sample size efficiency analysis showing estimator performance at n=250, 500, 1000, 5000, and 10000.
> 2. Robustness checks of the asymptotic properties when model assumptions are moderately violated
> 3. Sensitivity analysis for the $\alpha$ parameter in the Boltzmann estimator, specifically looking at behavior across different sample sizes.
>
> These additional analyses will provide stronger empirical validation of our theoretical results and offer more practical guidance for implementation.

---

> > ### Comment · Reviewer_Faet · 2025-08-03
> >
> > I thank the authors for their replies, which I feel were mostly satisfactory.
> >
> > However, I remain unconvinced by the target estimand defined for the combined population of the RCT and observational data. This concern has also been raised by other reviewers. My concern is that the mixture of two populations is generally not identifiable unless the sampling fractions from the RCT and observational sources are known. Based on my understanding, and drawing on the framework outlined by Dahabreh et al. (2021) regarding nested and non-nested study designs, it seems that your setting falls under the non-nested case. In that scenario, as noted in Table 1 of the reference,  in which case $E[Y(a)]$ is non-identifiable (whereas $E[Y(a) \mid S=s]$ still can be identifiable).
> >
> > But to help clarify my understanding, I would like to ask the authors directly as well: do you consider your study to fall under the nested or non-nested design framework, as outlined by Dahabreh et al. (2021)? If the former, I feel like the estimand for the combined population makes sense, but otherwise I might still be confused by your target estimand.
> >
> > #### References
> > Dahabreh, Issa J., et al. "Study designs for extending causal inferences from a randomized trial to a target population." American journal of epidemiology 190.8 (2021): 1632-1642.

---

> ### Author Response · Authors · 2025-08-03
>
> Fantastic point and you are absolutely spot on. To clarify more precisely: In our setup, due to the internal validity of the RCT, $E[Y(a) \mid S=1, X]$ is always identifiable. However, the reason $E[Y(a)]$ is not point identifiable without additional assumptions is because $E[Y(a) \mid S=0, X]$ remains non-identifiable, in general
>
> This relationship can be expressed mathematically through the law of total expectation:
> $E[Y(a)] = E[E[Y(a) \mid X]] = E[E[Y(a) \mid S=0, X] P(S=0 \mid X) + E[Y(a) \mid S=1, X] P(S=1 \mid X)]$
>
> Since the second term inside the expectation is identifiable but the first term isn't (without additional assumptions), this explains the partial identification challenge. This formulation demonstrates why obtaining partial identification bounds for $E[Y(a)]$ and $E[Y(a) \mid S=0]$ are mathematically synonymous.
>
> As promised, we will update our paper to explicitly incorporate $E[Y(a) \mid S=0]$ as an alternate estimand, with appropriate adaptations to our framework.
>
> Regarding study design: Our work addresses what is often called a "nested" study design. This is precisely the case in our STAR example, where students from the same Tennessee school districts were either part of the randomized trial or the observational component of the study. We will, for sure, clarify this in the paper.
>
> Thank you so much for helping us improve the paper.

---

> > ### Comment · Reviewer_Faet · 2025-08-04
> >
> > Thank you for your follow-up. I'm happy with your response and will increase my score to a 5.
> >
> > I just have one final comment, more as constructive feedback than a critique. I think it's worth emphasising that for the decomposition of  $E[Y(a)]$ to hold, $Pr(S=1\mid X)$ must also be identifiable (hence my questions about nested vs. non-nested study designs).

---

> > > ### Author Response · Authors · 2025-08-04
> > >
> > > Thank you again for your thorough review and continued discussion. We appreciate your feedback on the importance of articulating the specifics of the study design setup and agree that incorporating this into our final version will help clarify the potential applications of the approach.

---

### Official Review · Reviewer_bEkz · 2025-07-02

**Clarity:** 3
**Significance:** 2
**Originality:** 2
**Rating:** 4
**Confidence:** 3

**Summary:**

The paper proposes a new type of sensitivity analysis for causal data fusion to infer bounds on the CATE / ATE when both an unconfounded trial (potentially misrepresenting the population of interest) and a confounded observational dataset are available. Sensitivity models are imposed, bounds are derived, and estimators based on semiparametric efficiency theory are proposed. The authors evaluate their method using both synthetic and real-world data.

**Questions:**

Can the authors provide an example where sensitivity parameters can be chosen by domain knowledge for their sensitivity model?
Can the authors justify their choice of sensitivity model as compared to using an established model such as the marginal sensitivity model (MSM)?
Can the authors compare their modeling choices and results to related works and elaborate on what is novel or similar/ adapted from existing literature?
Can the authors extend their results to CATE/ ATE on the observational dataset?

**Ethical Concerns:**

["NO or VERY MINOR ethics concerns only"]

**Final Justification:**

After the rebuttal I decided to raise my score to 4 (see may responses to the author rebuttal for details).

**Limitations:**

Limitations are discussed in Sec. 7. I would suggest adding limitations regarding the interpretability of the sensitivity parameters, which is a common problem in sensitivity analysis.

**Paper Formatting Concerns:**

No concerns.

**Quality:**

2

**Strengths And Weaknesses:**

Strengths
- Rigorous work with theoretically valid bounds that incorporates semiparametric estimation for increased statistical efficiency
- Data fusion in causal inference is an interesting problem with various potential applications

Weaknesses
- Consistency assumptions are missing in A1-5
- The target quantity of interest seems to be the CATE or ATE averaged over both datasets. That does not seem to be the quantity one would usually be interested in, as it may depend heavily on the selection probability $P(S = 1 | X=x)$. Indeed, the CATE in the RCT and the observational data are not the same (if they were, we would achieve point identification). I suspect that in practice, one would be more interested in the quantity $E[Y(1) - Y(0) | X=x, S=0] (i.e., the CATE on the observational dataset) as the observational dataset generalizes better to the population of interest.
- The related work could be expanded to include recent works on data fusion and partial identification, e.g., https://arxiv.org/abs/2406.02873, https://arxiv.org/abs/2406.02464, https://arxiv.org/abs/2412.11511, https://arxiv.org/abs/2405.07186
- The novelty as compared to related work is not entirely clear. For example, are the proposed sensitivity models novel or adapted from other streams of sensitivity analysis (e.g., on unobserved confounding)? Are the bounds novel or adapted from existing literature? There is a large literature on both sensitivity models and bounds for unobserved confounding. Can the authors justify their choice of sensitivity model as compared to using an established model such as the marginal sensitivity model (MSM)?
- The authors claim that their sensitivity parameters are interpretable, but it is not entirely clear to me how one would interpret a specific choice of sensitivity parameter in practice. Can the authors provide an example where sensitivity parameters can be chosen by domain knowledge for their sensitivity model?

---

> ### Author Rebuttal · Authors · 2025-07-30
>
> Thank you for your detailed review and critical questions. We hope the following notes help clarify some misunderstandings as well as elaborate on feedback we will incorporate into our final version.
>
> **Regarding sensitivity parameter selection by domain experts**
>
> Thank you for this important question. We designed our framework with interpretability in mind, particularly for domain experts who may not be statisticians. In Section 6.2 with the Project STAR dataset, we demonstrate how our parameters translate to meaningful quantities: when $\gamma = 0.05$, this means a confounder would need to shift student scores by more than 30 points on average (on a scale where scores range from 486-745) to invalidate our findings. Such concrete interpretations make the framework accessible to education researchers.
>
> The breakdown frontier approach specifically addresses the challenge of parameter selection by visualizing robustness across a range of values rather than requiring precise point estimates. In practice, domain knowledge informs reasonable parameter ranges—for instance, when observational and RCT participants come from the same hospital system, limiting $\gamma$ to 0-0.05 may be reasonable, while cross-system comparisons might warrant $\gamma$ in 0-0.25 to account for greater heterogeneity.
>
>
> **Choice of sensitivity model and comparison to alternative approaches**
>
> While we build on established sensitivity analysis principles, our approach offers several key innovations for the data fusion context:
>
> 1. First, we developed a dual-parameter framework that simultaneously captures both unmeasured confounding ($\rho$) and study exchangeability violations ($\gamma$) in a unified model. This enables analysis of how these distinct sources of bias interact—revealing that they can either compound or counteract each other, an insight not captured by applying single-dataset sensitivity approaches separately to each data source.
>
> 2. Second, regarding our choice not to use MSM: while MSM effectively parameterizes propensity score sensitivity for a single observational dataset, this approach becomes problematic in multi-dataset settings where propensity scores have fundamentally different interpretations (true randomization probabilities in RCTs versus estimated probabilities in observational data). Our direct parameterization of outcome deviations provides a more consistent interpretation across the fused datasets.
>
> 3. Third, our parameters are deliberately designed as scale-invariant, relative magnitude measures. This allows meaningful interpretation regardless of the outcome scale—a critical feature when combining datasets that may have different measurement properties. For instance, $\gamma = 0.1$ consistently means "up to 10% relative deviation in outcomes" whether we're measuring blood pressure, test scores, or income.
> Our approach thus represents a principled extension of sensitivity analysis to the data fusion setting rather than a direct application of existing single-dataset methods.
>
> **Choice of Target estimand**
>
> Thank you for this important observation. Our target estimand (ATE over the combined populations) was deliberately chosen because it addresses the common data fusion objective of leveraging complementary strengths of both datasets to estimate treatment effects for a broader population. This aligns with the paradigm described by Parikh et al (2023), where RCT findings are extended to wider populations rather than transported to a specific target.
>
> We fully agree that in many applications, the observational population (S=0) may be the primary population of interest. Our framework is mathematically flexible to accommodate this alternative estimand. The adaptation requires modifying the marginal covariate distribution in our estimand from $P(\mathbf{X})$ to $P(\mathbf{X}|S=0)$ and adjusting the breakdown frontier calculations accordingly. This maintains all the sensitivity analysis benefits while targeting the population highlighted in your comment. In the revised manuscript, we explicitly present both estimand options (combined population and observational-only population), discuss their appropriate use cases, and demonstrate how our sensitivity analysis framework applies to each. This will better highlight the flexibility of our approach while addressing the valid concern you've raised about target population selection.
>
> **Missing Consistency assumptions**
>
> Thank you for the suggestion. We note that the standard SUTVA and consistency assumptions are included in the first paragraph of Section 2. Following your recommendation, we have updated our manuscript and added them to our list of assumptions for completeness.
>
> **Literature review expansion**
>
> Thank you for these helpful resources. We agree that these are relevant to our approach and have added them to our literature review in the updated manuscript. In particular, Schweisthal et al. (2024) is similar to our work in that it proposes a partial ID framework for combining multiple datasets, albeit it focuses on combining only observational datasets and considers a different subset of assumption violations. We will clarify this distinction and note the complementarity between these approaches in the final version. Van der Laan et al. (2025) and Wang et al. (2024) are good examples of approaches for facilitating valid inference in our setting, and we will be sure to reference them in our final version. Although, they do not appear to focus on partial identification or sensitivity analysis. We have included an expanded literature review on data fusion methods for causal inference in our appendix, and thank the reviewer for providing these sources as additional relevant literature to include.
>
> References:
> - Parikh, H., Morucci, M., Orlandi, V., Roy, S., Rudin, C., & Volfovsky, A. (2023). A double machine learning approach to combining experimental and observational data. *arXiv preprint arXiv:2307.01449*.

---

> > ### Comment · Reviewer_bEkz · 2025-08-03
> >
> > Thank you for your rebuttal, which has addressed most of my concerns. One last point: I understand the reason for not using an MSM specifically, but there is a wide range of sensitivity models available in the literature. How does the proposed model relate to the broader literature here? For example, "Sensitivity Analysis for Marginal Structural Models", Bonvini et al. also propose outcome-based sensitivity models.
> >
> > I think it is crucial to be precise here, both in terms of crediting existing work and judging novelty (e.g., does most of the novelty come from extending existing sensitivity results to data fusion, or is the proposed approach completely new?)

---

> ### Author Response · Authors · 2025-08-03
> **Novelty and Relationship to Existing Sensitivity Analysis Literature**
>
> Dear Reviewer,
>
> We acknowledge and build upon the vast existing literature on sensitivity analysis while making several distinct contributions:
>
> 1. **Novel data fusion approach:** We uniquely leverage both experimental and observational designs to yield tighter partial identification bounds than would be possible with either data source alone. Our framework explicitly addresses the interaction between different bias types in data fusion contexts.
>
> 2. **Accessible sensitivity parameterization:** Our representation makes sensitivity analysis interpretable to domain experts without specialized statistical training. By parameterizing deviations as relative percentages from standard causal assumptions and combining this with breakdown frontier visualizations, we enable practitioners to intuitively assess the robustness of their conclusions.
>
> 3. **Scale-invariant parameterization:** While approaches like Bonvini et al. (2022) use sensitivity parameters on an absolute scale, our parameters are defined in relative, scale-invariant terms. This critical difference allows for meaningful comparisons across different outcome measures and studies, facilitating meta-analysis and cross-study interpretation.
>
> 4. **Methodological integration:** We bring together existing ideas from sensitivity analysis, data fusion, and efficient influence functions in a novel way that yields practical solutions for real-world problems. This integration bridges previously separate methodological approaches to address increasingly common analytical challenges.
>
> In the revised manuscript, we will include a more extensive literature review that thoroughly contextualizes our work within existing approaches, including important frameworks like those developed by Bonvini and colleagues. This will better highlight both our intellectual foundations and our novel contributions to the field.communication and arguments across contexts.

---

> > ### Comment · Reviewer_bEkz · 2025-08-04
> >
> > Thank you! I still find the distinction to existing sensitivity analysis literature somewhat unsatisfying. While I acknowledge that the authors propose a new version of a sensitivity model, I believe it is crucial to discuss parallels with existing works in terms of mathematical similarity (in both the sensitivity models and bounds).
> >
> > On the other hand, I acknowledge that the paper proposes a complete work, studying partial identification and semiparametric estimation, and may be valuable to the literature. I am happy to increase my score to 4.

---

> > > ### Author Response · Authors · 2025-08-04
> > >
> > > Thank you again for your engagement and thoughtful comments throughout the discussion. We appreciate your willingness to revisit your score and will take your suggestion about clarifying the connection to existing sensitivity models seriously in our final revision.

---

### Official Review · Reviewer_WjYm · 2025-07-03

**Clarity:** 3
**Significance:** 2
**Originality:** 3
**Rating:** 4
**Confidence:** 3

**Summary:**

This paper introduces an approach to sensitivity analysis in the context of estimating effects by combining RCT and observational studies. In essence, there are two main sources of non-identifiability here: unobserved confounding in the observational data, and non-transferability between the experimental and observational settings. The paper introduces a model where each of these can be characterized by a single value, and then applies this to visualize how sensitive study conclusions are to these parameters.

**Questions:**

See strengths and weaknesses. My main hesitation are about the significance of the work, and the degree to which conclusions from the sensitivity analysis rely on an unassessable sensitivity model.

**Ethical Concerns:**

["NO or VERY MINOR ethics concerns only"]

**Final Justification:**

I continue to support acceptance

**Limitations:**

yes

**Quality:**

2

**Strengths And Weaknesses:**

The problem seems well-motivated, the paper is clear, and the visualization apparently easy to interpret. I thought the experimental application was also reasonably well executed.

The main weaknesses in my view are both the relevance of the setting, which seems rather specialized, and the robustness of the sensitivity model.

---

> ### Author Rebuttal · Authors · 2025-07-30
>
> We appreciate your review and would like to address your concerns about the work's significance and the sensitivity model.
>
> **Regarding significance and relevance**
>
> We respectfully disagree that our setting is highly specialized. Data fusion causal inference methods are increasingly crucial across healthcare, economics, and policy research, where combining experimental and observational data is necessary but fraught with challenges. For instance, consider the case of drug repurposing Wegovy or Ozempic for weight loss. Any time one has an experiment to examine a treatment that already exists, we will probably have both.
> Our approach addresses a significant methodological gap: while sensitivity analysis has a rich history in causal inference and data fusion methods are growing rapidly, robust sensitivity frameworks specifically for data fusion settings have been notably absent. Our contribution thus serves an important and expanding research area rather than a niche application.
>
> **Regarding the sensitivity model**
>
> The critique about "unassessable sensitivity models" applies to virtually all sensitivity analysis methods, as their fundamental purpose is to explore the implications of untestable assumptions. The strength of our approach lies in its transparent parameterization:
> 1. We use interpretable parameters ($\rho$ and $\gamma$) that quantify relative deviations in intuitive terms (e.g., $\rho = 0.2$ means confounding shifts outcomes by at most 20%)
> 2. Our parameters are scale-invariant, facilitating meaningful cross-study comparisons
> 3. We take a conservative worst-case bounds approach that avoids optimistic modeling choices
> 4. Our visualization framework enables practitioners to reason clearly about robustness thresholds
>
> Rather than claiming to solve the inherent untestability of assumptions, we provide a principled framework for researchers to assess how strong violations would need to be to invalidate their conclusions. We believe this aligns with the core purpose of sensitivity analysis by enabling researchers to reason transparently about the robustness of their findings.

---

### Official Review · Reviewer_Vp4r · 2025-07-03

**Clarity:** 3
**Significance:** 4
**Originality:** 3
**Rating:** 5
**Confidence:** 5

**Summary:**

This paper contributes to the growing literature on data fusion /
transportability / generalizability, in which data from a randomized
controlled trial (RCT; "internally valid") and observational study
(OS; "externally valid") are to be combined to generalize to a target
population (typically the OS population). The proposed approach does
not require that all confounders are observed in the OS (with
violation of the usual assumption quantified through a sensitivity
parameter $\rho$), and importantly relaxes the very common assumption
that the outcome has the same distribution in the RCT and OS,
conditional on measured covariates and treatment (with violation
quantified through the sensitivity parameter $\gamma$). The
methodology is based on nonparametric partial identification /
sensitivity analysis, where under relaxed assumptions, bounds on
target causal quantities are characterized and identified in terms of
the observed data distribution, and subsequently estimated. The
resulting bound functionals are non-smooth, and the proposed
estimators are *doubly robust* estimators of **smooth approximations**
to these challenging estimands. Finally, the authors demonstrate their
approach in simulations and in a real study of classroom size on
learning outcomes.

**Questions:**

- **Sensitivity Model & Incompatibility**. Can the authors expand on
  why they used a ratio-based sensitivity model, as opposed to simply
  bounding the differences in means? For instance, one might have
  chosen $\rho = \sup_{\mathbf{x},
  t}|\mathbb{E}[Y(t) \mid \mathbf{X} = \mathbf{x},
  S = 0, T = 1 - t] - \mu(\mathbf{x}, 0, t)|$. This would result
  in simpler bounds, but perhaps the authors might argue that this
  model is less interpretable. More importantly, it is unclear why the
  chosen parametrization of sensitivity to study exchangeability
  ($\gamma$) is the appropriate choice. Note that the numerator for
  this quantity, $\mathbb{E}[Y(t) \mid \mathbf{X} =
  \mathbf{x}, S = 0]$, is partially identified. It seems to me rather
  more natural to only parametrize deviations from the unidentified
  component, e.g., $\gamma = \sup_{\mathbf{x}, t}|1 -
  \frac{\mathbb{E}[Y(t) \mid \mathbf{X} = \mathbf{x}, S = 0, T = 1 -
  t]}{\mathbb{E}[Y(t) \mid \mathbf{X} = \mathbf{x}, S = 1, T = 1 -
  t]}|$, where the denominator quantity is identified via
  $\mu(\mathbf{x}, 1, t)$ under the Assumptions A1-A3. This would
  result in a modified bounding scheme, i.e., $v(\mathbf{x}, t,
  \gamma) = e_t(\mathbf{x}, 0)\mu(\mathbf{x}, 0, t) + e_{1 -
  t}(\mathbf{x}, 0)(1 + \gamma)\mu(\mathbf{x}, 1, t)$. Moreover, while
  I am not 100% certain, this parametrization appears more
  compatible with $\rho$ and $w(\mathbf{x}, t, \rho)$, whereby the
  incompatibility issues discussed in Remark 1 and Appendix D could
  be simplified. Any clarity on these issues that the authors can provide I'm
  confident would be appreciated by readers.
- **Sharpness & Bounded Outcomes**. Are the bounds sharp / tight?
  Before the statement of Lemma 1, it is claimed that the bounds are
  "tight". However, it seems that only validity of the bounds is
  proven. In fact, it seems to me that the bounds can be sharpened
  under the assumptions of the paper. Concretely, suppose $b, B > 0$
  is such that $Y \in [b, B]$---the authors assume that "the outcome
  space is _bounded_ and _positive_". Shouldn't we have $v(\mathbf{x},
  t, \gamma) = \min[(1 + \gamma)\mu(\mathbf{x}, 1, t), B]$, and
  $w(\mathbf{x}, t, \rho) = e_t(\mathbf{x}, 0) \mu(\mathbf{x}, 0, t) +
  e_{1 - t}\min{[(1 + \rho)\mu(\mathbf{x}, 0, t), B]}$? Analogous
  lower bound adjustments could be made on the basis of $b > 0$ (there
  are potential issues with the sensitivity model if the outcome could
  take values equal to or very near zero). It would greatly stregthen
  the result if sharpness could be proven. Regardless, a more precise,
  careful discussion of sharpness / tightness will help clarify the
  contribution.
- **Estimation Considerations**. The functionals that identify the
  bounds are challenging, non-smooth (i.e., not pathwise
  differentiable) parameters. The authors choose an interesting smooth
  approximation-based approach, coupled with influence function-based
  estimators. While appropriate, I have some concerns and questions.
  1) **Existing literature**. Perhaps most importantly, examples
     belonging to the class of statistical problems studied here have
     been investigated in recent years, and important works have
     seemingly not been acknowledged---see *References* below. For
     example, Luedtke & van der Laan (2016) studied the functional
     $\mathbb{E}(\max{[\tau(\mathbf{X}), 0]})$, for
     which similar issues arise. They discuss necessary and sufficient
     conditions for pathwise differentiability, and propose procedures
     for efficient confidence interval construction. Bonvini & Kennedy
     (2022) also encounter similar functionals, and impose *margin
     conditions* sufficient to yield pathwise differentiability
     (obviating the need for smooth approximations). Levis et
     al. (2025) and Ben-Michael (2025) also encounter such non-smooth
     functionals, and carefully study different smooth
     approximation-based estimators as well as direct estimators under
     margin conditions. These or related works should likely be cited.
  2) **Approximation error**. For any choice $\alpha > 0$, the
     Boltzmann operator-based smooth approximation bounds exhibit
     error compared to the originally derived non-smooth bounds. While
     Lemma 2 establishes that the approximate bounds converge to the
     original bounds as $\alpha \to \infty$, the result would be
     greatly strengthened by *quantifying* the approximation error for
     any finite $\alpha > 0$---see also Levis et al. (2025) and
     Ben-Michael (2025) for examples of such analyses. There is likely
     a formal trade-off / tension between approximation error and the
     impact of the tuning parameter $\alpha$ on the second order
     remainder for the smooth bounds.
  3) **Nuisance error rates**. The statement of Theorem 2 is, in my
     view, highly misleading, as asymptotic normality is seemingly
     achieved without any conditions on the rates of convergence of
     nuisance estimators $\widehat{g}$, $\widehat{e}$,
     $\widehat{\mu}$. While mentioned in passing in the proof in the
     appendix, these necessary rate conditions need to be explicitly
     stated in the Theorem itself. Moreover, in their proof, the
     authors mention an "empirical process term" and a "second order
     remainder term", but do not actually state what these are. This
     needs to be laid out in greater detail. Importantly, I do not see
     a derivation of the second order bias anywhere in the
     appendices. It is crucial that the form of this bias is
     determined in order to conclude that the given nuisance error
     rates are sufficient for this term to be asymptotically
     negligible.
- **Target Estimand**. Can the authors provide some context for why they
  prefer the CATE, $\mathbb{E}(Y(1) - Y(0) \mid \mathbf{X} =
  \mathbf{x})$, which mixes the RCT and OS populations, rather than
  the effect in the OS population ($S = 0$, often deemed "externally
  valid") on its own? This is not a huge issue, since the proposed
  methodology implicitly bounds this effect, but a more in-depth
  discussion of the chosen estimand would help.


*References*
1. Luedtke AR, van der Laan MJ. Statistical inference for the mean
   outcome under a possibly non-unique optimal treatment
   strategy. Annals of statistics. 2016 Mar 17;44(2):713.
2. Bonvini M, Kennedy EH. Sensitivity analysis via the proportion of
   unmeasured confounding. Journal of the American Statistical
   Association. 2022 Sep 14;117(539):1540-50.
3. Levis AW, Bonvini M, Zeng Z, Keele L, Kennedy
   EH. Covariate-assisted bounds on causal effects with instrumental
   variables. Journal of the Royal Statistical Society Series B:
   Statistical Methodology. 2025 May 27:qkaf028.
4. Ben-Michael E. Partial identification via conditional linear
   programs: estimation and policy learning. arXiv preprint
   arXiv:2506.12215. 2025 Jun 13.

**Ethical Concerns:**

["NO or VERY MINOR ethics concerns only"]

**Final Justification:**

The paper, even as originally submitted, tackles an important problem using a rigorous, theoretically solid approach. I believe the more serious weaknesses identified by me and the other reviewers will be satisfactorily addressed by the authors' proposed revisions. Overall, the paper is solid and should have important implications for causal inference & sensitivity analysis.

**Limitations:**

Yes

**Quality:**

2

**Strengths And Weaknesses:**

**Strengths**
- The paper is well written and clear.
- This work tackles a very important problem, and the goal to relax
  both the unconfoundedness assumption and the study exchangeability
  assumption is commendable.
- The general use of nonparametric partial identification and
  influence function-based estimation is compelling and convincing.
- The breakdown frontier approach is interesting, and well
  illustrated.
- Empirical illustrations (simulations and the Project STAR
  application) demonstrate the approach nicely.

**Weaknesses** (see **Questions** below for full detail on each point)
- The incompatibility issue (Remark 1) resulting from the pair of
  sensitivity models is quite confusing as currently presented, and
  possibly improved with a different sensitivity model
  parametrization.
- Recent relevant literature on non-smooth functional estimation in
  related causal problems has not be acknowledged or cited. These
  works have many insights that could improve this paper.
- The statement of the main estimation result (Theorem 2) is
  misleading and, relatedly, its proof is seemingly incomplete.
- Certain important conceptual and theoretical issues (target
  estimand, sharpness/tightness) are inadequately or imprecisely
  discussed.

---

> ### Author Rebuttal · Authors · 2025-07-30
>
> Thank you for the extremely detailed and thoughtful review. We appreciate the time you took to read the paper closely and provide such substantive feedback.
>
> **Sensitivity Model & Incompatibility:**
>
> We chose a ratio-based sensitivity model in part for its scale-invariance, which allows it to generalize across different outcome scales and covariate profiles. While a difference-in-means model (as you suggest) could also be used, we see this as a modeling choice rather than a structural limitation, and our framework could accommodate such a formulation with mild adjustments. We agree that briefly justifying this design decision in the main text would strengthen the paper.
>
> Regarding our parametrization of study exchangeability violations via $\gamma$, we appreciate your proposed alternative of bounding deviations only from the unidentified component of the observational distribution. However, we chose to define $\gamma$ via a contrast that is marginalized over $T$ because it more directly reflects violations of study exchangeability as differences in potential outcomes across study populations.
>
> To illustrate, suppose that for some covariate-treatment pair $(\mathbf{x}, t)$, there is a nonzero difference in expected potential outcomes between experimental and observational units who received treatment t: $\left| \mathbb{E}[Y(t) \mid \mathbf{X}=\mathbf{x}, S=1, T=t] - \mathbb{E}[Y(t) \mid \mathbf{X}=\mathbf{x}, S=0, T=t] \right| > 0.$ Now, under your alternative parametrization, $\gamma = 0$ would correspond to equality in the unobserved counterfactual group (T = 1−t). But this would not imply overall study exchangeability. The marginal difference, $\left| \mathbb{E}[Y(t) \mid \mathbf{X}=\mathbf{x}, S=1] - \mathbb{E}[Y(t) \mid \mathbf{X}=\mathbf{x}, S=0] \right|$, would remain nonzero (as long as treatment propensities are nonzero). Thus, this alternative $\gamma$ may appear to label settings as “no study exchangeability violation” even when outcome distributions differ between the samples.
>
> We agree that your suggestion is principled and could support a valid alternative model. We chose our $\gamma$ specification because it provides a transparent interpretation of study exchangeability failure that integrates naturally with the interpretation of the rest of our sensitivity framework.
>
> **Target estimand:**
> This is a good question, and a point we plan to clarify further in our final version. Our decision to target the CATE over the combined population was deliberate. This estimand reflects the common goal in data fusion problems to combine complementary information from both the RCT and observational datasets to estimate effects across a broader population. This perspective aligns with the “generalizability” framework outlined by Parikh et al. (2023), where the objective is to extend findings from a randomized study to a larger or more representative cohort, rather than transporting them to an external population.
>
> That said, we fully agree that in many settings the observational population ($S=0$) may represent the primary target of interest. Our methodology is flexible with respect to the target estimand, and can be adapted to this case. Specifically, this involves adjusting the marginal covariate distribution in the estimand from $P(\mathbf{X})$ to $P(\mathbf{X}|S=0)$, with corresponding modifications to the breakdown frontier calculation. All core components of the sensitivity analysis remain applicable. In the revised manuscript, we will discuss both estimand options (combined population and observational-only), clarify their respective use cases, and explain how our framework supports either goal. We believe this addition will better highlight the flexibility of our approach and address the important point you’ve raised.
>
>
> **Estimation Considerations:**
>
> We really appreciate the depth of this feedback and the opportunity to address these estimation considerations. We address each below:
> - (1) *Related literature.*
> Thank you for the helpful references. We agree that these works are highly relevant, and we will cite them and expand our discussion of related approaches to handling non-smooth functionals. In particular, we will better position our contribution in relation to both smooth approximation–based strategies and alternative approaches that avoid smoothing through structural or margin assumptions. We chose a smooth approximation framework (via Boltzmann operators) to preserve pathwise differentiability and enable influence function–based inference in a principled and generalizable way. One of these references, Levis et al. (2023), adopts a similar approximation strategy using the LSE function. We will incorporate a discussion of our approach and highlight the tradeoffs between it, and similar smooth approximation approaches, compared to other approaches that do not employ smooth approximations.
> - (2) *Approximation error.*
> We agree that quantifying finite-$\alpha$ approximation error is an important theoretical question. Our current work focuses on establishing the validity and practical utility of the smooth bounds using Boltzmann approximations, and Lemma 2 guarantees their convergence to the sharp bounds as $\alpha \to \infty$. In practice, we found moderate values of $\alpha$ (i.e. \alpha=10) offered a favorable tradeoff between smoothness (for stable estimation) and tightness (for informative bounds), and we demonstrate their empirical behavior in Section 6. We will also include additional simulation results that explore the sensitivity of our estimator to different $\alpha$ values across varying sample sizes.
> We recognize that a more formal characterization of the bias–variance tradeoff introduced by finite $\alpha > 0$ would strengthen the theoretical guarantees. Prior work (like those you cited) offers promising techniques in this direction, and we will cite these in the final version and highlight this as a compelling direction for future work. However, we view this refinement as complementary to the current contribution, which introduces a general sensitivity framework with valid and efficiently estimable bounds. We believe our approach adds meaningful insight even without a full approximation error analysis, and hope it lays the foundation for future extensions that can incorporate these refinements.
> - (3) *Theorem 2 and Nuisance parameter rates.*
> We thank the reviewer for suggesting that the statement of Theorem 2 should include the regularity conditions and the convergence rate conditions for the nuisance parameter estimators. In the revised version, we will add the necessary rate conditions on $\hat{g}$, $\hat{e}$, and $\hat{\mu}$ into the theorem statement itself. These regularity conditions are analogous to the ones described in Appendix B of Robins, Rotnitzky and Zhao (1994), and Section 3 of Rudolph et al (2025). In the proof, we will also explicitly define the empirical process term, $(\mathbb{P}_n - P)(\hat{\phi} - \phi)$, and the decomposition of the second-order remainder term, $R_n$. We will further clarify the conditions under which the empirical process term is $o_p(n^{-1/2})$, and that the second-order remainder is $o_p(n^{-1/2})$ provided the nuisance estimators converge at sufficient rates (i.e., standard conditions such as $|\hat{\mu} - \mu| \cdot |\hat{g} - g| = o_p(n^{-1/2})$, etc.). While this is expected to follow under standard regularity conditions for EIF-derived estimators of smooth approximations, we agree that a formal derivation of the second-order term is necessary to justify these conditions rigorously, and we will include this in the revised version.
>
> **Sharpness & Bounded Outcomes:**
>
> You are correct that Lemma 1 establishes the validity of our bounds, but not sharpness. This was an error on our part. We will revise the text to reflect this more precisely.
> We appreciate the suggestion to incorporate outcome bounds directly into the sensitivity model, by truncating $v(\mathbf{x}, t, \gamma)$ and $w(\mathbf{x}, t, \rho)$ with known support $[b, B]$, to yield sharper bounds under the model assumptions. That said, we chose not to incorporate additional min/max operations because it would add further non-smooth structure to an already challenging estimation problem. While we acknowledge this could be remedied using a similar smoothing approach to the one we already use (or one of the other existing approaches you reference), our current framework already applies Boltzmann smoothing between the functions $v(\mathbf{x}, t, \gamma)$ and $w(\mathbf{x}, t, \rho)$ that are at the heart of our partial identification method. Introducing further smooth approximations for support truncation would require additional derivations and tuning, which we viewed as beyond the scope of this initial contribution.
> We agree that this is a promising direction, and we will add a discussion in the revised paper clarifying that sharper bounds may be attainable using known outcome support, but that doing so would involve additional modeling and inference challenges we chose not to pursue here. We hope future work can build on our framework to incorporate this work.
>
> References:
> - Parikh, H., Morucci, M., Orlandi, V., Roy, S., Rudin, C., & Volfovsky, A. (2023). A double machine learning approach to combining experimental and observational data. *arXiv preprint arXiv:2307.01449*.
> - Robins, J. M., Rotnitzky, A., & Zhao, L. P. (1994). Estimation of regression coefficients when some regressors are not always observed. *Journal of the American statistical Association, 89*(427), 846-866.
> - Rudolph, K. E., Williams, N. T., Stuart, E. A., & Diaz, I. (2025). Improving efficiency in transporting average treatment effects. *Biometrika*, asaf027.

---

> > ### Comment · Reviewer_Vp4r · 2025-08-01
> >
> > I very much appreciate the authors' serious effort in responding to each of my points, as well as those of the other reviewers. I believe that with the proposed additions, expanded discussion, and other changes, the paper will represent an exciting and important contribution to the literature. I will increase my score to 5.

---

> ### Author Response · Authors · 2025-08-03
> **Thank you so much!**
>
> Thank you so much for your inputs and suggestions. It will surely make our paper stronger and better.

---

### Decision · Program_Chairs · 2025-09-17

**Decision:**

Accept (poster)

**Comment:**

My recommendation is to accept the paper.

The paper proposes a sensitivity analysis and bound estimation framework for inference about ATEs and CATEs in a target population, when an unconfounded trial and a potentially confounded observational study are available on distinct populations, and there are potential violations to exchangeability. The estimation framework targets smooth approximations to non-smooth bound estimands. The authors demonstrate the methodology in simulations and on the STAR study.

Reviewers raised some concerns about choices in the sensitivity model, and, most importantly, connections to the existing literature. These issues were addressed well enough in the rebuttal period to reach consensus for acceptance. However, I would strongly encourage the authors to follow the advice of reviewer bEkz to draw explicit connections between the sensitivity model proposed here and other sensitivity models. This is not just about drawing distinctions to show novelty, but also about drawing deeper conceptual connections or distinctions that could guide future work on sensitivity models.